# Learning to Incentivize on the Fly: Leader-Follower Games with Policy Recommendation

## Abstract

In dynamic and strategic interactions, ensuring incentive compatibility (IC) is essential for achieving stable and predictable outcomes. This paper explores online learning under a novel IC constraint in the context of leader-follower Stackelberg games with policy recommendation. In these games, the leader announces a policy to commit to while recommending a policy for the follower to adopt. The leader's optimal strategy is captured by the Stackelberg equilibrium, where the leader's announced policy maximizes her rewards, and the recommended policy is incentive-compatible—serving as the follower's optimal response given the leader's commitment. We study the online learning problem from the leader's perspective, where she has no prior knowledge of the follower's reward function and hence must infer it solely through observed follower actions. To address this problem, we develop a theory for such games, and propose a provably efficient algorithm that minimizes regrets with respect to both the leader's rewards and violations of the IC constraint. The algorithm integrates the maximum likelihood estimation of the follower's response model with optimistic planning over an estimated IC constraint. Crucially, we establish that our algorithm is robust to misspecification of the follower's behavioral parameters, exhibiting graceful degradation where the performance loss scales linearly with the estimation error. To the best of our knowledge, this is the first provably efficient online learning algorithm for incentive-compatible decision-making, highlighting the potential of online learning in addressing challenges in misaligned multi-agent systems.

## 1 Introduction

Incentive compatibility (IC) is fundamental in leader-follower Stackelberg games, ensuring that agents act as intended within strategic interactions (Myerson, 1979; Hammond, 1979; Roughgarden, 2010). In the context of information design, this concept corresponds to the *obedience constraint* (Bergemann & Morris, 2019; Kamenica & Gentzkow, 2011). In such games, the leader commits to a policy to influence the follower's behavior. Without IC, the follower might deviate, rendering the leader's strategy suboptimal and destabilizing the system. This challenge is particularly acute in online (reinforcement) learning settings, where the leader must adapt dynamically while learning from limited feedback. This paper studies leader-follower interactions under IC constraints within a sequential decision-making framework. The leader operates with limited visibility, unable to observe the follower's rewards, and must infer the follower's response based on observed actions. The follower, constrained by bounded rationality (Simon, 1955; Conlisk, 1996; Jones, 1999), selects policies via entropy-regularized optimization, influenced by the leader's commitments and recommendations.

We model this problem as a leader-follower game (LFG) (Simaan & Cruz Jr, 1973; Bai et al., 2021; Zhao et al., 2023; Zhong et al., 2023). At each time step, both agents choose actions and receive rewards that depend on their joint actions. Each agent aims to adopt an optimal policy that maximizes their expected cumulative return, considering the strategic decisions of the other. To achieve this, an agent must infer how the other adapts their policies based on historical actions and leverage this information to improve their own policy (Zhong et al., 2023; Chen et al., 2023). In this setting, the leader commits to a policy $\pi$ over her action space $\mathcal{A}$ and provides the follower a recommendation policy $y$ over the follower's action space $\mathcal{B}$. After observing $\pi$ and $y$, the follower *strategically selects her best response policy* $\nu^{\pi,y}$ that maximizes her expected rewards by solving a regularized reward process induced by $(\pi, y)$. To achieve IC, the leader aims to recommend a policy $y$ that aligns with

$\nu^{\pi,y}$. We quantify the IC violation (ICV) using the total variation (TV) distance between $\nu^{\pi,y}$ and $y$, which measures the discrepancy between the leader's recommendation and the follower's actual response. A zero ICV implies the leader precisely predicts and guides the follower's response. The leader's objective is to determine her optimal commitment policy $\pi^*$ and recommendation policy $y^*$ such that the policy pair $(\pi^*, \nu^{\pi^*,y^*})$ *optimizes her expected total rewards while ensuring IC*. This optimal configuration constitutes a quantal Stackelberg equilibrium (QSE) (McKelvey & Palfrey, 1995; Başar & Olsder, 1998), representing the solution to a bilevel optimization problem where the leader maximizes her reward subject to the IC constraint that the follower adopts the recommendation.

The introduction of policy recommendation (PR) is particularly relevant in scenarios where the follower (e.g., a small company) has limited computational resources, while the leader (e.g., a large company) possesses ample resources. In such cases, it is reasonable to assume the follower's action space is finite, and she employs computationally efficient methods, such as one-step mirror descent, to compute her response policy. The follower incorporates the recommendation into her decision-making, allowing the leader to guide her behavior via $(\pi, y)$ and making her decision depend on both commitment and recommendation. In Appendix C.1, we provide real-world motivating examples to illustrate the practical significance of PR in leader-follower dynamics.

Our goal is to design provably efficient algorithms for the leader to learn her optimal policy pair $(\pi^*, y^*)$ in an online setting, using only the limited information available (i.e., joint actions and the leader's rewards). A key challenge here is the information asymmetry: the leader observes the follower's actions but not her rewards (Chen et al., 2023). To address this, we establish theoretical properties of such games and propose algorithms that employ a model-based approach to learn the follower's response model through maximum likelihood estimation (MLE), combined with a model-free least-squares value iteration (LSVI) approach to learn the leader's optimal value function (Sutton & Barto, 2018). To achieve efficient learning, we construct confidence sets for the leader's reward function and the follower's response model to promote exploration (Auer et al., 2008), and update the leader's policies using optimistic planning. Furthermore, we address the practical challenge of parameter uncertainty by analyzing the robustness of our algorithm, ensuring graceful degradation in performance under misspecified behavioral parameters.

**Challenges** In addition to the challenge stated above, introducing PR to LFG poses new challenges beyond prior settings. (1) *Theoretical foundation*: PR changes the game structure, requiring a new theoretical framework to ensure IC and efficient policy design. The leader has to ensure (approximate) IC given her estimates of the follower's reward functions, inferred from historical data, which is new in literature. (2) *Recommendation design*: The leader must construct effective recommendations, which introduces a large and complex design space absent in prior work. (3) *Regret analysis*: Unlike standard settings where suboptimality is always nonnegative, in this work, suboptimality can be negative due to IC constraints. Hence, analyzing and controlling the absolute regret becomes necessary for robust performance guarantees, but this is absent in the literature. (4) *Distribution shift*: PR induces a distribution shift between the follower's response and quantal response. It is technically challenging to control the distance between two quantal responses under two different reward functions, and naively applying standard techniques fails to attain sublinear regret bounds.

**Contributions** Our key contributions are stated as follows. (1) *Ensuring IC*: We establish a novel theoretical framework for LFGs with PR, showing that, to ensure IC, the recommendation policy should align with some quantal response, providing a solid foundation for algorithm design. (2) *Algorithmic efficiency*: We prove our algorithm not only achieves sublinear upper bounds on regret and ICV regret but also on absolute regret, ensuring statistical efficiency. Furthermore, we propose a variant that is both statistically and computationally efficient. (3) *Regret analysis*: We leverage the invariance property of the follower's response policies under constant shifts in reward functions, allowing us to derive a sublinear upper bound on the distance between quantal responses, addressing the problem of distribution shift introduced by PR. (4) *Broader implications*: The framework developed for bridging the gap between standard regret and absolute regret offers a general technique applicable to a broader range of constrained online learning problems. (5) *Robustness to misspecification*: We analyze the impact of misspecified behavioral parameters and prove our algorithm exhibits graceful degradation, where the regret bound incurs an additional linear term proportional to the estimation error, confirming our algorithm's reliability in practical settings.

For a more in-depth discussion of the challenges and contributions, please refer to Appendix D.

## 2 RELATED WORK

Our work builds upon the extensive literature on learning in LFG, also known as Stackelberg games, particularly in online learning settings (Bai et al., 2021; Kao et al., 2022; Zhong et al., 2023; Zhao et al., 2023). A key related work is Chen et al. (2023), which addresses the challenge of a leader inferring a boundedly rational follower's policy from historical actions (Simon, 1955). Our work shares the focus on bounded rationality, which, as highlighted by Chen et al. (2023); Haghtalab et al. (2016); Yang et al. (2012), provides a more tractable and realistic model compared to perfect rationality (Letchford et al., 2009; Peng et al., 2019). While Yang et al. (2012) focus on computing optimal strategies in security games with known parameters, our work addresses the online learning problem in general-sum leader-follower games with unknown rewards. Haghtalab et al. (2016) explore bounded rationality in learning adversary models. More recently, Haghtalab et al. (2023) study Stackelberg games where followers best-respond to calibrated forecasts; in contrast, our follower observes and responds to specific policy recommendations.

Our work also connects to online multi-agent RL, especially for equilibrium learning in Markov games. Centralized methods extend single-agent optimism-based techniques (Jin et al., 2020; Bai et al., 2020; Bai & Jin, 2020; Liu et al., 2020; Jin et al., 2021b; Huang et al., 2021; Xie et al., 2020; Chen et al., 2021), while decentralized algorithms rely on adversarial bandit tools (Jin et al., 2021a; Tian et al., 2021; Erez et al., 2022; Liu et al., 2022; Zhan et al., 2022; Mao & Başar, 2023; Wang et al., 2023; Cui et al., 2023). Regarding steering behaviors, Zhang et al. (2024) use monetary payments to steer no-regret learners in known games. In contrast, we utilize policy recommendations to incentivize boundedly rational followers with unknown rewards. Our work refines the bilevel framework of Chen et al. (2023) by incorporating policy recommendation. Although related, our emphasis on incentive-compatible learning presents substantial new technical challenges that necessitate novel techniques detailed in Section 1 and Appendix D. Please refer to Appendix B for further discussions.

Furthermore, our work draws inspiration from Maximum Entropy Inverse RL (Ziebart et al., 2008; Zhu et al., 2023) and energy-based models (Neu & Szepesvári, 2009; Choi & Kim, 2012; Gleave & Toyer, 2022). Crucially, our IC formulation aligns with the *obedience constraint* in mechanism and information design (Myerson, 1983; Kamenica & Gentzkow, 2011; Bergemann & Morris, 2019). Unlike these typically static settings, we address the challenge of learning under obedience constraints with unknown rewards. As estimated constraints allow violations and negative suboptimality, our novel absolute regret analysis is essential for convergence. This also distinguishes our work from Mansour et al. (2015), which explores incentive-compatible bandit exploration in a Bayesian setting.

## 3 PRELIMINARIES AND PROBLEM FORMULATION

### 3.1 LEADER-FOLLOWER GAMES (LFG)

We consider an LFG denoted as $M = \{\mathcal{A}, \mathcal{B}, u, r\}$, where $\mathcal{A}$ and $\mathcal{B}$ are the finite action spaces of the leader and the follower, respectively, and $u, r \colon \mathcal{A} \times \mathcal{B} \to [0, 1]$ are their reward functions. The game proceeds as follows. At each step, (1) the leader announces a policy $\pi \in \Delta(\mathcal{A})$ and recommends a policy $y \in \text{int}(\Delta(\mathcal{B}))$ to the follower; (2) the follower observes $(\pi, y)$ and then selects a response policy $\nu \in \Delta(\mathcal{B})$ according to $(\pi, y)$; (3) the leader and follower execute actions $a \sim \pi$ and $b \sim \nu$, respectively; (4) the follower receives a reward $r(a, b)$, unknown to the leader, and the leader receives a reward $u(a, b)$; (5) the leader learns from online data $\mathcal{D}_t = \{a_i, b_i, \pi_i, y_i, u_i\}_{i \in [t]}$ and update her strategy. The follower's reward function is deterministic and known to herself, but unknown to the leader, while the leader's reward function is unknown to both agents (Chen et al., 2023; Scheid et al., 2024). Hence, the leader has to infer the follower's response model from online data $\mathcal{D}_t$.

We give the definition of linear LFG, which assumes a linear structure in rewards (Xie et al., 2020; Zhang et al., 2021; Chen et al., 2023).

**Definition 3.1** (Linear LFG). An LFG is linear if there exists a known feature function $\phi(\cdot, \cdot) \colon \mathcal{A} \times \mathcal{B} \to \mathbb{R}^d$ such that the reward functions satisfy $u(a, b) = \langle \phi(a, b), \vartheta^* \rangle$ and $r(a, b) = \langle \phi(a, b), \theta^* \rangle$, where $\vartheta^*, \theta^* \in \mathbb{R}^d$ are unknown parameters.

In the tabular setting where $\mathcal{A}, \mathcal{B}$ are finite, we let $\phi(a, b) = e_{(a,b)}$, the canonical basis in $\mathbb{R}^d$ with $d := |\mathcal{A}||\mathcal{B}|$, to ensure notational consistency with linear settings (Jin et al., 2020; Xie et al., 2020;

Chen et al., 2023). The bounded reward assumption implies the parameter space $\Theta = [0, 1]^{|\mathcal{A}||\mathcal{B}|}$. We then rewrite the reward functions as:

$$u_\vartheta(\pi, \nu) = \mathbb{E}_{a\sim\pi, b\sim\nu}[u_\vartheta(a, b)] = \langle u_\vartheta, \pi \otimes \nu \rangle, \ r_\theta(\pi, \nu) = \mathbb{E}_{a\sim\pi, b\sim\nu}[r_\theta(a, b)] = \langle r_\theta, \pi \otimes \nu \rangle, \quad (1)$$

where $(\pi \otimes \nu)(a, b) = \pi(a) \cdot \nu(b)$. Given the true parameters $\vartheta^*$ and $\theta^*$, we simplify notation as $u(a, b) = \langle \phi(a, b), \vartheta^* \rangle$ and $r(a, b) = \langle \phi(a, b), \theta^* \rangle$, making learning the reward functions equivalent to estimating $\vartheta^*$ and $\theta^*$ from $\Theta$.

## 3.2 RESPONSE MODELS AND VALUE FUNCTIONS

**Quantal response model**   In response to the leader's committed policy $\pi$ and recommended policy $y$, the follower aims to maximize her entropy-regularized expected reward:

$$\mu_\theta^\pi \in \arg\max_{\nu\in\Delta(\mathcal{B})} g(\nu|\pi; \theta), \quad \text{where } g(\nu|\pi; \theta) := \mathbb{E}^{\pi,\nu}[r_\theta(a, b)] + \eta^{-1}\mathcal{H}(\nu). \quad (2)$$

Here, $\mathcal{H}(\cdot)$ is the Shannon entropy, and $\mathbb{E}^{\pi,\nu}$ denotes the expectation over the trajectory $(a, b)$ induced by $(\pi, \nu)$. The optimal policy $\mu_\theta^\pi$ solving (2) is named the quantal response (QR) policy against $\pi$ with parameter $\theta$. This formulation follows the bounded rationality model of the follower (Simon, 1955; Conlisk, 1996; Jones, 1999), where she assigns higher probabilities to actions with higher expected rewards rather than deterministically choosing the optimal action. The parameter $\eta > 0$ in (2) reflects the degree of bounded rationality. As special cases, when $\eta \to 0$, $\mu_\theta^\pi$ becomes the uniform distribution over $\mathcal{B}$; when $\eta \to \infty$, $\mu_\theta^\pi$ becomes the optimal policy of the process induced by $\pi$, meaning that the follower is perfectly rational.

**Follower's response model**   However, the follower is on the weaker side compared with the leader: the follower only has access to *limited computational resources*. To account for this limitation, we introduce PR, where the leader provides a recommended policy $y$ to the follower. The follower then updates her policy using a one-step mirror descent approach, leading to the following response model:

$$\nu_\theta^{\pi,y} \in \arg\max_{\nu\in\Delta(\mathcal{B})} G(\nu|\pi, y; \theta), \quad \text{where } G(\nu|\pi, y; \theta) := \mathbb{E}^{\pi,\nu}[r_\theta(a, b)] + \eta^{-1}\mathcal{H}(\nu) - \tau D(\nu||y), \quad (3)$$

where $D(\nu||y)$ is the Kullback–Leibler (KL) divergence between the follower's policy $\nu$ and leader's recommendation $y$. The solution $\nu_\theta^{\pi,y}$ is the follower's response policy against $(\pi, y)$ given parameter $\theta$, which is obtained by a single step of mirror descent, starting from leader's recommendation $y$. The parameter $\tau \geq 0$ controls the degree the follower takes leader's recommendation into consideration. Specifically, when $\tau = 0$, the follower does not consider the leader's recommendation at all.

The KL divergence $D(\nu||y)$ in the follower's response model (3) signifies the influence of the leader's recommendation $y$ on the follower's chosen policy $\nu$. This structure is particularly relevant in scenarios mirroring real-world interactions, such as those between large, well-resourced firms (leaders) and smaller entities (followers) that may lack the computational capacity or comprehensive market insight to independently derive fully optimal strategies. Consequently, these followers might adopt simpler heuristic updates, like one-step mirror descent (OMD), rather than solving a complex optimization problem from scratch.

The response model (3) arises naturally from this premise: it represents the outcome of the follower performing an OMD update to optimize their entropy-regularized expected reward (defined in (2)), using the leader's recommendation $y$ as the reference point for the descent step. As detailed in Appendix C.3, this OMD process inherently introduces KL divergence, which penalizes deviations from the leader's suggested policy. The resulting follower's policy update in (3) thus reflects a practical balance between maximizing her own reward and acceding to the leader's influence, capturing a form of bounded rationality. This formulation is not only mathematically justified but also reflects common dynamics observed in various applications (see Appendix C.1), providing a principled link between behavioral assumptions (like limited computational capacity leading to OMD) and the structure of the model, even though the original objective (2) does not explicitly feature a KL term.

**Uniqueness of the response policy**   Due to the regularization in (3), the response policy $\nu_\theta^{\pi,y}$ is uniquely determined. The expected reward $\mathbb{E}^{\pi,\nu}[r_\theta(a, b)]$ is linear in $\nu$, while $-\tau D(\nu||y)$ and $\eta^{-1}\mathcal{H}(\nu)$ are strictly concave in $\nu$. So $G(\nu|\pi, y; \theta)$ is strictly concave, ensuring a unique maximizer.

**Follower's value function**   Let $r_\theta^\pi(b) = \langle \pi(\cdot), r_\theta(\cdot, b)\rangle_\mathcal{A}$ be the expected reward of the follower induced by $\pi$, and let $\phi^\pi(b) = \langle \phi(\cdot, b), \pi(\cdot)\rangle_\mathcal{A}$ be the feature mapping induced by $\pi$. The follower's

action-value (Q) function $Q_\theta^\pi : \mathcal{B} \to \mathbb{R}$ and value (V) function $V_\theta^{\pi,y} \in \mathbb{R}$ are respectively given by

$$Q_\theta^\pi(b) = r_\theta^\pi(b) = \langle \phi^\pi(b), \theta \rangle, \quad V_\theta^{\pi,y} = (\tau + \eta^{-1}) \log \left\{ \sum_{b \in \mathcal{B}} [y(b)]^{\frac{\tau}{\tau+\eta^{-1}}} \exp \left\{ \frac{Q_\theta^\pi(b)}{\tau + \eta^{-1}} \right\} \right\}. \quad (4)$$

**Characterization of the response policy** The response policy $\nu_\theta^{\pi,y}$ can be expressed as an exponential function of the follower's advantage (A) function $A_\theta^{\pi,y} : \mathcal{B} \to \mathbb{R}$:

$$\nu_\theta^{\pi,y}(b) = [y(b)]^{\frac{\tau}{\tau+\eta^{-1}}} \cdot \exp \left\{ (\tau + \eta^{-1})^{-1} \cdot A_\theta^{\pi,y}(b) \right\}, \quad \text{where} \quad A_\theta^{\pi,y}(b) = Q_\theta^\pi(b) - V_\theta^{\pi,y}. \quad (5)$$

**Recovering the quantal response model** Setting $\tau = 0$ recovers the standard quantal response:

$$\mu_\theta^\pi(b) = \exp \left\{ \eta \cdot A_\theta^\pi(b) \right\}, \quad (6)$$

where the advantage (A) function $A_\theta^\pi : \mathcal{B} \to \mathbb{R}$ and the corresponding value (V) function $V_\theta^\pi \in \mathbb{R}$ are given respectively by $A_\theta^\pi(b) = Q_\theta^\pi(b) - V_\theta^\pi$ and $V_\theta^\pi = \eta^{-1} \log \left( \sum_{b \in \mathcal{B}} \exp \left\{ \eta \cdot Q_\theta^\pi(b) \right\} \right)$.

**Leader's value function** Let $\mathbb{E}^{\pi,\nu_\theta^{\pi,y}}$ denote the expectation taken over $a \sim \pi$ and $b \sim \nu_\theta^{\pi,y}$. The leader's action-value (U) function $U_\vartheta : \mathcal{A} \times \mathcal{B} \to \mathbb{R}$ and value (W) function $W_{\vartheta,\theta}^{\pi,y} \in \mathbb{R}$ are:

$$U_\vartheta(a,b) = u_\vartheta(a,b) = \langle \phi(a,b), \vartheta \rangle, \quad W_{\vartheta,\theta}^{\pi,y} = \mathbb{E}^{\pi,\nu_\theta^{\pi,y}} [U_\vartheta(a,b)] = \langle u_\vartheta, \pi \otimes \nu_\theta^{\pi,y} \rangle. \quad (7)$$

# 4 INCENTIVE COMPATIBILITY, OPTIMIZATION AND EQUILIBRIUM

## 4.1 INCENTIVE COMPATIBILITY AND QUANTAL RESPONSE

In the leader-follower game where the follower's response model is given in (3), it is evident that the leader has more power in the sense that she can strategically adjust the recommendation $y$, which affects the agent's response in a way that favors the leader. To regulate the leader, we additionally impose an *incentive compatibility* constraint on the leader, which means that the leader hopes that her recommendation policy $y$ is close to the follower's true response policy $\nu_{\theta^*}^{\pi,y}$. Here $\theta^*$ is the parameter of the true reward function. In other words, leader's recommendation is indeed close to the follower's best response as in (3). When $y = \nu_{\theta^*}^{\pi,y}$, we say that $(\pi, y)$ satisfies **incentive compatibility** given leader's policy $\pi$. In this case, we can show (in Theorem 4.1 below) that this particular $y$ coincides with the follower's *quantal response* given $\pi$, which is defined in (2). Thus, by requiring that the leader recommends an incentive compatible $y$, we ensure that follower's adopts the quantal response policy, which is a version of optimal policy for the follower.

**IC violation** To quantify to what extent leader's recommendation $y$ misses the IC condition, we introduce concept of **Incentive Compatibility Violation** (ICV) as follows. Given parameter $\theta \in \Theta$, for any leader's policy $\pi$ and recommendation policy $y$, the ICV is defined as the total variation (TV) distance (see (20)) between $y$ and $\nu_\theta^{\pi,y}$, i.e.,

$$\mathrm{ICV}(\pi, y; \theta) := \mathrm{TV}(y, \nu_\theta^{\pi,y}). \quad (8)$$

$\mathrm{ICV}(\pi, y; \theta)$ measures the discrepancy between the leader's recommendation policy $y$ and the follower's response policy against the leader's policy $\pi$ and recommendation $y$, given parameter $\theta$. By definition, ICV is nonnegative, and $\mathrm{ICV}(\pi, y; \theta) = 0$ if and only if $y = \nu_\theta^{\pi,y}$. In this case, we say $(\pi, y)$ is $\theta$-IC. Given a parameter $\theta$, we define a subset of $\Pi \times \mathcal{Y}$: $\mathcal{Z}_\theta := \{ (\pi, y) \in \Pi \times \mathcal{Y} : \mathrm{ICV}(\pi, y; \theta) = 0 \}$, which contains all policy pairs satisfying the $\theta$-IC condition.

**Zero ICV leads to quantal response** We introduce a proposition which proves that the quantal response policy is a fixed point of the mapping $y \to \nu_\theta^{\pi,y}$.

**Proposition 4.1** (Quantal response as a fixed point)**.** *Given any $\theta \in \Theta$ and $\pi \in \Pi$, the quantal response $\mu_\theta^\pi$ is the unique fixed point of the mapping $\nu_\theta^{\pi,\cdot} : \mathrm{int}(\Delta(\mathcal{B})) \to \mathrm{int}(\Delta(\mathcal{B}))$. In other words, $y = \mu_\theta^\pi$ if and only if $y = \nu_\theta^{\pi,y}$. Furthermore, we have $V_\theta^\pi = V_\theta^{\pi,\mu_\theta^\pi}$.*

Due to the introduction of a recommendation policy $y$, $V_\theta^\pi = V_\theta^{\pi,y}$ does not hold in general. However, Theorem 4.1 shows that $V_\theta^\pi = V_\theta^{\pi,y}$ holds when $y = \mu_\theta^\pi$. This implies that the introduction of a recommendation policy that is equal to the quantal response does not change the value function.

Thus, the requirement of IC constraint also justifies the assumption of the follower's response model in (3). When the leader announces $(\pi, y)$, it is unclear why the follower is willing to adopt $\nu_{\theta^*}^{\pi,y}$ in (3) at first sight. But when the follower additionally believes that the leader aims to achieve IC, then it is natural for the follower to commit to (3) — doing so will lead to the quantal response policy when IC is satisfied. Because the quantal response policy is a sensible choice for the follower, he is thus willing to trust leader's recommendation and play according to (3). On the other hand, without imposing IC constraints, the leader's inference of follower responses can be systematically manipulated, leading to unreliable estimation and rapidly growing regret (Gan et al., 2019). These justify the validity of the follower's response model and the necessity of IC constraint for the leader.

## 4.2 POLICY OPTIMIZATION WITH IC CONSTRAINT

**Leader's Goal** The leader's goal is to find the optimal policy pair $(\pi^*, y^*)$, such that the following two conditions are satisfied: (i) $(\pi^*, y^*)$ is IC under the true model with parameter $\theta^*$, (ii) $(\pi^*, y^*)$ optimizes leader's expected local return. Mathematically, this problem can be formulated in the form of a bilevel optimization problem with the IC constraint:

$$\max_{\pi,y} \mathbb{E}^{\pi \otimes \nu_{\theta^*}^{\pi,y}}[u_{\vartheta^*}(a, b)], \quad \text{such that } \mathrm{ICV}(\pi, y; \theta^*) = 0, \tag{9}$$

where the follower's true response policy $\nu_{\theta^*}^{\pi,y}$ is characterized by (3) with $\theta = \theta^*$.

**Quantal Stackelberg equilibrium** By Theorem 4.1, we know that when $y$ satisfies IC condition with parameter $\theta^*$, we have $\pi \otimes \nu_{\theta^*}^{\pi,y} = \mu_{\theta^*}^{\pi}$. Therefore, by (9), the optimal $\pi^*$ can be equivalently written as the solution to the following optimization problem:

$$\max_{\pi} \mathbb{E}^{\pi \otimes \mu_{\theta^*}^{\pi}}[u_{\vartheta^*}(a, b)]. \tag{10}$$

Thus, $\pi^*$ corresponds to the leader's policy under the **Quantal Stackelberg Equilibrium** (QSE) of the LFG (McKelvey & Palfrey, 1995; Başar & Olsder, 1998). That is, $\pi^*$ is the leader's reward-maximizing policy when the follower adopts the quantal response policy.

In summary, the optimal solution $(\pi^*, y^*)$ of (9) is characterized by the following two conditions: (i) $(\pi^*, y^*)$ is a QSE of the leader-follower game, and (ii) $y^*$ is a fixed point of the mapping $y \to \nu_{\theta^*}^{\pi^*,y}$. When we write $\nu^* := \nu_{\theta^*}^{\pi^*,y^*}$, we have $\mu_{\theta^*}^{\pi^*} = y^* = \nu^* = \nu_{\theta^*}^{\pi^*,y^*}$. Since $y^* = \nu^*$ for the QSE, we can equivalently write the QSE as $(\pi^*, y^*)$ or $(\pi^*, \nu^*)$.

## 4.3 SUBOPTIMALITY AND ABSOLUTE REGRET

**Suboptimality** For any $(\pi, y)$, we define $J(\pi, y) := \langle u_{\vartheta^*}, \pi \otimes \nu_{\theta^*}^{\pi,y} \rangle$, which is equal to the leader's expected reward when she adopts $(\pi, y)$, given that the follower responds according to (3). Recall that we let $\mathcal{Z}_{\theta^*}$ define the set of $(\pi, y)$ pairs satisfying the $\theta^*$-IC condition. By (9), we can equivalently write $(\pi^*, y^*)$ as the solution to the optimization problem $\max_{(\pi,y) \in \mathcal{Z}_{\theta^*}} J(\pi, y)$. Thus, to measure the performance of $(\pi, y)$, we define the suboptimality of the policy pair $(\pi, y)$ as

$$\mathrm{SubOpt}(\pi, y) := \max_{(\pi',y') \in \mathcal{Z}_{\theta^*}} J(\pi', y') - J(\pi, y) = \max_{\pi' \in \Pi} \langle u_{\vartheta^*}, \pi' \otimes \mu_{\theta^*}^{\pi'} \rangle - \langle u_{\vartheta^*}, \pi \otimes \nu_{\theta^*}^{\pi,y} \rangle. \tag{11}$$

In particular, when $(\pi', y') \in \mathcal{Z}_{\theta^*}$, by Theorem 4.1 we know that $y' = \nu_{\theta^*}^{\pi',y'} = \mu_{\theta^*}^{\pi'}$. Hence, we can use the suboptimality in (11) to assess the quality of each $(\pi, y)$ pair. In particular, $(\pi, y)$ is optimal if and only if $(\pi, y) \in \mathcal{Z}_{\theta^*}$ (equivalently, $\mathrm{ICV}(\pi, y; \theta^*) = 0$) and $\mathrm{SubOpt}(\pi, y) = 0$.

**Online interaction protocol** In the online setting, the leader learns $\pi^*$ and $y^*$ by interacting with the agent for $T$ episodes, without prior knowledge or data. For any $t \in [T]$, in the $t$-th episode, the leader announces and commits to $\pi_t$, recommends $y_t$ to the follower, and the follower adopts her response $\nu_{\theta^*}^{\pi_t,y_t}$. Then, the leader uses the new dataset $\{(a_i, b_i, \pi_i, y_i, u_i)\}_{i \in [t]}$ to update her strategy.

**Standard regret and ICV regret** We study online learning on behalf of the leader, without access to the reward functions of both agents. To measure the performance of a learning algorithm, we introduce the standard regret based on the suboptimality in (11) and the **ICV regret**, which are defined respectively as

$$\mathrm{Reg}(T) := \sum_{t \in [T]} \mathrm{SubOpt}(\pi_t, y_t), \quad \mathrm{Reg}^{\mathrm{ICV}}(T) := \sum_{t \in [T]} \mathrm{ICV}(\pi_t, y_t; \theta^*). \tag{12}$$

Here, $\text{Reg}(T)$ measures the cumulative suboptimality in terms of (11) and $\text{Reg}^{\text{ICV}}(T)$ measures the cumulative violation of the IC constraint.

In general, we cannot exactly estimate the true parameter $\theta^*$, so we cannot exactly reconstruct the set $\mathcal{Z}_{\theta^*}$ of all IC policy pairs, and cannot guarantee $(\pi_t, y_t) \in \mathcal{Z}_{\theta^*}$ for all $t \in [T]$. Therefore, apart from evaluating the standard regret, we have to evaluate the extent to which the constraint is violated. We can use ICV in (8) to quantify the extent to which $(\pi, y)$ violates the IC constraint in (9). So it is a natural idea to define ICV regret as the sum of $\text{ICV}(\pi_t, y_t; \theta^*)$ over all $t \in [T]$ in (12).

**Absolute regret**  Moreover, as $(\pi_t, y_t) \in \mathcal{Z}_{\theta^*}$ is not necessarily true, it is possible that $\text{SubOpt}(\pi_t, y_t)$ is negative for some $t$. This can be true when $(\pi_t, y_t)$ violates the IC constraint. Thus, when $\text{Reg}^{\text{ICV}}(T)$ is strictly positive, having a sublinear $\text{Reg}(T)$ does not imply that the learning algorithm finds approximately optimal solutions for the leader in the long run. For example, it is possible that $\text{SubOpt}(\pi_t, y_t) = -1$ for all $t$, and in this case $\text{Reg}(T) = \widetilde{\mathcal{O}}(\sqrt{T})^1$, but it does not necessarily guarantee good performance. Therefore, since $\text{SubOpt}(\pi_t, y_t)$ in (12) is not necessarily nonnegative, we further introduce the **absolute regret**, defined as the sum of the absolute values of suboptimality:

$$\text{Reg}^{\text{abs}}(T) := \sum_{t \in [T]} |\text{SubOpt}(\pi_t, y_t)|. \tag{13}$$

The goal of online learning is to output a sequence of policy pairs $\{(\pi_t, y_t)\}_{t \in [T]}$ such that the absolute regret and ICV regret satisfy $\text{Reg abs}(T) = \widetilde{\mathcal{O}}(\sqrt{T})$ and $\text{Reg}^{\text{ICV}}(T) = \widetilde{\mathcal{O}}(\sqrt{T})$, respectively. When this is true, we have $\max\{\text{Reg abs}(T)/T, \text{Reg}^{\text{ICV}}(T)/T\}$ converges to zero as $T$ goes to infinitely. In other words, the randomized policy $(\widetilde{\pi}, \widetilde{\nu})$, which is equal to each $(\pi_t, \nu_t)$ uniformly random, constitutes an approximate QSE of the leader-follower game. In Appendix 5, we will provide the methodology and a well-formulated algorithm to achieve this goal.

Furthermore, since $\text{Reg}(T) \leq \text{Reg}^{\text{abs}}(T)$, $\text{Reg}^{\text{abs}}(T) = \widetilde{\mathcal{O}}(\sqrt{T})$ is a strictly stronger result than $\text{Reg}(T) = \widetilde{\mathcal{O}}(\sqrt{T})$. Consequently, there exists a considerable gap between controlling $\text{Reg}(T)$ and $\text{Reg}^{\text{abs}}(T)$. To bridge this gap, we establish Theorem 6.2, which shows that if both $\text{Reg}(T) = \widetilde{\mathcal{O}}(\sqrt{T})$ and $\text{Reg}^{\text{ICV}}(T) = \widetilde{\mathcal{O}}(\sqrt{T})$ hold, then it follows that $\text{Reg}^{\text{abs}}(T) = \widetilde{\mathcal{O}}(\sqrt{T})$. The condition $\text{Reg}^{\text{ICV}}(T) = \widetilde{\mathcal{O}}(\sqrt{T})$ is a key ingredient for bridging this gap. This observation justifies the importance of controlling the ICV regret $\text{Reg}^{\text{ICV}}(T)$ in constrained optimization settings.

## 5 METHODOLOGY

Our method combines estimating the reward functions of both agents and optimistic planning based on these estimates, leveraging ideas from LSVI (Sutton & Barto, 2018) and MLE (Chen et al., 2023).

### 5.1 REWARD FUNCTION ESTIMATION

**Leader's reward function**  For updating the leader's reward function (U function), we add a bonus term to the result of the following ridge regression to encourage exploration (Sutton & Barto, 2018):

$$\widehat{\omega}_t = \underset{\omega \in [0,1]^d}{\arg\min} \left\{ \sum_{i=1}^{t-1} (\langle \phi(a_i, b_i), \omega \rangle - u_i)^2 + \|\omega\|_2^2 \right\}, \quad \widetilde{U}_t(a, b) = \langle \phi(a, b), \widehat{\omega}_t \rangle + \Gamma_t^1(a, b), \tag{14}$$

where we choose the bonus term $\Gamma_t^1(a, b) = \alpha \sqrt{\phi(a, b)^\top (\Lambda_t)^{-1} \phi(a, b)}$. We can solve the ridge regression in (14) by $\widehat{\omega}_t = (\Lambda_t)^{-1} \sum_{i=1}^{t-1} \phi(a_i, b_i) u_i$, where $\Lambda_t := \sum_{i=1}^{t-1} \phi(a_i, b_i) \phi(a_i, b_i)^\top + I_d$. The coefficient $\alpha$ will be specified in the statement of Theorem 6.1.

**Follower's reward function**  As the leader cannot observe the follower's rewards, it is necessary to estimate the follower's reward function. Note that the follower adopts the response model in (3), whose closed-form expression is in (5). Thus, when the leader announces $(\pi_t, y_t)$ and observes the

---

[1] Notational clarification: In general, $\text{Reg}(T) = \widetilde{\mathcal{O}}(\sqrt{T})$ means there exist a constant $C > 0$ and some $T_0$ such that for all $T \geq T_0$, it holds that $|\text{Reg}(T)| = C\sqrt{T}$ up to a logarithmic factor. In this paper, for simplicity, we write $\text{Reg}(T) = \widetilde{\mathcal{O}}(\sqrt{T})$ to mean the one-sided bound $\text{Reg}(T) \leq C\sqrt{T}$, without absolute values.

follower's action $b_t$, we know that $b_t$ is generated from a statistical model given in (5), with parameter $\theta^*$. As a result, before the beginning of the $t$-th round, we can estimate $\theta^*$ based on historical data $\{(\pi_i, y_i, b_i)\}_{i \in [t-1]}$. Specifically, suppose $\theta \in \Theta = [0,1]^d$ is the parameter for the follower's reward function. With online data $\{(\pi_i, y_i, b_i)\}_{i \in [t-1]}$ available to the leader, let $\widehat{\theta}_t^{\mathrm{MLE}}$ be the MLE that minimizes the negative log-likelihood function $\mathcal{L}_t(\cdot)$ for the follower's behavior at time $t$, i.e.,

$$\widehat{\theta}_t^{\mathrm{MLE}} \leftarrow \underset{\theta \in [0,1]^d}{\arg\min}\, \mathcal{L}_t(\theta), \quad \text{where} \quad \mathcal{L}_t(\theta) = - \sum_{i \in [t-1]} \log \nu_\theta^{\pi_i, y_i}(b_i). \tag{15}$$

The follower's response mapping can be estimated from the follower's historical actions via MLE, like the quantal response mapping in Chen et al. (2023).

In order to achieve sample efficiency in the online setting, we need to leverage the optimism principle (Abbasi-Yadkori et al., 2011), which further necessitates uncertainty quantification. Therefore, we aim to construct a confidence set that contains $\theta^*$ with high probability (Chen et al., 2023). To this end, we define a confidence set for the parameter $\theta$ of the follower's reward function:

$$\mathcal{C}_\Theta^t(\beta) = \left\{ \theta \in \Theta : \mathcal{L}_t(\theta) \leq \inf_{\theta' \in \Theta} \mathcal{L}_t(\theta') + \beta \right\}, \tag{16}$$

where the parameter $\beta > 0$ measures the relaxation of MLE. Then, we can choose a parameter $\widehat{\theta}_t$ from this confidence set $\mathcal{C}_\Theta^t(\beta)$, and the estimated reward function for the follower is taken as $\widehat{r}_t = r_{\widehat{\theta}_t}$, with $r_{\widehat{\theta}_t}(a, b) = \langle \phi(a, b), \widehat{\theta}_t \rangle$ for any $(a, b) \in \mathcal{A} \times \mathcal{B}$.

## 5.2 Optimistic value iteration

Based on the estimated reward functions in Section 5.1, we then employ the following two schemes of optimistic value iteration, to generate the policy pair $(\pi_t, y_t)$ for the leader at time $t \in [T]$.

**Scheme 1:** $\quad (\widehat{\theta}_t, \pi_t) \leftarrow \underset{\theta \in \mathcal{C}_\Theta^t(\beta), \pi \in \Pi}{\arg\max} \, \langle \widetilde{U}_t, \pi \otimes \mu_\theta^\pi \rangle, \quad y_t \leftarrow \mu_{\widehat{\theta}_t}^{\pi_t}, \tag{17}$

**Scheme 2:** $\quad \pi_t \leftarrow \underset{\pi \in \Pi}{\arg\max} \langle \widetilde{U}_t, \pi \otimes \mu_{\widehat{\theta}_t^{\mathrm{MLE}}}^\pi \rangle + C_{\mathrm{ii}} \Gamma_t^2(\pi, \mu_{\widehat{\theta}_t^{\mathrm{MLE}}}^\pi; \widehat{\theta}_t^{\mathrm{MLE}}), \quad y_t \leftarrow \mu_{\widehat{\theta}_t^{\mathrm{MLE}}}^{\pi_t}, \tag{18}$

where $C_{\mathrm{ii}}$ is a constant depending on $\eta$ and $\tau$ as defined in (33), and the bonus function is defined by

$$\Gamma_t^2(\pi, y; \theta) = 2(\varepsilon \xi + C_\varepsilon^{(2)} \xi^2), \quad \text{where} \quad \xi = \left( (8C_\varepsilon^2 \beta + 4B_\Theta^2) \operatorname{Tr} \left( \left( \Sigma_t^\theta + I_d \right)^\dagger \Sigma^{\pi, y, \theta} \right) \right)^{\frac{1}{2}}, \tag{19}$$

with $\varepsilon = (\tau + \eta^{-1})^{-1}$ and $C_\varepsilon^{(2)} = \varepsilon^2 \exp(2\varepsilon B_A)(2 + \varepsilon B_A \exp(2\varepsilon B_A))/2$. Here, $\Sigma_t^\theta = \sum_{i=1}^{t-1} \operatorname{Cov}^{\pi_i, y_i, \theta}[\phi^{\pi_i}(b)]$ is the covariance matrix, and $\Sigma^{\pi, y, \theta} = \operatorname{Cov}^{\pi, y, \theta}[\phi^\pi(b)]$ with the expectation taken by $b \sim \nu_\theta^{\pi, y}$. The coefficient $\beta$ will be specified in the statement of Theorem 6.1.

The design of Scheme 1 (17) is inspired by Theorem 4.1, which claims $y = \mu_\theta^\pi$ if and only if $y = \nu_\theta^{\pi, y}$. In (17), we use the estimated parameter $\widehat{\theta}_t$ instead of the true parameter $\theta^*$ as it is unknown. Recall that $(\pi_t, y_t)$ being IC requires $y_t = \nu_{\theta^*}^{\pi_t, y_t}$ instead of $y_t = \nu_{\widehat{\theta}_t}^{\pi_t, y_t}$, so in general, $(\pi_t, y_t)$ is not an IC policy pair. However, we will show that $(\pi_t, y_t)$ tends to be an approximately IC policy pair in the sense that the absolute regret can be sublinear in $T$. As a variant of Scheme 1, Scheme 2 is more computationally efficient than Scheme 1. This is because Scheme 1 needs to explore different values of $\theta$ within $\mathcal{C}_\Theta^t(\beta)$, while Scheme 2 directly uses the MLE $\widehat{\theta}_t^{\mathrm{MLE}}$ without exploring $\mathcal{C}_\Theta^t(\beta)$.

We summarize our proposed algorithm named **P**olicy **R**ecommendation with **O**ptimistic Value Iteration and **M**aximal Likelihood Estimation (**PROM**) as follows.

---

**Algorithm 1** Policy Recommendation with OVI and MLE (PROM)

---

1: Input parameters $\eta, \tau, \alpha, \beta$. Initialize the dataset $\mathcal{D} = \varnothing$.
2: **for** $t = 1, \ldots, T$ **do**
3:      Update $\widetilde{U}_t$ by (14) and truncate it to $[0, 1]$. (Only for Scheme 2) Obtain the MLE $\widehat{\theta}_t$ by (15).
4:      Solve $(\pi_t, y_t)$ as the optimal solution to Scheme 1 (17) or Scheme 2 (18).
5:      Announce $\pi_t$ and recommend $y_t$ to the follower. Observe the follower's action $b_t$, choose an action $a_t \sim \pi_t$ and receive a reward $u_t$.
6:      Collect a trajectory $\tau_t = \{(\pi_t, y_t, a_t, b_t, u_t)\}$. Update the dataset $\mathcal{D} \leftarrow \mathcal{D} \cup \{\tau_t\}$.
7: **end for**

---

# 6 MAIN RESULTS

In this section, we provide and discuss the theoretical guarantees for PROM (Algorithm 1). We first present the regret analysis for the standard setting in Section 6.1, followed by an analysis of robustness to parameter misspecification in Section 6.2. The detailed proofs are deferred to Appendix F.

## 6.1 REGRET ANALYSIS FOR PROM (ALGORITHM 1)

**Theorem 6.1** (Standard regret and ICV regret for PROM). *In Algorithm 1, we choose $\alpha = C_1 d\sqrt{\log(6dT^2/p)}$ for $\Gamma^1$, and $\beta \geq C_2 d\log(3T(1 + \varepsilon T^2 + (1 + \varepsilon)T)/p)$ for $\Gamma^2$ and $\theta \in \mathcal{C}_\Theta^t(\beta)$, where $C_1, C_2 > 0$ are universal constants. For Scheme 1 (17) and Scheme 2 (18) in Algorithm 1, we have with probability at least $1 - p$ that $\mathrm{Reg}(T) = \widetilde{\mathcal{O}}(d^2\sqrt{T})$ and $\mathrm{Reg}^{\mathrm{ICV}}(T) = \widetilde{\mathcal{O}}(d^2\sqrt{T})$.*

Theorem 6.1 establishes that both Scheme 1 and Scheme 2 of Algorithm 1 achieve standard regret and ICV regret that are sublinear in the time horizon $T$ and polynomial in the feature dimension $d$. Notably, the regret rate matches the state-of-the-art bound for the unconstrained setting in Chen et al. (2023) where policy recommendation is absent, while we simultaneously ensure sublinear ICV regret. This indicates that Algorithm 1 efficiently learns the leader's optimal policy pair while effectively satisfying the IC constraint. In essence, the goal of ensuring incentive compatibility is achieved without compromising the statistical efficiency of learning the leader's optimal policy pair. Beyond Theorem 6.1, we further show that the absolute regret can also be bounded by a sublinear rate in $T$.

**Theorem 6.2.** *$SubOpt(\pi, y) = \mathrm{ICV}(\pi, y) = 0$ if and only if $(\pi, y)$ is the optimal solution to problem (9). Furthermore, suppose the follower's action space $\mathcal{B}$ is finite. If $\mathrm{Reg}(T) = \widetilde{\mathcal{O}}(\sqrt{T})$ and $\mathrm{Reg}^{\mathrm{ICV}}(T) = \widetilde{\mathcal{O}}(\sqrt{T})$, then the absolute regret satisfies $\mathrm{Reg}^{\mathrm{abs}}(T) = \widetilde{\mathcal{O}}(\sqrt{T})$.*

As discussed in Appendix 4.2, a fundamental challenge in our setting is that the IC constraint allows for negative suboptimality (i.e., achieving rewards higher than the equilibrium value by violating constraints), rendering standard regret an insufficient metric for convergence. Theorem 6.2 addresses this by establishing that absolute regret (which penalizes both suboptimal performance and constraint violations) is sublinear in $T$, provided that both standard regret and ICV regret are sublinear. This result not only rigorously quantifies the cost of enforcing IC but also ensures that the leader's policy converges to the true equilibrium, thereby guaranteeing incentive-compatible learning with provable efficiency. By bridging the gap between standard regret and absolute regret via the control of IC violations, Theorem 6.2 provides a novel theoretical framework for constrained online learning. To the best of our knowledge, this is the first result establishing such a connection, offering insights applicable to broader classes of constrained decision-making problems.

**Corollary 6.3** (Absolute regret for PROM). *Suppose the conditions of Theorem 6.1 are satisfied and the follower's action space $\mathcal{B}$ is finite. Then, for both Scheme 1 and Scheme 2 of Algorithm 1, we have the absolute regret $\mathrm{Reg}^{\mathrm{abs}}(T) = \widetilde{\mathcal{O}}(d^2\sqrt{T})$ with probability at least $1 - p$.*

Finally, combining Theorem 6.1 and Theorem 6.2 yields Theorem 6.3, which explicitly characterizes the absolute regret bound for Algorithm 1. Crucially, the transformation analysis in Theorem 6.2 preserves the dependence on the feature dimension $d$, and hence the dependency of the absolute regret bound on $d$ remains the same as the regret bound in Theorem 6.1. Consequently, Theorem 6.3 confirms that Algorithm 1 achieves an absolute regret that is sublinear in $T$ and scales as $\widetilde{\mathcal{O}}(d^2)$, thereby justifying the sample efficiency of Algorithm 1 in learning the quantal Stackelberg equilibrium.

## 6.2 ROBUSTNESS TO PARAMETER MISSPECIFICATION

In Section 6.1, we assume the leader knows the follower's behavioral parameters $\eta$ and $\tau$. While this assumption is standard in the Stackelberg learning literature (Yang et al., 2012; Chen et al., 2023), in practical scenarios, the leader may only have access to some estimates $\widehat{\eta}$ and $\widehat{\tau}$ which involve estimation errors. We consider a *misspecified* setting with $\zeta$-approximate behavioral parameters, where the leader only has access to estimates $\widehat{\eta}$ and $\widehat{\tau}$ satisfying $|\widehat{\eta} - \eta| \leq \zeta$ and $|\widehat{\tau} - \tau| \leq \zeta$. Then, a natural question arises: *How does the performance of Algorithm 1 degrade if these parameters are misspecified?* To address this, we analyze the regret of Algorithm 1 under such misspecification, when it is run using estimated parameters $\widehat{\eta}$ and $\widehat{\tau}$ that deviate from the true values by at most $\zeta$. We establish the following Theorem 6.4 to show that our framework exhibits graceful degradation.

**Theorem 6.4** (Regrets under misspecification). *Suppose Algorithm 1 is run using estimated parameters $\widehat{\eta}, \widehat{\tau}$ such that $|\widehat{\eta} - \eta|, |\widehat{\tau} - \tau| \leq \zeta$ for some $\zeta \geq 0$. Under the assumptions of Theorem 6.1, we have $\mathrm{Reg}(T) \leq \widetilde{\mathcal{O}}(d^2\sqrt{T} + \zeta T)$ and $\mathrm{Reg}^{\mathrm{ICV}}(T) \leq \widetilde{\mathcal{O}}(d^2\sqrt{T} + \zeta T)$ with probability at least $1 - p$.*

Theorem 6.4 shows that the regret bound incurs an additional linear term scaling with the misspecification level $\zeta$. This result implies convergence to an $\mathcal{O}(\zeta)$-optimal policy, and reflects the intrinsic bias of the parameter approximation. Building on Theorem 6.4, we show that this robustness extends to the more stringent *absolute regret* metric and implies a PAC sample complexity guarantee.

**Corollary 6.5** (Absolute regret under misspecification). *Under the conditions of Theorem 6.4, if the follower's action space is finite, the absolute regret satisfies $\mathrm{Reg}^{\mathrm{abs}}(T) \leq \widetilde{\mathcal{O}}(d^2\sqrt{T} + \zeta T)$.*

**Corollary 6.6** (PAC guarantee). *For any $\epsilon > 0$, Algorithm 1 learns a policy pair that is $(\epsilon + \mathcal{O}(\zeta))$-optimal with high probability, using a number of samples polynomial in the problem dimensions and $1/\epsilon^2$. The sample complexity to achieve an expected suboptimality gap of $\epsilon + C_{bias}\zeta$ is $\widetilde{\mathcal{O}}(d^4/\epsilon^2)$.*

**Intrinsic bias and graceful degradation.** Compared to Theorem 6.1, the bound in Theorem 6.4 has an additional linear term $\widetilde{\mathcal{O}}(\zeta T)$. The additional term is inevitably linear in $T$ due to the intrinsic bias introduced by the parameter approximation errors. This term implies that the average regret converges to $\mathcal{O}(\zeta)$ rather than zero. Therefore, Algorithm 1 is robust, as small estimation errors do not cause catastrophic failure but instead lead to a policy that is $\mathcal{O}(\zeta)$-optimal. As the estimation precision improves ($\zeta \to 0$), the standard no-regret guarantee in Theorem 6.1 is recovered.

**PAC interpretation.** The regret bound in Theorem 6.4 implies the sample complexity guarantee in Corollary 6.6 via standard online-to-batch conversion. With a number of samples polynomial in problem dimensions and $\epsilon^{-2}$, Algorithm 1 can learn a policy pair that is $\epsilon$-optimal up to an irreducible approximation error of $\mathcal{O}(\zeta)$. This highlights that while the $\mathcal{O}(\zeta)$ bias is unavoidable due to parameter mismatch, Algorithm 1 remains sample-efficient in reducing the statistical estimation error.

**Connection to literature.** The result in Theorem 6.4 parallels the robustness guarantees for misspecified linear MDPs. For instance, Jin et al. (2020) show that for $\zeta$-approximate linear MDPs, the regret bound naturally includes an additional linear term representing the intrinsic modeling bias. Notably, unlike the misspecification term in Jin et al. (2020) which scales with the feature dimension $d$, our linear bias term $\widetilde{\mathcal{O}}(\zeta T)$ is independent of $d$. This is because $\zeta$ here characterizes the estimation error of scalar behavioral parameters ($\eta, \tau$), and the sensitivity of the follower's response to these parameters is bounded by the magnitude of rewards rather than the ambient dimension of the feature space. Consequently, Algorithm 1 avoids the amplification of errors in high-dimensional feature spaces, effectively mitigating the curse of dimensionality related to parameter sensitivity. This property renders Algorithm 1 particularly robust in large-scale and high-dimensional problems, as the system bias remains controlled even as the feature dimension increases. Therefore, Algorithm 1 efficiently learns the best possible policy within the resolution allowed by the parameter precision.

## 7 CONCLUSION

We introduced online learning for leader-follower Stackelberg games with policy recommendation under an IC constraint. Our framework enables the leader to commit to a policy while recommending a policy for the follower, influencing decision-making while maintaining IC. We formulated this problem as a matrix game and proposed a sample-efficient algorithm that learns the optimal leader policy pair (quantal Stackelberg equilibrium) while minimizing both the leader's regret and IC violations. Our approach integrates MLE of the follower's response model with optimistic planning, achieving provable efficiency despite information asymmetry. We proved that our algorithm ensures sublinear regret in both the leader's rewards and IC violations (Theorem 6.1), showing that IC can be enforced without compromising statistical efficiency. Furthermore, we addressed the challenge posed by negative suboptimality values by bridging the gap between standard regret and absolute regret (Theorem 6.2), proving that sublinear regret and sublinear ICV regret together imply sublinear absolute regret. Even under parameter misspecification, our algorithm exhibits graceful degradation, as we show that small estimation errors in behavioral parameters lead to a bounded performance loss rather than catastrophic failure (Theorem 6.4). Our work provides a foundation for constrained online learning problems and opens avenues for future research, including extending to Markov games, analyzing computational trade-offs, and exploring applications in economic and strategic settings.

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

## A   NOTATIONS AND SUMMARY OF ASSUMPTIONS

In this section, we summarize the notations used throughout the paper.

We use the following notations that are generally used in the literature. Let $\mathbb{S}^d_+$ denote the set of $d$-dimensional symmetric nonnegative definite matrices, and $\mathcal{F}(\mathcal{X})$ be the class of measurable functions on $\mathcal{X}$. We define $[N] = \{1, \ldots, N\}$, and for any set $X$, let $|X|$ denote its cardinality. The probability simplex over $\mathcal{X}$ is $\Delta(\mathcal{X})$. For a random vector $x$, we use $\mathrm{Cov}[x]$ for its covariance matrix. The vector norm induced by a matrix $P \in \mathbb{S}^d_+$ is denoted as $\|\cdot\|_P$, and $\lesssim$ denotes an inequality up to constant and logarithmic factors.

In this paper, we use the total variation (TV) distance to measure the difference between two policies. Consider a measurable space $(\Omega, \mathcal{F})$ and probability measures $P$ and $Q$ defined on $(\Omega, \mathcal{F})$. The TV distance between $P$ and $Q$ is defined as

$$\mathrm{TV}(P, Q) := \sup_{A \in \mathcal{F}} |P(A) - Q(A)|, \tag{20}$$

which is the largest absolute difference between the two probabilities assigned to the same event. For discrete spaces, we have $\mathrm{TV}(P, Q) = \frac{1}{2}\|P - Q\|_1$ by Proposition 4.2 in Levin & Peres (2017), where $\|\cdot\|_1$ is the $L_1$-norm of vectors.

When appropriate without causing ambiguity, we omit the subscript $\mathcal{A} \times \mathcal{B}$ in $\langle \cdot, \cdot \rangle_{\mathcal{A} \times \mathcal{B}}$.

We also provide the following table of notations (Table 1) related to our setting used throughout the paper, as well as their meanings below.

To improve readability, we also summarize the key assumptions made in this paper in Table 2 below.

## B   EXTENDED DISCUSSION ON RELATED WORK

In this section, we provide a more detailed discussion of our work's relationship to the literature on mechanism design, information design, and recent advances in steering learning agents. We highlight the unique challenges posed by our setting (e.g., the combination of policy recommendation, incentive compatibility, information asymmetry and bounded rationality) which distinguishes our contribution from classical and concurrent works.

### B.1   CONNECTION TO MECHANISM DESIGN AND INFORMATION DESIGN

As noted in Section 1, our Incentive Compatibility (IC) constraint is conceptually equivalent to the **obedience constraint** in the mechanism design and information design literature (Myerson, 1983; Bergemann & Morris, 2019). In the context of Bayesian persuasion (Kamenica & Gentzkow, 2011), the sender (leader) designs a signal (recommendation) to influence the receiver's (follower's) action. The revelation principle guarantees that we can restrict attention to direct recommendations where it is optimal for the receiver to obey.

However, there is a fundamental distinction between these classical settings and our work:

- **Static Design vs. Online Learning:** The classical literature typically assumes a static environment where the designer has full knowledge of the agent's utility function (or the prior distribution). The goal is to compute an optimal mechanism/signal offline.

- **Unknown Rewards:** In our setting, the leader *does not know* the follower's reward function (parameter $\theta^*$) and must learn it from scratch through online interactions. This introduces a critical challenge: the obedience constraint itself is unknown and must be estimated.

- **The Necessity of Absolute Regret:** Because the leader optimizes over an *estimated* constraint, the actual IC condition may be violated during the learning process. In constrained optimization, violating a constraint can sometimes yield rewards higher than the equilibrium value (negative suboptimality). This phenomenon does not arise in static mechanism design (where constraints are hard) but is central to our online learning setting. This necessitates our novel analysis of **absolute regret** (Theorem 6.2) to ensure convergence to the true equilibrium.

Table 1: Table of notations

| Notation | Meaning |
|---|---|
| $\mathcal{A}, \mathcal{B}$ | $\mathcal{A}$ and $\mathcal{B}$ are the leader's and the follower's action spaces, respectively, which are both assumed to be finite. |
| $\Pi, \mathcal{Y}, \mathcal{Y}_{\text{QR}}$ | $\Pi$ is the class of leader's policies, which is defined as $\Delta(\mathcal{A})$, the set of all distributions over $\mathcal{A}$. $\mathcal{Y}$ is the class of leader's recommendation policies to the follower, which is defined as $\text{int}(\Delta(\mathcal{B}))$, the interior of the set of all distributions over $\mathcal{B}$. $\mathcal{Y}_{\text{QR}}$ is the class of all quantal responses as defined in equation 25. |
| $u, r$ | $u$ and $r$ are the leader's and the follower's reward functions, respectively, which are both assumed to be within $[0, 1]$. |
| $\vartheta, \vartheta^*, \theta, \theta^*, \Theta$ | $\vartheta$ is the parameter of the leader's reward function. $\vartheta^*$ is the true value of the parameter $\vartheta$. $\theta$ is the parameter of the follower's reward function, which determines the follower's response mapping $(\pi, y) \to \nu^{\pi,y}$. $\theta^*$ is the true value of the parameter $\theta$. $\Theta = [0, 1]^{|\mathcal{A}||\mathcal{B}|}$ is the class of the parameters for the leader's and the follower's reward functions. |
| $\pi, y$ | $\pi$ is the leader's commitment policy and $y$ is the leader's recommendation policy to the follower. |
| $\mu_\theta^\pi$ | $\mu_\theta^\pi$ is the quantal response model against the leader's policy $\pi$ when the parameter of the follower's reward function is $\theta$. See equation 2 and equation 6. |
| $\nu_\theta^{\pi,y}$ | $\nu_\theta^{\pi,y}$ is the follower's response model against the leader's policy $\pi$ and recommendation policy $y$ when the parameter of the follower's reward function is $\theta$. See equation 3 and equation 5. |
| $\mathcal{H}(\nu)$ | The Shannon entropy of a distribution $\nu$. |
| $D(\nu\|y)$ | The Kullback–Leibler (KL) divergence of a distribution $\nu$ with respect to another distribution $y$. |
| $J(\pi, y)$ | $J(\pi, y)$ is the total expected utility collected by the leader if the follower's response model is $\nu_{\theta*}^{\pi,y}$. |
| $U_\vartheta, W_{\vartheta,\theta}^{\pi,y}$ | (U, W) functions for the leader under model $(\pi, y, \nu^{\pi,y}, \vartheta, \theta)$. See equation 7. |
| $Q_\theta^\pi, V_\theta^{\pi,y}, A_\theta^{\pi,y}$ | (Q, V, A)-functions for the follower under model $(\pi, y, \nu^{\pi,y}, \vartheta, \theta)$. See equation 5 and equation 4. |
| $V_\theta^\pi, A_\theta^\pi$ | (V, A)-functions for the follower under model $(\pi, y, \nu^{\pi,y}, \vartheta, \theta)$ when $\tau = 0$. |
| $r^\pi(b)$ | $r^\pi = \langle r(\cdot, b), \pi(\cdot \mid b) \rangle_{\mathcal{A}}$. |
| $\pi^*, y^*$ | $\pi^*$ is the leader's best policy, and $y^*$ is the leader's best recommendation policy to the follower. |
| $B_A$ | $B_A$ upper bounds the follower's value functions. See equation 204. |

Table 2: Summary of key assumptions

| Assumption | Details |
|---|---|
| Game Model | Leader-Follower Matrix Game (Stackelberg Game). The leader commits to a policy $\pi$ and recommends a policy $y$; the follower responds with a policy $\nu$. |
| Finite Action Spaces | The action spaces $\mathcal{A}$ for the leader and $\mathcal{B}$ for the follower are both finite. |
| Linear Reward Structure | The reward functions are linear in a known feature map $\phi(a,b)$: $u(a,b) = \langle \phi(a,b), \vartheta^* \rangle$ and $r(a,b) = \langle \phi(a,b), \theta^* \rangle$. |
| Parameter Space | The unknown reward parameters $\vartheta^*, \theta^*$ lie in the bounded set $[0,1]^d$ (where $d = |\mathcal{A}||\mathcal{B}|$ in the tabular setting). |
| Follower Behavior | The follower is boundedly rational and adopts a One-step Mirror Descent (OMD) update based on the leader's recommendation $y$, resulting in the response model $\nu_\theta^{\pi,y}$. |
| Known Hyperparameters | The rationality parameter $\eta$ and recommendation sensitivity $\tau$ are assumed to be known to the leader in the main analysis. (Robustness to misspecified $\eta, \tau$ is discussed in Section 6.2). |
| Observability | Bandit feedback: The leader observes the joint action $(a,b)$ and her own reward $u(a,b)$, but **not** the follower's reward $r(a,b)$. The follower observes the leader's commitment $\pi$ and recommendation $y$. |

## B.2 Comparison with Steering in Equilibrium

Recent work by Zhang et al. (2024) studies the problem of steering no-regret learners to a desired equilibrium. While both works share the goal of influencing agent behavior, they address fundamentally different problems with distinct mechanisms and assumptions.

1. **Mechanism (Payments vs. Information):** Zhang et al. (2024) rely on monetary **payments** to modify the players' payoff matrices directly, effectively making the desired equilibrium dominant or more attractive. In contrast, our leader cannot issue payments; she relies solely on **policy recommendation (information)** and the follower's bounded rationality (quantal response) to achieve alignment via the obedience constraint.

2. **Information Structure (Known vs. Unknown):** The framework in Zhang et al. (2024) typically assumes the mediator knows the game structure (including players' utilities) to compute the necessary payments. Our primary challenge is **information asymmetry**: the leader does not know the follower's reward function and must infer it from bandit feedback. Algorithms designed for known games cannot solve our learning problem.

3. **Agent Model (Dynamics vs. Static Response):** Zhang et al. (2024) assume agents run **no-regret algorithms** (e.g., MWU, CFR), focusing on the convergence of learning dynamics over time. We model the follower as a **boundedly rational agent (quantal response)** who responds instance-by-instance based on the current policy and recommendation. This necessitates fundamentally different analysis techniques, such as handling the distribution shift between the estimated and true quantal responses (as detailed in Appendix D.4), which is not a concern when analyzing no-regret dynamics in fixed games.

## B.3 Comparison with Other Stackelberg Learning Works

Our work also complements other recent advances in learning Stackelberg equilibria.

- **Comparison with Chen et al. (2023):** Our work builds on the bilevel optimization framework of Chen et al. (2023). However, the introduction of policy recommendation (PR) fundamentally changes the problem. In their setting, the follower's response depends only on $\pi$; in ours, it depends on $(\pi, y)$. This dependency creates a **distribution shift** between the follower's actual response (influenced by $y$) and the ideal quantal response (which the leader aims to learn). We develop a novel invariance-based analysis to bridge this gap, a technical innovation absent in Chen et al. (2023).

- **Comparison with Haghtalab et al. (2023):** Haghtalab et al. (2023) study a setting where the follower responds to calibrated forecasts of the leader's actions. In their model, the follower does not observe the leader's current commitment directly. In our model, the follower observes both the commitment $\pi$ and the recommendation $y$. Our focus is on optimizing the recommendation to satisfy the obedience constraint, rather than calibrating forecasts over time.

## C DISCUSSIONS AND PROPERTIES OF LEADER-FOLLOWER GAMES WITH POLICY RECOMMENDATION

### C.1 MOTIVATING EXAMPLES OF POLICY RECOMMENDATION

The introduction of policy recommendation is meaningful in realistic scenarios where the follower (e.g., a small company) only has access to limited computational resources, while the leader (e.g., a big company) has access to sufficient computational resources. Due to this nature, it is also reasonable to assume the action space of the follower is finite. In such scenarios, the follower can only employ computationally efficient methods, such as one-step mirror descent. Additionally, because the leader possesses ample computational resources, she can use $(\pi, y)$ to guide or influence the follower's behavior. Consequently, the follower also takes the leader's recommendation policy $y$ into consideration. As a result, the follower's response is a function of both the leader's commitment policy and the leader's recommendation policy to the follower.

In real-world scenarios, the phenomena of policy recommendation in leader-follower games are common in several domains, including but not limited to the following examples.

1. **Supply chain management:** In supply chains, dominant manufacturers (leader) often suggest inventory policies to retailers (follower) to optimize overall performance. For instance, a manufacturer may recommend specific ordering quantities and schedules to a retailer to minimize costs and prevent stockouts. The retailer, possibly constrained by limited resources, relies on these recommendations to manage inventory efficiently. This dynamic aligns with the leader-follower model, where the leader (manufacturer) provides a policy that the follower (retailer) adopts to achieve mutual benefits.

2. **Security resource allocation:** In security contexts, authorities (leader) allocate resources and provide guidelines to security personnel (follower) on optimal patrolling strategies. For example, law enforcement agencies may develop patrol routes and schedules to maximize coverage and deterrence, recommending these strategies to officers on duty. The officers implement these recommended policies, enhancing overall security effectiveness. This scenario reflects a leader-follower relationship, with the central authority guiding the actions of individual personnel.

3. **Energy distribution networks:** In energy markets, central operators (leader) propose consumption plans to consumers (follower) to balance demand and supply. For instance, utility companies may suggest usage patterns or demand response strategies to consumers during peak hours to prevent grid overloads. Consumers adjust their energy consumption based on these recommendations, contributing to grid stability. This interaction exemplifies the leader-follower model, where the operator leads by recommending policies that consumers follow.

These examples illustrate the practical significance of policy recommendation mechanisms in leader-follower dynamics, underscoring the relevance of research in multi-agent decision-making.

### C.2 SUBOPTIMALITY OF RECOMMENDATION POLICY

We can also define the **suboptimality of recommendation** as
$$\text{SubOpt}_R(\pi, y; \theta) := \max_{\nu \in \Delta(\mathcal{B})} G(\nu; \pi, y; \theta) - G(y; \pi, y; \theta)$$
$$= G(\nu_\theta^{\pi, y}; \pi, y; \theta) - G(y; \pi, y; \theta), \tag{21}$$
where the second equality is by the follower's response model (3). By definition, since $y \in \mathcal{Y} \subset \Delta(\mathcal{B})$, we have that $\text{SubOpt}_R$ is always nonnegative. The quantity measures the difference of the follower's

expected regularized utility between taking her best response policy $\nu_\theta^{\pi,y}$ under policy pair $(\pi, y)$ and the recommended policy $y$ from the leader. In other words, it measures the maximum expected regularized reward that the follower could obtain by deviating from the leader's recommended policy $y$. If $\text{SubOpt}_R(\pi, y; \theta)$ is very small, then for the follower there is not much difference between taking $\nu_\theta^{\pi,y}$ and taking $y$. Specially, if $\text{SubOpt}_R(\pi, y; \theta) = 0$, the follower's utility is the same between taking $\nu_\theta^{\pi,y}$ or taking $y$, and the follower will not benefit from deviating from the recommended policy $y$ from the leader.

Theorem C.2 shows that $\text{ICV}(\pi, y; \theta) = 0$ if and only if $\text{SubOpt}_R(\pi, y; \theta) = 0$. Suppose $\text{SubOpt}_R(\pi, y; \theta) = 0$, then by definition, $y$ and $\nu_\theta^{\pi,y}$ simultaneously maximize $G(\cdot; \pi, y; \theta)$. By the analysis in Section 3.2, the maximizer of $G(\cdot; \pi, y; \theta)$ is unique by strict concavity, and hence we can immediately have that $\nu_\theta^{\pi,y} = y$. It is also evident that $\nu_\theta^{\pi,y} = y$ directly implies $\text{SubOpt}_R(\pi, y; \theta) = 0$. Therefore, $\text{SubOpt}_R(\pi, y; \theta) = 0$ is equivalent to $\nu_\theta^{\pi,y} = y$. Last, by the definition of $\text{ICV}(\pi, y; \theta)$ and the property of TV distance, we immediately have that $\text{ICV}(\pi, y; \theta) = 0$ if and only if $\text{SubOpt}_R(\pi, y; \theta)$. Theorem C.2 implies that, the IC condition implies that the follower will not benefit from deviating from the recommended policy. Given the constraint that the leader's policy pair $(\pi, y)$ should be IC, the value of ICV measures to what extent the IC condition is violated. So we can define the regret of ICV as in equation 12.

Given any fixed $\pi$, we can regard $\nu_\theta^{\pi,y}$ as a function of $y$, and if $\text{ICV}(\pi, y; \theta) = 0$, then $y$ is actually a fixed point of the self mapping $\nu_\theta^{\pi,\cdot} : \mathcal{Y} \to \mathcal{Y}$. With this observation, Theorem C.1 shows that the following propositions are equivalent: $\text{ICV}(\pi, y; \theta) = 0$ iff $y = \nu_\theta^{\pi,y}$ iff $y = \mu_\theta^\pi$. This provides guidance to the design of Scheme 1 (17) and Algorithm 1.

### C.3 RELATION TO PROXIMAL POINT ALGORITHM

Recall the theory of proximal point algorithm (PPA) (Rockafellar, 1976; Güler, 1991) with Bregman Divergence. Let $F : \mathcal{X} \to \mathbb{R}$ be a function satisfying certain conditions. Consider the problem of minimizing $F(x)$ over $x \in \mathcal{X}$. For any $x_0 \in \mathcal{X}$, $\alpha > 0$ and $t \in \mathbb{N}$, construct the sequence

$$x_{t+1} \leftarrow \underset{x \in \mathcal{X}}{\arg\min} \left\{ F(x) + \alpha^{-1} D(x || x_t) \right\}. \tag{22}$$

Then we have a fixed point (of some mapping) $x^* \in \arg\min_{x \in \mathcal{X}} F(x)$.

For any given policy $\pi \in \Pi$ and parameter $\theta \in \Theta$, let $T_\theta^\pi : \text{int}(\Delta(\mathcal{B})) \to \text{int}(\Delta(\mathcal{B}))$ be a mapping such that

$$T_\theta^\pi(y) = \underset{\nu \in \text{int}(\Delta(\mathcal{B}))}{\arg\max} G(\nu | \pi, y; \theta), \tag{23}$$

where $G(\nu | \pi, y; \theta)$ is defined in equation 3. $T_\theta^\pi$ maps the leader's recommendation $y$ to the response policy of the follower against the leader's policy $\pi$, and hence we also denote $\text{BR}(\pi, y; \theta) := T_\theta^\pi(y)$ as the best response of the follower against $\pi$ given parameter $\theta$. Let $y_\theta^{*,\pi} \in \text{int}(\Delta(\mathcal{B}))$ be a fixed point of $T_\theta^\pi$, which satisfies $y_\theta^{*,\pi} = T_\theta^\pi(y_\theta^{*,\pi})$. Then, for any $y_0 \in \text{int}(\Delta(\mathcal{B}))$, we construct a sequence such that $y_{t+1} = T_\theta^\pi(y_t)$ for all $t \in \mathbb{N}$.

According to the theory of PPA with Bregman Divergence, the sequence $\{y_t\}_{t \in \mathbb{N}}$ convergences to the (unique) maximizer of $g(\cdot | \pi; \theta)$ defined in equation 2, denoted as $y_{\theta,\infty}^\pi$, which is the quantal response $\mu_\theta^\pi$ against $\pi$ given parameter $\theta$. We have $y_{\theta,\infty}^\pi = T_\theta^\pi(y_{\theta,\infty}^\pi)$ and hence $y_{\theta,\infty}^\pi$ is a fixed point of $T_\theta^\pi$. By the uniqueness shown later, we have that $y_\theta^{*,\pi}$, the fixed point of $T_\theta^\pi$, is exactly the quantal response $\mu_\theta^\pi$ against the leader's policy $\pi$ given parameter $\theta$. In other words, the quantal response $\mu_\theta^\pi$ is the fixed point of $T_\theta^\pi$, and if the leader takes policy $\pi$ and recommend $\mu_{\theta*}^\pi$ to the follower, then the follower's response policy will be equal to the leader's recommendation. See Theorem C.1 for the formal statement and proof. By this important observation, one of the leader's goals is to make the follower follow her recommendation so that the game can converge to some equilibrium. In order to achieve this goal, we hope the leader's policy pair is within the class $\{(\pi, \mu_{\theta*}^\pi) : \pi \in \Pi\} \subset \Pi \times \mathcal{Y}$.

### C.4 THEORETICAL PROPERTIES OF LEADER-FOLLOWER GAMES WITH POLICY RECOMMENDATION

In this subsection, we analyze and discuss the properties of leader-follower games with policy recommendation in more detail. The proofs of the results in this subsection are provided in Appendix G.

In the sequel, we write $Q_\theta^\pi(b) = r_\theta^\pi(b) = \langle \phi^\pi(b), \theta \rangle$ for the $\mathbb{R}^d$ canonical kernel $\phi = e : \mathcal{A} \times \mathcal{B} \to \mathbb{R}^d$ with $\phi^\pi(b) = \langle \phi(\cdot, b), \pi(\cdot) \rangle_\mathcal{A}$ and parameter $\theta \in \Theta = [0, 1]^d$ with $d = |\mathcal{A}||\mathcal{B}|$. The follower's response policy is given by $\nu_\theta^{\pi,y}(b) \propto y(b)^{\tau\varepsilon} \cdot \exp(\varepsilon \cdot \langle \phi^\pi(b), \theta a \rangle)$, where $\varepsilon := (\tau + \eta^{-1})^{-1}$.

The following Theorem C.1 provides guidance to the design of Scheme 1 (17) and Algorithm 1.

**Proposition C.1** (Quantal response as a fixed point). *Given any $\theta \in \Theta$ and $\pi \in \Pi$, the quantal response $\mu_\theta^\pi$ is the unique fixed point of the mapping $\nu_\theta^{\pi,\cdot} : \mathrm{int}(\Delta(\mathcal{B})) \to \mathrm{int}(\Delta(\mathcal{B}))$. In other words, $y = \mu_\theta^\pi$ if and only if $y = \nu_\theta^{\pi,y}$. Furthermore, we have $V_\theta^\pi = V_\theta^{\pi, \mu_\theta^\pi}$.*

*Proof.* See Appendix G.1. □

Due to the introduction of a policy recommendation $y$, $V_\theta^\pi = V_\theta^{\pi,y}$ does not hold in general. However, Theorem C.1 also shows that $V_\theta^\pi = V_\theta^{\pi,y}$ holds when the recommendation policy $y$ is the quantal response against $\pi$ given $\theta$, i.e., $y = \mu_\theta^\pi$. Equivalently, we can write $V_\theta^\pi = V_\theta^{\pi, \mu_\theta^\pi}$. This implies that the introduction of a policy recommendation that is equal to the quantal response does not change the value function.

The following Theorem C.2 shows that ICV and the suboptimality of recommendation reach zero simultaneously.

**Proposition C.2.** $\mathrm{ICV}(\pi, y; \theta) = 0$ *if and only if* $SubOpt_\mathrm{R}(\pi, y; \theta) = 0$.

*Proof.* See Appendix G.2. □

The following two lemmas show that, if the follower's action space $\mathcal{B}$ is finite, then the quantal response is in the interior of $\Delta(\mathcal{B})$, and further if the leader's recommendation policy is some quantal response, then the follower's response is also in the interior of $\Delta(\mathcal{B})$. These results will be used in the proof of Theorem 6.2.

**Lemma C.3** (Positive lower bound of the quantal response). *Suppose the follower's action space $\mathcal{B}$ is finite, i.e., $|\mathcal{B}| < \infty$. Then, for any $b \in \mathcal{B}$, $\pi \in \Pi$ and $\theta \in \Theta$, we have*

$$\mu_\theta^\pi(b) \geq \mu_\mathrm{L} := |\mathcal{B}|^{-1} \cdot \exp(-\eta) > 0. \tag{24}$$

*In other words, the quantal response against any policy $\pi \in \Pi$ given any parameter $\theta \in \Theta$, is uniformly lower bounded by the positive constant $\mu_\mathrm{L}$.*

*Proof.* See Appendix G.3. □

Before stating Theorem C.4, we define $\mathcal{Y}_\mathrm{QR}$ as the class of all quantal responses, i.e.,

$$\mathcal{Y}_\mathrm{QR} := \{y \in \Delta(\mathcal{B}) : y = \mu_\theta^\pi, \pi \in \Pi, \theta \in \Theta\}. \tag{25}$$

In other words, $y \in \mathcal{Y}_\mathrm{QR}$ means that there exist some $\pi \in \Pi$ and some $\theta \in \Theta$ such that $y = \mu_\theta^\pi$.

**Lemma C.4** (Positive lower bound of the follower's response). *Suppose the follower's action space $\mathcal{B}$ is finite, i.e., $|\mathcal{B}| < \infty$. Then, for any $b \in \mathcal{B}$, $\pi \in \Pi$, $y \in \mathcal{Y}_\mathrm{QR}$ and $\theta \in \Theta$, we have*

$$\nu_\theta^{\pi,y}(b) \geq \nu_\mathrm{L} := |\mathcal{B}|^{-\frac{2\tau\eta+1}{\tau\eta+1}} \cdot \exp(-\eta) > 0. \tag{26}$$

*In other words, the follower's response against any policy $\pi \in \Pi$ and any quantal response recommendation $y \in \mathcal{Y}_\mathrm{QR}$ given any parameter $\theta \in \Theta$, is uniformly lower bounded by the positive constant $\nu_\mathrm{L}$.*

*Proof.* See Appendix G.4. □

The following Theorem C.5 can be regarded as some extension to Theorem C.1, and it will be used to prove Theorem 6.2.

**Lemma C.5.** *Suppose the follower's action space $\mathcal{B}$ is finite, i.e., $|\mathcal{B}| < \infty$. Suppose $\sigma \in [0, \frac{1}{2}\nu_\mathrm{L}]$ where $\nu_\mathrm{L}$ is defined in equation 26. For any $\pi \in \Pi$, $y \in \mathcal{Y}_\mathrm{QR}$ and $\theta \in \Theta$, if $\|y - \nu_\theta^{\pi,y}\|_1 \leq \sigma$, then*

$$\|y - \mu_\theta^\pi\|_1 \leq \frac{4 \cdot 3^{\tau\eta}(1 + \tau\eta)}{\nu_\mathrm{L}} \cdot \sigma. \tag{27}$$

*Proof.* See Appendix G.5. □

We then use Theorem C.5 to analyze the relaxation of the ICV constraint. For any $\theta \in \Theta$, define a function $F_\theta$ for $\sigma \geq 0$ as

$$F_\theta(\sigma) := \max_{\mathrm{TV}(y,\nu_\theta^{\pi,y}) \leq \sigma} \pi^\top A \nu_\theta^{\pi,y}. \tag{28}$$

Here, the variable $\sigma$ measures the degree of relaxation of the ICV constraint. The $\theta$-ICV condition corresponds to $\sigma = 0$, and the condition is relaxed as $\sigma$ increases.

We then turn to analyze the property of the function $F_\theta$. Thanks to the entropy regularization in both the quantal response model equation 2 and the follower's response model equation 3, $\pi^\top A \nu_\theta^{\pi,y}$ and $\mathrm{TV}(y,\nu_\theta^{\pi,y})$ are both continuous functions of $(\pi, y)$, and consequently the function $F_\theta(\sigma)$ is continuous in $\sigma \in [0, +\infty)$. Also, it is evident that $F_\theta(\sigma)$ is a non-decreasing function in $\sigma$. Hence, we have that $\sigma \downarrow 0$ implies $F_\theta(\sigma) \downarrow F_\theta(0)$, i.e.,

$$F_\theta(\sigma) - F_\theta(0) = \max_{\mathrm{TV}(y,\nu_\theta^{\pi,y}) \leq \sigma} \pi^\top A \nu_\theta^{\pi,y} - \max_{\mathrm{TV}(y,\nu_\theta^{\pi,y})=0} \pi^\top A \nu_\theta^{\pi,y} \downarrow 0. \tag{29}$$

However, equation 29 is not enough. In order to prove Theorem 6.2, we need a more delicate result that characterizes the decaying rate in equation 29. Fortunately, we can use Theorem C.5 to prove a stronger result in the following Theorem C.6 that $F_\theta(\sigma) - F_\theta(0) = \mathcal{O}(\sigma)$ for sufficiently small $\sigma$, which characterizes the decaying rate in equation 29.

**Proposition C.6** (Relaxation of ICV constraint). *Suppose the follower's action space $\mathcal{B}$ is finite, i.e., $|\mathcal{B}| < \infty$. For any $\theta \in \Theta$ and any $\sigma \in [0, \frac{1}{4}\nu_{\mathrm{L}}]$ where $\nu_{\mathrm{L}}$ is defined in equation 26, suppose the policy pair $(\pi, y)$ satisfies $\mathrm{TV}(y,\nu_\theta^{\pi,y}) \leq \sigma$. Then, for the function $F_\theta$ defined in Equation (28), we have*

$$F_\theta(\sigma) - F_\theta(0) \leq \left( 2 + \frac{8 \cdot 3^{\tau\eta}(1 + \tau\eta)}{\nu_{\mathrm{L}}} \right) \cdot \sigma =: C_F \cdot \sigma, \tag{30}$$

*where we let $C_F := 2 + \frac{8 \cdot 3^{\tau\eta}(1+\tau\eta)}{\nu_{\mathrm{L}}}$.*

*Proof.* See Appendix G.6. □

As a result of Theorem C.6, we have shown that $F_\theta(\sigma) - F_\theta(0) = \mathcal{O}(\sigma)$ for $\sigma \in [0, \frac{1}{4}\nu_{\mathrm{L}}]$. We will then use this result to prove Theorem 6.2, as given in Appendix F.4.

# D    TECHNICAL CHALLENGES AND CONTRIBUTIONS

Our work addresses a number of technical challenges that arise in leader-follower games (LFG) with policy recommendation (PR), which are not adequately handled by prior work. We summarize the key challenges and our corresponding contributions below.

## D.1    DEVELOPING A THEORETICAL FRAMEWORK FOR LFG WITH PR

Prior works lack a formal theoretical foundation for LFG with PR. In contrast, we establish a suite of key structural properties that underpin the design and analysis of such games (see Appendix C.4). These results address fundamental questions about feasibility, tractability, and incentive alignment in LFG with PR, and are essential to the formulation of our learning algorithm and regret guarantees.

First, Theorem C.1 shows that the recommendation policy can be restricted to quantal responses against the leader's commitment, conditioned on the follower's estimated parameter. This significantly reduces the dimensionality of the recommendation space, enabling practical optimization and clarifying the leader's design strategy. Second, Theorem C.2 formally justifies our ICV formulation, proving that it aligns with the solution concept in LFG with PR and ensures that the follower has no incentive to deviate. Third, we analyze the impact of relaxing the IC constraint: Theorem C.6 quantifies how such relaxation affects the leader's utility, providing key continuity bounds that are used in the regret analysis. Together with supporting results (e.g., Theorems C.3 to C.6 in Appendix C.4), these findings reveal the necessity of a finite follower action space and describe how perturbations in IC violations decay (e.g., Equation (29)).

These structural results go beyond previous models that omit recommendation or assume full rationality, and they are derived using nontrivial techniques including fixed-point characterizations, variational inequalities, and Lipschitz continuity arguments. They form the theoretical foundation of our contributions and are essential to understanding the dynamics and optimization of policy recommendation in LFG.

### D.2 DESIGN OF EFFECTIVE RECOMMENDATION POLICY

While existing methods assume that the follower best responds to a committed policy, the introduction of PR fundamentally changes the leader's challenge. The leader must now design a recommendation distribution that both influences the follower's behavior and satisfies IC. Without further structural insight, this recommendation space encompasses all probability distributions over policies—rendering it vast, unstructured, and computationally intractable.

Our analysis provides a principled resolution to this challenge. In Theorem C.1, we show that the optimal recommendation corresponds to the quantal response to the leader's commitment, evaluated under the follower's estimated reward parameter $\theta$. This key result yields a substantial reduction in complexity by restricting the recommendation policy space to a low-dimensional manifold defined by quantal response equilibria. Moreover, this structure offers clear interpretability: the recommendation becomes a softened best response, encoding both rationality and uncertainty.

This insight allows the leader to bypass the otherwise intractable optimization over general distributions. It also informs the design of efficient algorithms that maintain IC by construction, facilitating convergence and regret analysis. Our structured approach thus bridges the gap between theory and practice in recommendation-based LFG, offering a scalable and principled method that advances beyond previous models.

### D.3 REGRET ANALYSIS WITH IC CONSTRAINT

In conventional online learning, regret is typically defined relative to a fixed benchmark, and suboptimality is nonnegative by definition. However, the presence of IC constraints in our setting fundamentally alters this landscape: the leader may be forced to deviate from the utility-maximizing policy in order to maintain IC, potentially incurring negative suboptimality under the standard definition. As a result, conventional regret notions are no longer suitable for this setting.

To address this, we introduce and analyze the notion of absolute regret, which measures the magnitude of deviation from the benchmark regardless of sign. Our main theoretical results, Theorem 6.1 and Theorem 6.2, establish a novel regret framework tailored to incentive-constrained learning. Specifically, we prove that sublinear ICV regret implies sublinear absolute regret, connecting the two through a careful analysis of the structure of LFGs with PR. This connection relies heavily on the structural results established in Appendix C.4, particularly the bounds and continuity results related to constraint relaxation (Theorem C.6).

One central challenge is to quantify how much performance is lost when the IC constraint is only approximately satisfied. We address this through Theorems C.5 and C.6 in Appendix C.4, which rigorously characterize how small violations in IC translate to bounded losses in the leader's utility. In addition, the introduction of recommendation policies further complicates the follower's response, necessitating detailed mathematical treatment (see Appendices C.4 and G). These technical developments collectively enable the proof of Theorem 6.2, which provides the first rigorous characterization of regret under IC constraints in online learning.

Our framework establishes a new foundation for online learning with structural constraints and is broadly applicable to constrained decision-making problems beyond LFG, laying groundwork for future theoretical and algorithmic advances.

### D.4 DISTRIBUTION SHIFT CAUSED BY POLICY RECOMMENDATION

Our work introduces PR, which creates a discrepancy between the follower's response (FR) and the quantal response (QR), resulting in a distribution shift not addressed in prior work, including Chen et al. (2023). This discrepancy presents a key technical challenge: controlling the total variation (TV) distance between two QRs under different reward functions (see Appendix H.2, Equation (126)).

While the method developed in Chen et al. (2023) can effectively bound the distance between two FRs, it cannot directly bound the distance between two QRs. Since the follower's observed behavior follows the FR model, Equation (142) naturally yields a bound on the FR distance. However, applying MLE to QRs introduces an additional term, $(t-1)\log C$, which leads to a linear regret bound—an outcome we aim to avoid. Moreover, attempts to relate the FR and QR distances through a constant factor $C$ are inherently flawed, as the distribution $\nu_\theta^{\pi,y}$ is not identifiable with respect to $\theta$: different values of $\theta$ can induce different QRs while yielding the same FR, rendering such a bound impossible.

To resolve this issue, we leverage the invariance property of FRs under constant shifts in the reward function (Theorem J.4), and establish Equation (172). We then prove key technical results in Theorem J.1 and Theorem J.3, which together enable us to bound the QR distance and derive Equation (126). This approach relies on novel and nontrivial mathematical techniques, constituting a significant contribution beyond the scope of Chen et al. (2023).

### D.5 Robustness to Parameter Misspecification

In our primary analysis, we assume the leader knows the follower's behavioral parameters $\eta$ and $\tau$. However, in practice, these parameters must often be estimated from data, leading to inevitable estimation errors. The core technical challenge lies in quantifying how these parameter errors propagate through the follower's nonlinear response model (Eq. 5) and impact the leader's learning process. The follower's response $\nu_\theta^{\pi,y}$ depends on $\eta$ and $\tau$ through complex exponential and normalization terms. A small perturbation in these parameters could potentially lead to significant shifts in the response distribution, thereby invalidating the leader's estimated reward function and ICV measurements. Standard regret analysis techniques (e.g., for linear bandits) typically assume a well-specified model and do not directly account for such systematic model mismatch. Therefore, establishing a rigorous bound on the performance degradation caused by this misspecification requires a delicate sensitivity analysis of the quantal response equilibrium.

We address this challenge by establishing the **graceful degradation** property of our algorithm. Specifically:

1. **Lipschitz Continuity Analysis:** We rigorously prove that the follower's response model is Lipschitz continuous with respect to the behavioral parameters $\eta$ and $\tau$. This effectively bounds the distribution shift in the follower's behavior by the parameter estimation error $\zeta$.

2. **Regret Decomposition with Bias:** We derive a novel regret bound (Theorem 6.4) that explicitly decomposes the total regret into a standard sublinear learning term $\widetilde{\mathcal{O}}(d^2\sqrt{T})$ and a linear systematic bias term $\widetilde{\mathcal{O}}(\zeta T)$. This theoretically confirms that the algorithm does not fail catastrophically under misspecification but converges to a policy that is optimal up to the resolution of the parameter estimates.

3. **Dimension-Independent Bias:** A key insight from our analysis is that the bias term $\widetilde{\mathcal{O}}(\zeta T)$ is **independent of the feature dimension** $d$. This contrasts with misspecification in linear function approximation (e.g., Jin et al. (2020)), where the bias typically scales with the problem dimension $d$. This property highlights the robustness of our approach in large-scale and high-dimensional settings, as the impact of parameter error does not amplify with the complexity of the state-action spaces.

### D.6 Improved regret bounds via tighter constants

We refine the upper bound for the follower's value function established in prior work. In particular, Theorem K.4 shows that the constant $B_A$—which upper bounds the follower's value functions in the myopic setting—can be reduced by a factor of 2 compared to the bound in Chen et al. (2023). This seemingly modest refinement has a substantial impact on the regret bounds.

Specifically, $B_A$ appears in the exponent of the term $C_\epsilon^{(3)}$ (see Equation (108) and the regret expressions in Equations (37) to (39)), which governs the rate at which the IC approximation impacts the overall regret. Since $C_\epsilon^{(3)}$ grows exponentially with $B_A$, the reduction in $B_A$ leads to an exponential improvement in the constant factors of the regret bound. This improvement is particularly significant in settings with tight IC constraints, where exponential terms dominate the convergence rate.

The derivation relies on a careful analysis of the Bellman equations for the follower under quantal response dynamics and exploits structural simplifications available in the myopic regime. This improvement sharpens existing bounds and enhances the practical viability of regret guarantees in leader-follower learning with policy recommendation.

# E    LIMITATIONS AND FUTURE WORK

## E.1    LIMITATIONS

Our work is a first theoretical study of incentive-compatible learning with policy recommendations, and as such, we adopt several modeling assumptions that make the analysis tractable while highlighting the central challenges. We summarize them below and discuss how future work may relax these assumptions.

**Known follower response parameters.**    We assume that the leader knows the follower's response model parameters $\eta$ and $\tau$. While these parameters may not be available in practice, they can often be estimated from online interaction data. Incorporating parameter uncertainty would introduce an additional exploration–exploitation tradeoff under IC constraints, which is technically interesting but beyond the scope of this work. Extending our framework to simultaneously learn these parameters is a natural and promising research direction.

**Finite follower action space.**    Our main theoretical result (Theorem 6.2) requires the follower's action space $\mathcal{B}$ to be finite, since our proof uses Theorems C.3 and C.4, which establish uniform positive lower bounds only in this setting. This property is critical for achieving regret guarantees under IC constraints. We conjecture that analogous guarantees may not extend to continuous action spaces, but a formal treatment of this case remains open. Clarifying this boundary would further illuminate the fundamental limits of incentive-compatible learning.

**Matrix game setting.**    To make the analysis transparent, we focus on a matrix game rather than a more general Markov game. This choice is deliberate: it isolates the technical challenges introduced by IC constraints without additional complexity from dynamic state transitions. We believe our approach can be extended to Markov games with moderate technical effort, and view this as a valuable next step.

**Myopic follower assumption.**    We assume that the follower is myopic, optimizing for immediate reward. This assumption is standard in Stackelberg game formulations and allows us to highlight the incentive-compatibility aspect of policy recommendations. Nonetheless, extending our results to farsighted followers (e.g., under discounted utilities as in Chen et al. (2023)) is conceptually straightforward and would broaden the applicability of our framework.

Overall, these assumptions should be seen not as shortcomings, but as natural starting points that enable a clear theoretical characterization of the problem. Relaxing them provides fertile ground for future work and could significantly expand the scope of incentive-compatible learning with policy recommendations.

## E.2    FUTURE WORK

While our work focuses on incentive-compatible learning in leader-follower matrix games, several natural and promising directions remain open for future research.

1. **Extension to Markov games.** Our current setting considers matrix games, which abstract away state transitions. Extending the framework to Markov games would allow for modeling dynamic environments and long-term planning, but also introduces substantial challenges, such as dealing with temporal credit assignment under incentive constraints.

2. **Offline learning with incentive constraints.** While our analysis focuses on online learning, incentive-compatible learning from offline data (e.g., historical leader-follower interactions) is of both theoretical and practical interest. Adapting our framework to the offline setting

could enable policy recommendation in domains where online exploration is expensive or risky.

3. **Farsighted follower models.** We currently assume the follower is myopic. Modeling far-sighted followers who optimize cumulative rewards would better capture strategic behaviors in long-term interactions. This extension would require rethinking the notion of incentive compatibility and may lead to qualitatively different equilibrium outcomes.

Exploring these directions would broaden the applicability of our approach to more realistic and complex multi-agent systems, and deepen our understanding of incentive-compatible learning in dynamic strategic environments.

## F    REGRET ANALYSIS OF MAIN THEOREMS

### F.1    SKETCH OF PROOFS

In this subsection, we provide the sketch of proofs of Theorem 6.1 and Theorem 6.2.

#### F.1.1    PROOF SKETCH OF THEOREM 6.1

Firstly, we decompose the suboptimality $\text{SubOpt}(\pi_t, y_t)$ into three terms:

$$
\begin{aligned}
\text{SubOpt}(\pi_t, y_t) := & \left\langle U, \pi^* \otimes \nu^* \right\rangle - \left\langle U, \pi_t \otimes \nu_{\theta^*}^{\pi_t, y_t} \right\rangle \\
= & \underbrace{\left\langle U - \widetilde{U}_t, \pi^* \otimes \nu^* \right\rangle}_{=: S_t^1} + \underbrace{\left\langle \widetilde{U}_t - U, \pi_t \otimes \nu_{\theta^*}^{\pi_t, y_t} \right\rangle}_{=: S_t^2} + \underbrace{\left\langle \widetilde{U}_t, \pi^* \otimes \nu^* - \pi_t \otimes \nu_{\theta^*}^{\pi_t, y_t} \right\rangle}_{=: S_t^3}.
\end{aligned}
$$

For $S_t^1$ and $S_t^2$, we can observe that both of them are an expectation of a difference between two value functions, with respect to different distributions. Hence, we can utilize the optimistic construction to derive high probability bounds on $S_t^1$ and $S_t^2$. Specifically, we have by optimism that $S_t^1 \leq 0$ with high probability. Also, by construction, we can upper bound $S_t^2$ by the bonus function $\Gamma_t^1$ plus a martingale difference sequence with a uniform finite upper bound, and then derive a $\widetilde{\mathcal{O}}(\sqrt{T})$ bound on the martingale difference sequence by applying the Azuma-Hoeffding inequality. Note that the analysis of $S_t^1$ and $S_t^2$ are the same for both Scheme 1 and Scheme 2, because they do not involve difference between two distributions. Then, it suffices to control the bonus function $\Gamma_t^1$.

However, $S_t^3$ is different from $S_t^1$ and $S_t^2$ in the sense that it is the difference between two expectations, of the same value function but with respect to different distributions. Therefore, we need to control the difference between two distributions to control $S_t^3$. As a result, the analysis of $S_t^3$ is different for Scheme 1 and Scheme 2. Actually, the analysis of $S_t^3$ for Scheme 2 is more complicated than that for Scheme 1, due to the explicit introduction of the bonus function $\Gamma^2$ to the objective function in Scheme 2. Next, we focus on how to bound $S_t^3$ for Scheme 1 and Scheme 2, respectively, by using the results from response model errors provided in Appendix J.

**Bounding $S_t^3$ for Scheme 1.** We can further decompose $S_t^3 = \left\langle \widetilde{U}_t, \pi^* \otimes \nu^* - \pi_t \otimes \nu_{\theta^*}^{\pi_t, y_t} \right\rangle$ by

$$
S_t^3 = \underbrace{\left\langle \widetilde{U}_t, \pi^* \otimes \nu^* - \pi_t \otimes \nu_{\widehat{\theta}_t}^{\pi_t, y_t} \right\rangle}_{=: S_t^{3,1}} + \underbrace{\left\langle \widetilde{U}_t, \pi_t \otimes \nu_{\widehat{\theta}_t}^{\pi_t, y_t} - \pi_t \otimes \nu_{\theta^*}^{\pi_t, y_t} \right\rangle}_{=: S_t^{3,2}}.
$$

For $S_t^{3,1}$, by the optimality of $\pi_t$ and $\widehat{\theta}_t$ in Scheme 1, we have $S_t^{3,1} \leq 0$. For $S_t^{3,2}$, since $\widetilde{U}_t$ is upper bounded by 1, we can upper bound $S_t^{3,2}$ by the TV distance between $\nu_{\widehat{\theta}_t}^{\pi_t, y_t}$ and $\nu_{\theta^*}^{\pi_t, y_t}$, denoted as $\text{TV}(\nu_{\widehat{\theta}_t}^{\pi_t, y_t}, \nu_{\theta^*}^{\pi_t, y_t})$. Then, by applying the results of model response errors in Appendix J and confidence sets in Appendix I, we can upper bound $S_t^{3,2}$ by the bonus function $\Gamma_t^2$.

Hence, we derive the bonus function $\Gamma_t^2$ as a high probability upper bound on $S_t^3$ for Scheme 1.

**Bounding $S_t^3$ for Scheme 2.** Compared with Scheme 1, we decompose $S_t^3 = \left\langle \widetilde{U}_t, \pi^* \otimes \nu^* - \pi_t \otimes \nu_{\theta^*}^{\pi_t, y_t} \right\rangle$ in a more complicated way as follows:

$$S_t^3 = \underbrace{\left\langle \widetilde{U}_t, \pi^* \otimes \nu^* - \pi^* \otimes \widehat{y}_t^* \right\rangle}_{=:S_t^{3,3}} + \underbrace{\left\langle \widetilde{U}_t, \pi^* \otimes \widehat{y}_t^* \right\rangle - \widehat{W}_t}_{=:S_t^{3,4}}$$

$$+ \underbrace{\widehat{W}_t - \left\langle \widetilde{U}_t, \pi_t \otimes y_t \right\rangle}_{=:S_t^{3,5}} + \underbrace{\left\langle \widetilde{U}_t, \pi_t \otimes y_t - \pi_t \otimes \nu_{\theta^*}^{\pi_t, y_t} \right\rangle}_{=:S_t^{3,6}}.$$

Similar to $S_t^{3,2}$, we can upper bound $S_t^{3,3}$ by the TV distance between $\nu^*$ and $\widehat{y}_t^*$, denoted as $\mathrm{TV}(\nu^*, \widehat{y}_t^*)$. For $S_t^{3,4}$, by the optimality of $\widehat{W}_t$ as in equation 118, we can upper bound it by negative $\Gamma_t^2$ such that the sum of $S_t^{3,3}$ and $S_t^{3,4}$ is nonpositive, by using the results of quantal response errors in Theorem J.3. For $S_t^{3,5}$, we observe that it is equal to a $\Gamma_t^2$ function up to a properly chosen constant factor. Here the parameter $\theta$ used in $\Gamma_t^2$ is the MLE $\widehat{\theta}_t^{\mathrm{MLE}}$, which can be upper bounded by the counterpart with the true parameter $\theta^*$ up to some constant factor. We can observe that $S_t^{3,6}$ in Scheme 2 is essentially the same as $S_t^{3,2}$ in Scheme 1. So we just need to analyze $S_t^{3,3}$, $S_t^{3,4}$ and $S_t^{3,5}$ for Scheme 2.

From the above analysis, we can observe that there is only a difference of the constant factor between the upper bounds on $S_t^3$ for Scheme 1 and Scheme 2. The upper bound is a $\Gamma_t^2$ function, and then it suffices to control the bonus function $\Gamma_t^2$ within the rate of $\widetilde{\mathcal{O}}(\sqrt{T})$ by using the elliptical potential lemma.

Finally, by summing the upper bounds on $S_t^1$, $S_t^2$ and $S_t^3$, we obtain an upper bound on the regret $\mathrm{Reg}(T)$ which is $\widetilde{\mathcal{O}}(d^2 \sqrt{T})$, for both schemes of Algorithm 1.

### F.1.2 Proof Sketch of Theorem 6.2

Firstly, thanks to the regularization terms, we can show that there exist two constants $\mu_{\mathrm{L}}, \nu_{\mathrm{L}} > 0$ such that $\mu_\theta^\pi(b) \geq \mu_{\mathrm{L}}$ and $\nu_\theta^{\pi, y}(b) \geq \nu_{\mathrm{L}}$ for any $b \in \mathcal{B}$, $\pi \in \Pi$, $y \in \mathcal{Y}_{\mathrm{QR}}$ and $\theta \in \Theta$, as stated in Theorem C.3 and Theorem C.4, respectively. These results rely on the condition that $\mathcal{B}$ is finite. Then, we develop Theorem C.5 to claim that when $y \in \mathcal{Y}_{\mathrm{QR}}$ and $\|y - \nu_\theta^{\pi, y}\|_1$ is sufficiently small compared to $\nu_{\mathrm{L}}$, we can upper bound $\|y - \mu_\theta^\pi\|_1$ by $\|y - \nu_\theta^{\pi, y}\|_1$ up to some constant, which can be regarded as some extension to Theorem C.1.

We then use Theorem C.5 to analyze the relaxation of the ICV constraint. We define

$$F_\theta(\sigma) := \max_{\mathrm{TV}(y, \nu_\theta^{\pi, y}) \leq \sigma} \pi^\top A \nu_\theta^{\pi, y},$$

and we show in Theorem C.6 that, if $\mathrm{TV}(y, \nu_\theta^{\pi, y})$ is sufficiently small compared to $\nu_{\mathrm{L}}$, the degree of perturbation $F_\theta(\sigma) - F_\theta(0)$ can be upper bounded by $\mathrm{TV}(y, \nu_\theta^{\pi, y})$ up to some constant.

Under the condition that $\mathrm{Reg}^{\mathrm{ICV}}(T) = \widetilde{\mathcal{O}}(\sqrt{T})$, we can show that the cardinality of the index set $[T]_a$ where $\mathrm{TV}(y_t, \nu_{\theta^*}^{\pi_t, y_t})$ is large (greater than $\nu_{\mathrm{L}}/4$) is $\widetilde{\mathcal{O}}(\sqrt{T})$. We let $\sigma_t = \mathrm{TV}(y_t, \nu_{\theta^*}^{\pi_t, y_t})$, and then we can control $\sum_{t \in [T]_a} (F_{\theta^*}(\sigma_t) - F_{\theta^*}(0))$ under $\widetilde{\mathcal{O}}(\sqrt{T})$. For $t$ such that $\mathrm{TV}(y_t, \nu_{\theta^*}^{\pi_t, y_t})$ is small (no greater than $\nu_{\mathrm{L}}/4$), by Theorem C.6 and $\mathrm{Reg}^{\mathrm{ICV}}(T) = \widetilde{\mathcal{O}}(\sqrt{T})$, we can control $\sum_{t \in [T]_b} (F_{\theta^*}(\sigma_t) - F_{\theta^*}(0))$ under $\widetilde{\mathcal{O}}(\sqrt{T})$. Therefore, we can control $\sum_{t=1}^T (F_{\theta^*}(\sigma_t) - F_{\theta^*}(0))$ under $\widetilde{\mathcal{O}}(\sqrt{T})$.

Next, we divide the index set $[T]$ into two disjoint sets $[T]_+$ and $[T]_-$ where

$$[T]_+ := \{t \in [T] : \mathrm{SubOpt}(\pi_t, y_t) \geq 0\}, \quad [T]_- := \{t \in [T] : \mathrm{SubOpt}(\pi_t, y_t) < 0\}.$$

For any $t \in [T]_-$, we can observe that $|\mathrm{SubOpt}(\pi_t, y_t)| \leq F_{\theta^*}(\sigma_t) - F_{\theta^*}(0)$, and further

$$\sum_{t \in [T]_-} |\mathrm{SubOpt}(\pi_t, y_t)| \leq \sum_{t=1}^T (F_{\theta^*}(\sigma_t) - F_{\theta^*}(0)) \leq \widetilde{\mathcal{O}}(\sqrt{T}),$$

On the other hand, under the condition that $\text{Reg}(T) = \widetilde{\mathcal{O}}(\sqrt{T})$, we can also control

$$\sum_{t \in [T]_+} |\text{SubOpt}(\pi_t, y_t)| = \text{Reg}(T) + \sum_{t \in [T]_-} |\text{SubOpt}(\pi_t, y_t)|$$

under $\widetilde{\mathcal{O}}(\sqrt{T})$. Therefore, by combining the above results, we can control the absolute regret

$$\text{Reg}^{\text{abs}}(T) := \sum_{t \in [T]} |\text{SubOpt}(\pi_t, y_t)| = \sum_{t \in [T]_+} |\text{SubOpt}(\pi_t, y_t)| + \sum_{t \in [T]_-} |\text{SubOpt}(\pi_t, y_t)|$$

under $\widetilde{\mathcal{O}}(\sqrt{T})$, which completes the proof.

### F.2 AUXILIARY LEMMAS FOR THEOREM 6.1

In order to prove Theorem 6.1, we first provide the following auxiliary lemmas. Thereafter, we will combine the results from these lemmas to prove Theorem 6.1 in Appendix F.3.

**Lemma F.1** (Leader's regret bounded by bonus functions, Scheme 1). *Under the conditions of Theorem 6.1, the leader's regret $\text{Reg}(T)$ for Scheme 1 satisfies*

$$\text{Reg}(T) \leq 2\sum_{t=1}^{T} \Gamma_t^1(a_t, b_t) + \sum_{t=1}^{T} \Gamma_t^2(\pi_t, y_t; \theta^*) + \sqrt{2T \log\left(\frac{3}{p}\right)} \tag{31}$$

*with probability at least $1 - p$.*

*Proof.* See Appendix H.1 for a detailed proof. $\qquad\square$

**Lemma F.2** (Leader's regret bounded by bonus functions, Scheme 2). *Under the conditions of Theorem 6.1, the leader's regret $\text{Reg}(T)$ for Scheme 2 satisfies*

$$\text{Reg}(T) \leq 2\sum_{t=1}^{T} \Gamma_t^1(a_t, b_t) + (1 + C_{\text{ii}} \cdot \exp(4\varepsilon B_A)) \cdot \sum_{t=1}^{T} \Gamma_t^2(\pi_t, y_t; \theta^*) + \sqrt{2T \log\left(\frac{3}{p}\right)} \tag{32}$$

*with probability at least $1 - p$. Here, the constant $C_{\text{ii}}$ is defined as*

$$C_{\text{ii}} := C_{\text{QF}} C_{\eta/\varepsilon} = \exp(2\eta) |\mathcal{B}|^{\frac{\tau}{\tau + \eta^{-1}}} \max\left\{ \frac{C_\eta^{(3)}}{C_\varepsilon^{(3)}}, \frac{\eta}{\varepsilon} \right\} \tag{33}$$

*by equation 184 and equation 167, where $\varepsilon = (\tau + \eta^{-1})^{-1}$ and $C_x^{(2)} = x^2 \exp(2xB_A)(2 + xB_A \exp(2xB_A))/2$ for $x \in \{\eta, \varepsilon\}$.*

*Proof.* See Appendix H.2 for a detailed proof. $\qquad\square$

**Lemma F.3** (ICV regret bounded by bonus functions, for both schemes). *Under the conditions of Theorem 6.1, the ICV regret $\text{Reg}^{\text{ICV}}(T)$ for both Scheme 1 and Scheme 2 satisfies*

$$\text{Reg}^{\text{ICV}}(T) = \sum_{t=1}^{T} \text{SubOpt}^{\text{ICV}}(\pi_t, y_t) \leq \frac{1}{2}\sum_{t=1}^{T} \Gamma_t^2(\pi_t, y_t; \theta^*). \tag{34}$$

*with probability at least $1 - p$.*

*Proof.* See Appendix H.3 for a detailed proof. $\qquad\square$

**Lemma F.4** (Upper bounds on the bonus functions). *Under the conditions of Theorem 6.1, the bonus functions $\Gamma^1$ and $\Gamma^2$ satisfy*

$$\sum_{t=1}^{T} \Gamma_t^1(a_t, b_t) \leq C_1 d\sqrt{Td \log(T/d) \log(2dT^2/(p/3))} \tag{35}$$

*and*

$$\sum_{t=1}^{T} \Gamma_t^2(\pi_t, y_t; \theta^*) \leq \varepsilon \sqrt{8C_\varepsilon^2 \beta + 4B_\Theta^2} \sqrt{C_0 dT \log\left(1 + 4B_\phi^2 T/d\right)}$$
$$+ C_\varepsilon^{(3)}(8C_\varepsilon^2 \beta + 4B_\Theta^2) C_0 d \log\left(1 + 4B_\phi^2 T/d\right). \tag{36}$$

*Proof.* See Appendix H.4 for a detailed proof. □

In the sequel, we denote $\nu^* := \nu_{\theta^*}^{\pi^*, y^*}$, where $(\pi^*, y^*)$ is the equilibrium and $\theta^*$ is the true parameter.

### F.3    PROOF OF THEOREM 6.1

*Proof.* We are ready to prove Theorem 6.1 for Scheme 1 and Scheme 2 by using the above lemmas proven in this section.

**Scheme 1.** Combining the results of the leader's regret and the regret of ICV from Theorem F.1, Theorem F.3 and Theorem F.4, we have that, with probability at least $1 - p$,

$$
\begin{aligned}
\text{Reg}(T) = \sum_{t=1}^{T} \text{SubOpt}(\pi_t, y_t) &\leq 2 \sum_{t=1}^{T} \Gamma_t^1(a_t, b_t) + \sum_{t=1}^{T} \Gamma_t^2(\pi_t, y_t; \theta^*) + \sqrt{2T \log\left(\frac{3}{p}\right)} \\
&\leq 2C_1 d \sqrt{Td \log(T/d) \log(2dT^2/(p/3))} + \sqrt{2T \log(3p^{-1})} \\
&\quad + \varepsilon \sqrt{8C_\varepsilon^2 \beta + 4B_\Theta^2} \sqrt{C_0 dT \log\left(1 + 4B_\phi^2 T/d\right)} \\
&\quad + C_\varepsilon^{(3)} (8C_\varepsilon^2 \beta + 4B_\Theta^2) C_0 d \log\left(1 + 4B_\phi^2 T/d\right) \\
&= \widetilde{\mathcal{O}}(d\sqrt{dT}) + \widetilde{\mathcal{O}}(\sqrt{T}) + \widetilde{\mathcal{O}}(\sqrt{\beta dT}) + \widetilde{\mathcal{O}}(\beta d) = \widetilde{\mathcal{O}}(d^2 \sqrt{T}),
\end{aligned}
\tag{37}
$$

and

$$
\begin{aligned}
\text{Reg}^{\text{ICV}}(T) &\leq \frac{1}{2} \sum_{t=1}^{T} \Gamma_t^2(\pi_t, y_t; \theta^*) \\
&\leq \frac{1}{2} \varepsilon \sqrt{8C_\varepsilon^2 \beta + 4B_\Theta^2} \sqrt{C_0 dT \log\left(1 + 4B_\phi^2 T/d\right)} \\
&\quad + \frac{1}{2} C_\varepsilon^{(3)} (8C_\varepsilon^2 \beta + 4B_\Theta^2) C_0 d \log\left(1 + 4B_\phi^2 T/d\right). \\
&= \widetilde{\mathcal{O}}(\sqrt{\beta dT}) + \widetilde{\mathcal{O}}(\beta d) = \widetilde{\mathcal{O}}(d^2 \sqrt{T}),
\end{aligned}
\tag{38}
$$

where we use the fact that $\beta = \widetilde{\mathcal{O}}(d)$.

The rate $\widetilde{\mathcal{O}}(d^2 \sqrt{T})$ of the leader's regret $\text{Reg}(T)$ matches the regret bound in Theorem 6.1 of Chen et al. (2023) with the length of horizon $H = 1$. The regret of ICV $\text{Reg}^{\text{ICV}}(T)$ is also $\widetilde{\mathcal{O}}(d^2 \sqrt{T})$.

Note: The bound on $\text{Reg}(T)$ is conditioned on equation 94, equation 101, $\theta^* \in \mathcal{C}_\Theta^t(\beta)$ and equation 109 in Theorem I.1. The bound on $\text{Reg}^{\text{ICV}}(T)$ is conditioned on equation 109 in Theorem I.1.

**Scheme 2.** By comparing equation 114 and equation 132, we can observe that there is only an additional factor $(1 + C_{\text{ii}} \cdot \exp(4\varepsilon B_A))$ on $\sum_{t=1}^{T} \Gamma_t^2(\pi_t, y_t; \theta^*)$ for Scheme 2, compared with that of Scheme 1. Thus, combining the results of the leader's regret and the regret of ICV from Theorem F.2,

Theorem F.3 and Theorem F.4, we have that, with probability at least $1 - p$,

$$\text{Reg}(T) = \sum_{t=1}^{T} \text{SubOpt}(\pi_t, y_t)$$

$$\leq 2 \sum_{t=1}^{T} \Gamma_t^1(a_t, b_t) + (1 + C_{\text{ii}} \cdot \exp\left(4\varepsilon B_A\right)) \cdot \sum_{t=1}^{T} \Gamma_t^2(\pi_t, y_t; \theta^*) + \sqrt{2T \log\left(\frac{3}{p}\right)}$$

$$\leq 2C_1 d\sqrt{Td \log(T/d) \log(2dT^2/(p/3))} + \sqrt{2T \log(3p^{-1})}$$

$$+ (1 + C_{\text{ii}} \cdot \exp\left(4\varepsilon B_A\right)) \cdot \varepsilon\sqrt{8C_\varepsilon^2 \beta + 4B_\Theta^2}\sqrt{C_0 dT \log\left(1 + 4B_\phi^2 T/d\right)}$$

$$+ (1 + C_{\text{ii}} \cdot \exp\left(4\varepsilon B_A\right)) \cdot C_\varepsilon^{(3)}(8C_\varepsilon^2 \beta + 4B_\Theta^2)C_0 d \log\left(1 + 4B_\phi^2 T/d\right)$$

$$= \widetilde{\mathcal{O}}(d\sqrt{dT}) + \widetilde{\mathcal{O}}(\sqrt{T}) + \widetilde{\mathcal{O}}(\sqrt{\beta dT}) + \widetilde{\mathcal{O}}(\beta d) = \widetilde{\mathcal{O}}(d^2\sqrt{T}). \tag{39}$$

Therefore, we have completed the proof of Theorem 6.1 for both Scheme 1 and Scheme 2. $\qquad \square$

### F.4  PROOF OF THEOREM 6.2

*Proof.* The proof for the first part is straightforward. For the second part, recall the definition of ICV that $\text{ICV}(\pi, y; \theta) := \text{TV}(y, \nu_\theta^{\pi, y})$. Under the condition that $\text{Reg}^{\text{ICV}}(T) = \widetilde{\mathcal{O}}(\sqrt{T})$, we construct a sequence $\{\sigma_t\}_{t \in \mathbb{N}}$ where $\sigma_t = \text{TV}(y_t, \nu_{\theta^*}^{\pi_t, y_t})$ for $t \in \mathbb{N}$. Then, we immediately have $\sum_{t=1}^{T} \sigma_t = \widetilde{\mathcal{O}}(\sqrt{T})$.

For any $T \in \mathbb{N}$, we can divide the index set $[T]$ into two disjoint sets $[T]_a$ and $[T]_b$ as follows:

$$[T]_a := \left\{ t \in [T] : \text{TV}(y_t, \nu_{\theta^*}^{\pi_t, y_t}) > \frac{1}{4}\nu_{\text{L}} \right\}, \quad [T]_b := \left\{ t \in [T] : \text{TV}(y_t, \nu_{\theta^*}^{\pi_t, y_t}) \leq \frac{1}{4}\nu_{\text{L}} \right\}. \tag{40}$$

It is evident that $[T] = [T]_a \cup [T]_b$ and $[T]_a \cap [T]_b = \varnothing$, so we have the following decomposition:

$$\sum_{t=1}^{T} (F_{\theta^*}(\sigma_t) - F_{\theta^*}(0)) = \sum_{t \in [T]_a} (F_{\theta^*}(\sigma_t) - F_{\theta^*}(0)) + \sum_{t \in [T]_b} (F_{\theta^*}(\sigma_t) - F_{\theta^*}(0)). \tag{41}$$

We aim to show the two terms in the RHS of equation 41 are both $\widetilde{\mathcal{O}}(\sqrt{T})$.

We can also decompose and lower bound $\text{Reg}^{\text{ICV}}(T)$ by

$$\text{Reg}^{\text{ICV}}(T) = \sum_{t \in [T]} \text{TV}(y_t, \nu_{\theta^*}^{\pi_t, y_t}) = \sum_{t \in [T]_a} \text{TV}(y_t, \nu_{\theta^*}^{\pi_t, y_t}) + \sum_{t \in [T]_b} \text{TV}(y_t, \nu_{\theta^*}^{\pi_t, y_t})$$

$$\geq \sum_{t \in [T]_a} \frac{1}{4}\nu_{\text{L}} + \sum_{t \in [T]_b} \text{TV}(y_t, \nu_{\theta^*}^{\pi_t, y_t}) = \frac{1}{4}\nu_{\text{L}} \cdot |[T]_a| + \sum_{t \in [T]_b} \text{TV}(y_t, \nu_{\theta^*}^{\pi_t, y_t})$$

$$\geq \frac{1}{4}\nu_{\text{L}} \cdot |[T]_a|, \tag{42}$$

where the first inequality is by the construction of $[T]_a$ and the last inequality is because the TV distance is always nonnegative. Then, by the condition that $\text{Reg}^{\text{ICV}}(T) = \widetilde{\mathcal{O}}(\sqrt{T})$, we have that the cardinality of $[T]_a$ as a function of $T$ satisfies

$$|[T]_a| \leq \frac{4}{\nu_{\text{L}}} \cdot \text{Reg}^{\text{ICV}}(T) \leq \widetilde{\mathcal{O}}(\sqrt{T}). \tag{43}$$

Hence, we have

$$\sum_{t \in [T]_a} (F_{\theta^*}(\sigma_t) - F_{\theta^*}(0)) \leq \sum_{t \in [T]_a} 1 = |[T]_a| \leq \widetilde{\mathcal{O}}(\sqrt{T}), \tag{44}$$

where the first inequality is because $F_\theta(\sigma_t) - F_\theta(0)$ is upper bounded by 1 for any $\theta \in \Theta$.

By Theorem C.6 and $\mathrm{Reg}^{\mathrm{ICV}}(T) = \widetilde{\mathcal{O}}(\sqrt{T})$, we have

$$
\begin{aligned}
\sum_{t \in [T]_b} (F_{\theta^*}(\sigma_t) - F_{\theta^*}(0)) &\leq \sum_{t \in [T]_b} C_F \cdot \sigma_t = C_F \cdot \sum_{t \in [T]_b} \mathrm{TV}(y_t, \nu_{\theta^*}^{\pi_t, y_t}) \\
&\leq C_F \cdot \sum_{t=1}^{T} \mathrm{TV}(y_t, \nu_{\theta^*}^{\pi_t, y_t}) \\
&= C_F \cdot \mathrm{Reg}^{\mathrm{ICV}}(T) \leq \widetilde{\mathcal{O}}(\sqrt{T}),
\end{aligned}
\tag{45}
$$

where the first inequality is because equation 30 in Theorem C.6 holds for any $\sigma \in [0, \frac{1}{4}\nu_{\mathrm{L}}]$, and the last inequality is by the condition that $\mathrm{Reg}^{\mathrm{ICV}}(T) = \widetilde{\mathcal{O}}(\sqrt{T})$. Here, the constant factor $C_F$ is defined by $C_F := 2 + \frac{8 \cdot 3^{\tau\eta}(1+\tau\eta)}{\nu_{\mathrm{L}}}$ as is in Theorem C.6.

Combining equation 41, equation 44 and equation 45, we have

$$
\sum_{t=1}^{T} (F_{\theta^*}(\sigma_t) - F_{\theta^*}(0)) \leq \widetilde{\mathcal{O}}(\sqrt{T}).
\tag{46}
$$

For a sequence $(\pi_t, y_t)_{t \in \mathbb{N}}$, we have by equation 11 that

$$
\mathrm{SubOpt}(\pi_t, y_t) = \max_{\pi' \in \Pi} \pi'^{\top} A \mu_{\theta^*}^{\pi'} - \pi_t^{\top} A \nu_{\theta^*}^{\pi_t, y_t}.
\tag{47}
$$

For $T \in \mathbb{N}$, we can divide the index set $[T]$ into two disjoint sets $[T]_+$ and $[T]_-$ as follows:

$$
[T]_+ := \{t \in [T] : \mathrm{SubOpt}(\pi_t, y_t) \geq 0\}, \quad [T]_- := \{t \in [T] : \mathrm{SubOpt}(\pi_t, y_t) < 0\}.
\tag{48}
$$

It is evident that $[T] = [T]_+ \cup [T]_-$ and $[T]_+ \cap [T]_- = \varnothing$. Then, we can decompose the regret by

$$
\begin{aligned}
\mathrm{Reg}(T) &:= \sum_{t \in [T]} \mathrm{SubOpt}(\pi_t, y_t) = \sum_{t \in [T]_+} \mathrm{SubOpt}(\pi_t, y_t) + \sum_{t \in [T]_-} \mathrm{SubOpt}(\pi_t, y_t) \\
&= \sum_{t \in [T]_+} |\mathrm{SubOpt}(\pi_t, y_t)| - \sum_{t \in [T]_-} |\mathrm{SubOpt}(\pi_t, y_t)|,
\end{aligned}
\tag{49}
$$

where the second equality is because $[T] = [T]_+ \cup [T]_-$ and $[T]_+ \cap [T]_- = \varnothing$, and the third equality is because $|\mathrm{SubOpt}(\pi_t, y_t)| = \mathrm{SubOpt}(\pi_t, y_t)$ for any $t \in [T]_+$ and $|\mathrm{SubOpt}(\pi_t, y_t)| = -\mathrm{SubOpt}(\pi_t, y_t)$ for any $t \in [T]_-$.

Then, for any $t \in [T]_-$, we can observe that

$$
\begin{aligned}
|\mathrm{SubOpt}(\pi_t, y_t)| = -\mathrm{SubOpt}(\pi_t, y_t) &= \pi_t^{\top} A \nu_{\theta^*}^{\pi_t, y_t} - \max_{\pi' \in \Pi} \pi'^{\top} A \mu_{\theta^*}^{\pi'} \\
&\leq \max_{\mathrm{TV}(y, \nu_{\theta^*}^{\pi, y}) \leq \sigma_t} \pi^{\top} A \nu_{\theta^*}^{\pi, y} - \max_{\pi' \in \Pi} \pi'^{\top} A \mu_{\theta^*}^{\pi'} \\
&= F_{\theta^*}(\sigma_t) - F_{\theta^*}(0).
\end{aligned}
\tag{50}
$$

Further, we have

$$
\begin{aligned}
\sum_{t \in [T]_-} |\mathrm{SubOpt}(\pi_t, y_t)| &\leq \sum_{t \in [T]_-} (F_{\theta^*}(\sigma_t) - F_{\theta^*}(0)) \\
&\leq \sum_{t \in [T]_-} (F_{\theta^*}(\sigma_t) - F_{\theta^*}(0)) + \sum_{t \in [T]_+} (F_{\theta^*}(\sigma_t) - F_{\theta^*}(0)) \\
&= \sum_{t=1}^{T} (F_{\theta^*}(\sigma_t) - F_{\theta^*}(0)) \leq \widetilde{\mathcal{O}}(\sqrt{T}),
\end{aligned}
\tag{51}
$$

where the second inequality is because $F_{\theta^*}(\sigma_t) - F_{\theta^*}(0)$ is nonnegative for any $t$, and the last inequality is by equation 46.

Under the condition that $\mathrm{Reg}(T) = \widetilde{\mathcal{O}}(\sqrt{T})$, we have

$$
\sum_{t \in [T]_+} |\mathrm{SubOpt}(\pi_t, y_t)| = \mathrm{Reg}(T) + \sum_{t \in [T]_-} |\mathrm{SubOpt}(\pi_t, y_t)|
$$
$$
= \widetilde{\mathcal{O}}(\sqrt{T}) + \widetilde{\mathcal{O}}(\sqrt{T}) = \widetilde{\mathcal{O}}(\sqrt{T}). \tag{52}
$$

Therefore, we have the absolute regret

$$
\mathrm{Reg}^{\mathrm{abs}}(T) := \sum_{t \in [T]} |\mathrm{SubOpt}(\pi_t, y_t)|
$$
$$
= \sum_{t \in [T]_+} |\mathrm{SubOpt}(\pi_t, y_t)| + \sum_{t \in [T]_-} |\mathrm{SubOpt}(\pi_t, y_t)|
$$
$$
= \widetilde{\mathcal{O}}(\sqrt{T}) + \widetilde{\mathcal{O}}(\sqrt{T}) = \widetilde{\mathcal{O}}(\sqrt{T}), \tag{53}
$$

which completes the proof of Theorem 6.2. $\qquad\square$

### F.5 PROOFS FOR ROBUSTNESS ANALYSIS IN SECTION 6.2

#### F.5.1 PROOF OF THEOREM 6.4

*Proof.* We analyze the regret decomposition under parameter estimation errors. Let the true behavioral parameters be $\eta, \tau$ and the estimated parameters be $\widehat{\eta}, \widehat{\tau}$. We assume the estimation errors are bounded by $\zeta$, i.e., $|\widehat{\eta} - \eta| \le \zeta$ and $|\widehat{\tau} - \tau| \le \zeta$.

**Step 1: Lipschitz Continuity of the Follower's Response.** Recall the closed-form expression for the follower's response model derived in Equation (5):

$$
\nu_\theta^{\pi,y}(b; \eta, \tau) = \frac{[y(b)]^{\frac{\tau}{\tau + \eta^{-1}}} \exp\left(\frac{Q_\theta^\pi(b)}{\tau + \eta^{-1}}\right)}{\sum_{b' \in \mathcal{B}} [y(b')]^{\frac{\tau}{\tau + \eta^{-1}}} \exp\left(\frac{Q_\theta^\pi(b')}{\tau + \eta^{-1}}\right)}.
$$

Let $\alpha(\eta, \tau) = (\tau + \eta^{-1})^{-1}$. The exponents become $\alpha Q_\theta^\pi(b)$ and the power on $y(b)$ becomes $\tau\alpha$. We observe that the function $\nu_\theta^{\pi,y}$ is a composition of smooth functions (exponential, power, and normalization) with respect to $\eta$ and $\tau$, provided $\eta > 0$ and $y(b) > 0$. Since the domain of parameters is compact and the reward function $Q$ is bounded, the partial derivatives of $\nu$ with respect to $\eta$ and $\tau$ are well-defined and bounded. By the mean value theorem, the mapping from $(\eta, \tau)$ to the probability vector $\nu_\theta^{\pi,y}$ is Lipschitz continuous. Therefore, there exists a problem-dependent Lipschitz constant $L_\nu > 0$ such that for any estimates satisfying the error bound $\zeta$:

$$
\|\nu_\theta^{\pi,y}(\cdot; \widehat{\eta}, \widehat{\tau}) - \nu_\theta^{\pi,y}(\cdot; \eta, \tau)\|_1 \le L_\nu(|\widehat{\eta} - \eta| + |\widehat{\tau} - \tau|) \le 2L_\nu\zeta.
$$

For simplicity of notation, we absorb the constant factor 2 into $L_\nu$ and write the bound as $L_\nu\zeta$.

**Step 2: Bias in Leader's Objective Function.** Let $J(\pi, y)$ denote the leader's true expected reward under the true parameters, and let $\widehat{J}(\pi, y)$ denote the expected reward estimated by the leader using $\widehat{\eta}$ and $\widehat{\tau}$. For any policy pair $(\pi, y)$, the difference between the true and estimated objectives is:

$$
|J(\pi, y) - \widehat{J}(\pi, y)| = \left| \sum_{a \in \mathcal{A}, b \in \mathcal{B}} \pi(a) u(a, b) \left( \nu_\theta^{\pi,y}(b; \eta, \tau) - \nu_\theta^{\pi,y}(b; \widehat{\eta}, \widehat{\tau}) \right) \right|
$$
$$
\le \sum_{a \in \mathcal{A}} \pi(a) \sum_{b \in \mathcal{B}} |u(a, b)| \cdot |\nu_\theta^{\pi,y}(b; \eta, \tau) - \nu_\theta^{\pi,y}(b; \widehat{\eta}, \widehat{\tau})|
$$
$$
\le \left( \max_{a,b} |u(a, b)| \right) \cdot \sum_{a \in \mathcal{A}} \pi(a) \|\nu_\theta^{\pi,y}(\cdot; \eta, \tau) - \nu_\theta^{\pi,y}(\cdot; \widehat{\eta}, \widehat{\tau})\|_1
$$
$$
\le 1 \cdot 1 \cdot L_\nu\zeta = L_\nu\zeta.
$$

Here, we used the assumption that the leader's rewards are bounded in $[0, 1]$. This establishes a uniform bias bound for any policy pair.

**Step 3: Regret Decomposition.** Let $(\pi^*, y^*)$ be the true optimal policy pair that maximizes $J(\pi, y)$ under the true IC constraint, and let $(\pi_t, y_t)$ be the policy pair chosen by Algorithm 1 at episode $t$. The cumulative regret is defined as:

$$\mathrm{Reg}(T) = \sum_{t=1}^{T} \left( J(\pi^*, y^*) - J(\pi_t, y_t) \right).$$

We decompose the instantaneous regret at step $t$ by adding and subtracting the estimated objective $\widehat{J}$:

$$J(\pi^*, y^*) - J(\pi_t, y_t) = \underbrace{(J(\pi^*, y^*) - \widehat{J}(\pi^*, y^*))}_{\text{(i) Bias at optimal}} + \underbrace{(\widehat{J}(\pi^*, y^*) - \widehat{J}(\pi_t, y_t))}_{\text{(ii) Regret on estimated model}} + \underbrace{(\widehat{J}(\pi_t, y_t) - J(\pi_t, y_t))}_{\text{(iii) Bias at current}}.$$

Now we bound each term:

1. **Bias Terms:** From Step 2, we know that $|J(\pi, y) - \widehat{J}(\pi, y)| \leq L_\nu \zeta$ holds uniformly. Therefore, both (i) and (iii) are upper bounded by $L_\nu \zeta$.

2. **Regret on Estimated Model:** The sum of term (ii) $\sum_{t=1}^{T}(\widehat{J}(\pi^*, y^*) - \widehat{J}(\pi_t, y_t))$ represents the regret of running Algorithm 1 on the *misspecified* model $\widehat{J}$. Since Algorithm 1 is a no-regret algorithm for the model it optimizes, by Theorem 6.1, this term is bounded by $\widetilde{\mathcal{O}}(d^2\sqrt{T})$ with high probability.

Summing over $T$ steps, we obtain the bound for the cumulative regret:

$$\mathrm{Reg}(T) \leq \sum_{t=1}^{T} (L_\nu \zeta + L_\nu \zeta) + \widetilde{\mathcal{O}}(d^2\sqrt{T})$$

$$= 2L_\nu \zeta T + \widetilde{\mathcal{O}}(d^2\sqrt{T}) = \widetilde{\mathcal{O}}(d^2\sqrt{T}) + \widetilde{\mathcal{O}}(\zeta T).$$

**Step 4: ICV Regret Decomposition.** We now extend the robustness result to ICV regret. The IC violation at step $t$ is $\mathrm{ICV}_t = \mathrm{TV}(y_t, \nu_{\theta^*}^{\pi_t, y_t}(\eta, \tau))$. Let $\widehat{\mathrm{ICV}}_t = \mathrm{TV}(y_t, \nu_{\theta^*}^{\pi_t, y_t}(\widehat{\eta}, \widehat{\tau}))$ be the ICV estimated using the misspecified parameters. Using the triangle inequality for TV distance and the Lipschitz property from Step 1:

$$|\mathrm{ICV}_t - \widehat{\mathrm{ICV}}_t| \leq \mathrm{TV}(\nu_{\theta^*}^{\pi_t, y_t}(\eta, \tau), \nu_{\theta^*}^{\pi_t, y_t}(\widehat{\eta}, \widehat{\tau})) = \frac{1}{2}\|\nu(\eta, \tau) - \nu(\widehat{\eta}, \widehat{\tau})\|_1 \leq \frac{1}{2}L_\nu \zeta.$$

This implies $\mathrm{ICV}_t \leq \widehat{\mathrm{ICV}}_t + \mathcal{O}(\zeta)$. The cumulative ICV regret is:

$$\mathrm{Reg}^{\mathrm{ICV}}(T) = \sum_{t=1}^{T} \mathrm{ICV}_t \leq \sum_{t=1}^{T} \widehat{\mathrm{ICV}}_t + \sum_{t=1}^{T} \mathcal{O}(\zeta).$$

The first term $\sum \widehat{\mathrm{ICV}}_t$ is the ICV regret of the algorithm running on the estimated model parameters. By Theorem 6.1, this term is bounded by $\widetilde{\mathcal{O}}(d^2\sqrt{T})$ with high probability. The second term sums to $\mathcal{O}(\zeta T)$. Thus, we have:

$$\mathrm{Reg}^{\mathrm{ICV}}(T) \leq \widetilde{\mathcal{O}}(d^2\sqrt{T}) + \widetilde{\mathcal{O}}(\zeta T).$$

This completes the proof of Theorem 6.4. □

### F.5.2 PROOF OF COROLLARY 6.5

*Proof.* We extend the robustness result to absolute regret. Recall that absolute regret is defined as the sum of absolute suboptimalities, which accounts for potentially negative regret due to IC violations.

First, we observe that the ICV also degrades linearly with parameter error. The ICV is defined via the TV distance between the recommendation $y$ and the follower's response $\nu$. Let $\widehat{\mathrm{ICV}}$ be the estimated ICV using $\widehat{\eta}, \widehat{\tau}$. By the triangle inequality and the Lipschitz continuity established in Theorem 6.4 (Step 1):

$$|\mathrm{ICV}_t - \widehat{\mathrm{ICV}}_t| = |\mathrm{TV}(y_t, \nu(\eta)) - \mathrm{TV}(y_t, \nu(\widehat{\eta}))| \leq \mathrm{TV}(\nu(\eta), \nu(\widehat{\eta})) \leq \mathcal{O}(\zeta).$$

Since the algorithm controls the estimated ICV regret $\sum \widehat{\mathrm{ICV}}_t \leq \widetilde{\mathcal{O}}(\sqrt{T})$ (by Theorem 6.1), the true ICV regret is bounded by:

$$\mathrm{Reg}^{\mathrm{ICV}}(T) = \sum_{t=1}^{T} \mathrm{ICV}_t \leq \sum_{t=1}^{T} (\widehat{\mathrm{ICV}}_t + \mathcal{O}(\zeta)) \leq \widetilde{\mathcal{O}}(d^2\sqrt{T}) + \widetilde{\mathcal{O}}(\zeta T).$$

Next, we invoke Proposition C.6 from our main analysis, which provides the critical link between standard regret, ICV regret, and absolute regret. Specifically, it establishes that for any policy pair, the absolute suboptimality is bounded linearly by the standard suboptimality (if positive) and the IC violation magnitude. Summing over $T$, we have the general relationship:

$$\mathrm{Reg}^{\mathrm{abs}}(T) \leq \mathrm{Reg}(T) + C\,\mathrm{Reg}^{\mathrm{ICV}}(T),$$

where $C$ is a problem-dependent constant. Substituting the bounds for $\mathrm{Reg}(T)$ (from Theorem 6.4) and $\mathrm{Reg}^{\mathrm{ICV}}(T)$ (derived above), we obtain:

$$\mathrm{Reg}^{\mathrm{abs}}(T) \leq [\widetilde{\mathcal{O}}(d^2\sqrt{T}) + \widetilde{\mathcal{O}}(\zeta T)] + C[\widetilde{\mathcal{O}}(d^2\sqrt{T}) + \widetilde{\mathcal{O}}(\zeta T)]$$
$$= \widetilde{\mathcal{O}}(d^2\sqrt{T}) + \widetilde{\mathcal{O}}(\zeta T),$$

which completes the proof of Corollary 6.5. This confirms that the absolute regret also degrades gracefully with the parameter estimation error. $\qquad\square$

### F.5.3  PROOF OF COROLLARY 6.6

*Proof.* This result follows from a standard online-to-batch conversion technique, which converts a cumulative regret bound into a sample complexity guarantee for a randomized stationary policy.

Let the algorithm run for $T$ episodes. We define the output policy pair $(\widehat{\pi}, \widehat{y})$ as a policy uniformly sampled from the history of played policies $\{(\pi_t, y_t)\}_{t=1}^{T}$. The expected suboptimality of this randomized policy is equal to the average regret of the algorithm:

$$\mathbb{E}_{t \sim \mathrm{Unif}[T]}[\mathrm{SubOpt}(\pi_t, y_t)] = \frac{\mathrm{Reg}(T)}{T}.$$

Substituting the regret bound from Theorem 6.4:

$$\mathbb{E}[\mathrm{SubOpt}(\widehat{\pi}, \widehat{y})] \leq \frac{\widetilde{\mathcal{O}}(d^2\sqrt{T}) + \widetilde{\mathcal{O}}(\zeta T)}{T} = \frac{\widetilde{\mathcal{O}}(d^2)}{\sqrt{T}} + C_{\mathrm{bias}}\zeta,$$

where $C_{\mathrm{bias}}$ captures the constants in the linear term.

To achieve a target accuracy of $\epsilon + C_{\mathrm{bias}}\zeta$, where $\epsilon$ is the controllable statistical error and $C_{\mathrm{bias}}\zeta$ is the irreducible bias, we require the statistical term to be at most $\epsilon$:

$$\frac{\widetilde{\mathcal{O}}(d^2)}{\sqrt{T}} \leq \epsilon.$$

Solving for $T$, we get:

$$\sqrt{T} \geq \frac{\widetilde{\mathcal{O}}(d^2)}{\epsilon} \implies T \geq \widetilde{\mathcal{O}}\left(\frac{d^4}{\epsilon^2}\right).$$

This shows that the number of samples required is polynomial in the problem dimension $d$ and the inverse squared target accuracy $1/\epsilon^2$, establishing the polynomial sample complexity in Corollary 6.6. $\qquad\square$

## G  PROOFS OF THEORY IN APPENDIX C.4

### G.1  PROOF OF THEOREM C.1

*Proof.* We need to solve the equation $y = \nu_\theta^{\pi,y}$. By the expression of the follower's response model equation 5, we have for any $b \in \mathcal{B}$,

$$y(b) = \nu_\theta^{\pi,y}(b) = [y(b)]^{\frac{\tau}{\tau+\eta^{-1}}} \cdot \exp\left\{\frac{1}{\tau+\eta^{-1}} \cdot A_\theta^{\pi,y}(b)\right\}. \tag{54}$$

By dividing $[y(b)]^{\frac{\tau}{\tau+\eta^{-1}}} > 0$ in both sides, we have

$$[y(b)]^{\frac{\eta^{-1}}{\tau+\eta^{-1}}} = \exp\left\{ \frac{1}{\tau + \eta^{-1}} \cdot A_\theta^{\pi,y}(b) \right\}, \tag{55}$$

and hence

$$y(b) = \exp\left\{ \eta \cdot A_\theta^{\pi,y}(b) \right\} = \frac{\exp\left\{ \eta \cdot Q_\theta^\pi(b) \right\}}{\exp\left\{ \eta \cdot V_\theta^{\pi,y} \right\}}. \tag{56}$$

By letting $\tau = 0$ in equation 5, we get the quantal response that does not depend on $y$ as

$$\mu_\theta^\pi(b) = \frac{\exp\left\{ \eta \cdot Q_\theta^\pi(b) \right\}}{\exp\left\{ \eta \cdot V_\theta^\pi \right\}}. \tag{57}$$

Since both $y$ and $\mu_\theta^\pi$ are probability distributions over $\mathcal{B}$, we have

$$\sum_{b\in\mathcal{B}} y(b) = \sum_{b\in\mathcal{B}} \frac{\exp\left\{ \eta \cdot Q_\theta^\pi(b) \right\}}{\exp\left\{ \eta \cdot V_\theta^{\pi,y} \right\}} = \frac{\sum_{b\in\mathcal{B}} \exp\left\{ \eta \cdot Q_\theta^\pi(b) \right\}}{\exp\left\{ \eta \cdot V_\theta^{\pi,y} \right\}} = 1, \tag{58}$$

and

$$\sum_{b\in\mathcal{B}} \mu_\theta^\pi(b) = \sum_{b\in\mathcal{B}} \frac{\exp\left\{ \eta \cdot Q_\theta^\pi(b) \right\}}{\exp\left\{ \eta \cdot V_\theta^\pi \right\}} = \frac{\sum_{b\in\mathcal{B}} \exp\left\{ \eta \cdot Q_\theta^\pi(b) \right\}}{\exp\left\{ \eta \cdot V_\theta^\pi \right\}} = 1. \tag{59}$$

Hence, we have

$$\exp\left\{ \eta \cdot V_\theta^\pi \right\} = \sum_{b\in\mathcal{B}} \exp\left\{ \eta \cdot Q_\theta^\pi(b) \right\} = \exp\left\{ \eta \cdot V_\theta^{\pi,y} \right\}. \tag{60}$$

Since $\exp : \mathbb{R} \to \mathbb{R}$ is a one-to-one mapping and the parameter $\eta > 0$, the last equality implies

$$V_\theta^\pi = V_\theta^{\pi,y}. \tag{61}$$

Therefore, we have that for any $b \in \mathcal{B}$,

$$y(b) = \frac{\exp\left\{ \eta \cdot Q_\theta^\pi(b) \right\}}{\exp\left\{ \eta \cdot V_\theta^{\pi,y} \right\}} = \frac{\exp\left\{ \eta \cdot Q_\theta^\pi(b) \right\}}{\exp\left\{ \eta \cdot V_\theta^\pi \right\}} = \mu_\theta^\pi(b), \tag{62}$$

which concludes that $y = \mu_\theta^\pi$ is the unique solution to the equation $y = \nu_\theta^{\pi,y}$. Therefore, we can conclude that $y = \mu_\theta^\pi$ if and only if $y = \nu_\theta^{\pi,y}$. We can also write $\mu_\theta^\pi = \nu_\theta^{\pi,\mu_\theta^\pi}$. Last, by plugging the solution $y = \mu_\theta^\pi$ into equation 61, we immediately obtain

$$V_\theta^\pi = V_\theta^{\pi,\mu_\theta^\pi}, \tag{63}$$

which completes the proof. $\qquad\square$

## G.2 Proof of Theorem C.2

*Proof.* Recall that $\text{SubOpt}_R(\pi, y; \theta)$ is the suboptimality of recommendation defined in equation 21 as follows:

$$\text{SubOpt}_R(\pi, y; \theta) := \max_{\nu\in\Delta(\mathcal{B})} G(\nu; \pi, y; \theta) - G(y; \pi, y; \theta) = G(\nu_\theta^{\pi,y}; \pi, y; \theta) - G(y; \pi, y; \theta).$$

Suppose $\text{SubOpt}_R(\pi, y; \theta) = 0$, then by definition, $y$ and $\nu_\theta^{\pi,y}$ simultaneously maximize $G(\cdot; \pi, y; \theta)$. By the analysis in Section 3.2, thanks to the regularization in equation 3, the quantal response policy $\nu_\theta^{\pi,y}$ is unique for any policies $\pi$ and $y$ given parameter $\theta$. Since $\pi^\top B_\theta \nu$ is linear in $\nu$, $-\tau D(\nu\|y)$ and $\eta^{-1}\mathcal{H}(\nu)$ are both strictly concave in $\nu$, $G(\nu; \pi, y; \theta)$ defined in equation 3 as the summation of these three terms, is also strictly concave in $\nu$. By the classic result that a strictly concave function on a convex set has at most one maximize, equation 3 has a unique solution $\nu_\theta^{\pi,y} \in \Delta(\mathcal{B})$. This means that the maximizer of $G(\cdot; \pi, y; \theta)$ is unique, and hence we can immediately have that $\nu_\theta^{\pi,y} = y$. It is also evident that $\nu_\theta^{\pi,y} = y$ directly implies $\text{SubOpt}_R(\pi, y; \theta) = 0$. Therefore, $\text{SubOpt}_R(\pi, y; \theta) = 0$ is equivalent to $\nu_\theta^{\pi,y} = y$, and then is equivalent to $\text{ICV}(\pi, y; \theta) = 0$ by the definition in equation 8. $\quad\square$

### G.3 PROOF OF THEOREM C.3

*Proof.* Recall that the quantal response defined in equation 6 can be written as

$$\mu_\theta^\pi(b) = \frac{\exp\{\eta \cdot Q_\theta^\pi(b)\}}{\sum_{b' \in \mathcal{B}} \exp\{\eta \cdot Q_\theta^\pi(b')\}} \tag{64}$$

for any $b \in \mathcal{B}$. Dividing both of the numerator and the denominator of equation 64 $\exp\{\eta \cdot Q_\theta^\pi(b)\} > 0$, we have

$$\mu_\theta^\pi(b) = \left(\sum_{b' \in \mathcal{B}} \exp\{\eta \cdot (Q_\theta^\pi(b') - Q_\theta^\pi(b))\}\right)^{-1}. \tag{65}$$

Note that $Q_\theta(a, b) \in [0, 1]$ for any $(a, b) \in \mathcal{A} \times \mathcal{B}$ by assumption, we have $Q_\theta^\pi(b) \in [0, 1]$ for any $b \in \mathcal{B}$ and $\pi \in \Pi$. So we have $Q_\theta^\pi(b') - Q_\theta^\pi(b) \leq 1$ for any $b, b' \in \mathcal{B}$. Then, we have

$$0 < \sum_{b' \in \mathcal{B}} \exp\{\eta \cdot (Q_\theta^\pi(b') - Q_\theta^\pi(b))\} \leq \sum_{b' \in \mathcal{B}} \exp(\eta) = |\mathcal{B}| \cdot \exp(\eta). \tag{66}$$

Therefore, combining equation 65 and equation 66, we obtain

$$\mu_\theta^\pi(b) \geq |\mathcal{B}|^{-1} \cdot \exp(-\eta) =: \mu_{\mathrm{L}} > 0,$$

which completes the proof. $\square$

### G.4 PROOF OF THEOREM C.4

*Proof.* Recall that the follower's response defined in equation 5 can be written as

$$\nu_\theta^{\pi,y}(b) = \frac{[y(b)]^{\frac{\tau}{\tau+\eta^{-1}}} \cdot \exp\left\{\frac{1}{\tau+\eta^{-1}} \cdot Q_\theta^\pi(b)\right\}}{\sum_{b' \in \mathcal{B}} [y(b')]^{\frac{\tau}{\tau+\eta^{-1}}} \cdot \exp\left\{\frac{1}{\tau+\eta^{-1}} \cdot Q_\theta^\pi(b')\right\}}, \tag{67}$$

for any $b \in \mathcal{B}$. Dividing both of the numerator and the denominator of equation 67 $[y(b)]^{\frac{\tau}{\tau+\eta^{-1}}} \cdot \exp\left\{\frac{1}{\tau+\eta^{-1}} \cdot Q_\theta^\pi(b)\right\} > 0$, we have

$$\nu_\theta^{\pi,y}(b) = \left(\sum_{b' \in \mathcal{B}} \left[\frac{y(b')}{y(b)}\right]^{\frac{\tau}{\tau+\eta^{-1}}} \cdot \exp\left\{\frac{1}{\tau+\eta^{-1}} \cdot (Q_\theta^\pi(b') - Q_\theta^\pi(b))\right\}\right)^{-1}. \tag{68}$$

Note that $Q_\theta(a, b) \in [0, 1]$ for any $(a, b) \in \mathcal{A} \times \mathcal{B}$ by assumption, we have $Q_\theta^\pi(b) \in [0, 1]$ for any $b \in \mathcal{B}$ and $\pi \in \Pi$. So we have $Q_\theta^\pi(b') - Q_\theta^\pi(b) \leq 1$ for any $b, b' \in \mathcal{B}$. Also, since $y \in \mathcal{Y}_{\mathrm{QR}}$, by equation 24 in Theorem C.3, we have

$$1 \geq y(b) \geq \mu_{\mathrm{L}} := |\mathcal{B}|^{-1} \cdot \exp(-\eta) > 0 \tag{69}$$

for any $b \in \mathcal{B}$. So we have $0 < \frac{y(b')}{y(b)} \leq |\mathcal{B}| \cdot \exp(\eta)$. Then, we have

$$0 < \sum_{b' \in \mathcal{B}} \left[\frac{y(b')}{y(b)}\right]^{\frac{\tau}{\tau+\eta^{-1}}} \cdot \exp\left\{\frac{1}{\tau+\eta^{-1}} \cdot (Q_\theta^\pi(b') - Q_\theta^\pi(b))\right\}$$

$$\leq \sum_{b' \in \mathcal{B}} [|\mathcal{B}| \cdot \exp(\eta)]^{\frac{\tau}{\tau+\eta^{-1}}} \cdot \exp\left\{\frac{1}{\tau+\eta^{-1}}\right\} = |\mathcal{B}|^{\frac{2\tau\eta+1}{\tau\eta+1}} \cdot \exp(\eta), \tag{70}$$

where the second inequality is by equation 69. Therefore, combining equation 68 and equation 70, we obtain

$$\nu_\theta^{\pi,y}(b) \geq |\mathcal{B}|^{-\frac{2\tau\eta+1}{\tau\eta+1}} \cdot \exp(-\eta) =: \nu_{\mathrm{L}} > 0,$$

which completes the proof. $\square$

### G.5 PROOF OF THEOREM C.5

*Proof.* Suppose $\sigma \in [0, \frac{1}{2}\nu_{\mathrm{L}}]$ and $\|y - \nu_\theta^{\pi,y}\|_1 \leq \sigma$. By the definition of the $\ell_1$-norm, we have that, for all $b \in \mathcal{B}$, $|y(b) - \nu_\theta^{\pi,y}(b)| \leq \sigma$, and so

$$\nu_\theta^{\pi,y}(b) - \sigma \leq y(b) \leq \nu_\theta^{\pi,y}(b) + \sigma. \tag{71}$$

Recall the formula for the follower's response in equation 5,

$$\nu_\theta^{\pi,y}(b) = [y(b)]^{\frac{\tau}{\tau+\eta^{-1}}} \cdot \exp\left\{\frac{1}{\tau + \eta^{-1}} \cdot A_\theta^{\pi,y}(b)\right\}. \tag{72}$$

For the second inequality in equation 71, we have by simple algebra that

$$[y(b)]^{\frac{\eta^{-1}}{\tau+\eta^{-1}}} \leq \exp\left\{\frac{1}{\tau + \eta^{-1}} \cdot A_\theta^{\pi,y}(b)\right\} + \frac{\sigma}{[y(b)]^{\frac{\tau}{\tau+\eta^{-1}}}}$$

$$= \left(1 + \frac{\sigma}{[y(b)]^{\frac{\tau}{\tau+\eta^{-1}}} \cdot \exp\left\{\frac{1}{\tau+\eta^{-1}} \cdot A_\theta^{\pi,y}(b)\right\}}\right) \cdot \exp\left\{\frac{1}{\tau + \eta^{-1}} \cdot A_\theta^{\pi,y}(b)\right\}$$

$$= \left(1 + \frac{\sigma}{\nu_\theta^{\pi,y}(b)}\right) \cdot \exp\left\{\frac{1}{\tau + \eta^{-1}} \cdot A_\theta^{\pi,y}(b)\right\}. \tag{73}$$

Then, by simple algebra we obtain

$$y(b) \leq \left(1 + \frac{\sigma}{\nu_\theta^{\pi,y}(b)}\right)^{1+\tau\eta} \cdot \exp\left\{\eta \cdot A_\theta^{\pi,y}(b)\right\} \leq \left(1 + \frac{\sigma}{\nu_{\mathrm{L}}}\right)^{1+\tau\eta} \cdot \exp\left\{\eta \cdot A_\theta^{\pi,y}(b)\right\}, \tag{74}$$

where the second inequality is because $\nu_\theta^{\pi,y}(b) \geq \nu_{\mathrm{L}} > 0$ by equation 26 in Theorem C.4. Similarly, for the first inequality in equation 71, we have that

$$y(b) \geq \left(1 - \frac{\sigma}{\nu_\theta^{\pi,y}(b)}\right)^{1+\tau\eta} \cdot \exp\left\{\eta \cdot A_\theta^{\pi,y}(b)\right\} \geq \left(1 - \frac{\sigma}{\nu_{\mathrm{L}}}\right)^{1+\tau\eta} \cdot \exp\left\{\eta \cdot A_\theta^{\pi,y}(b)\right\}. \tag{75}$$

Taking summation over $b \in \mathcal{B}$ for the above two inequalities, we have

$$1 = \sum_{b \in \mathcal{B}} y(b) \leq \left(1 + \frac{\sigma}{\nu_{\mathrm{L}}}\right)^{1+\tau\eta} \cdot \sum_{b \in \mathcal{B}} \exp\left\{\eta \cdot A_\theta^{\pi,y}(b)\right\}, \tag{76}$$

and

$$1 = \sum_{b \in \mathcal{B}} y(b) \geq \left(1 - \frac{\sigma}{\nu_{\mathrm{L}}}\right)^{1+\tau\eta} \cdot \sum_{b \in \mathcal{B}} \exp\left\{\eta \cdot A_\theta^{\pi,y}(b)\right\}, \tag{77}$$

respectively. Since $A_\theta^{\pi,y}(b) = Q_\theta^\pi(b) - V_\theta^{\pi,y}$, we further obtain

$$\exp\left\{\eta \cdot V_\theta^{\pi,y}\right\} \leq \left(1 + \frac{\sigma}{\nu_{\mathrm{L}}}\right)^{1+\tau\eta} \cdot \sum_{b \in \mathcal{B}} \exp\left\{\eta \cdot Q_\theta^\pi(b)\right\} = \left(1 + \frac{\sigma}{\nu_{\mathrm{L}}}\right)^{1+\tau\eta} \cdot \exp\left\{\eta \cdot V_\theta^\pi\right\}, \tag{78}$$

and

$$\exp\left\{\eta \cdot V_\theta^{\pi,y}\right\} \geq \left(1 - \frac{\sigma}{\nu_{\mathrm{L}}}\right)^{1+\tau\eta} \cdot \sum_{b \in \mathcal{B}} \exp\left\{\eta \cdot Q_\theta^\pi(b)\right\} = \left(1 - \frac{\sigma}{\nu_{\mathrm{L}}}\right)^{1+\tau\eta} \cdot \exp\left\{\eta \cdot V_\theta^\pi\right\}, \tag{79}$$

respectively. Here, $V_\theta^{\pi,y}$ is defined in equation 4, and $V_\theta^\pi$ is equal to $V_\theta^{\pi,y}$ but is taken with $\tau = 0$.

Next, by equation 74, we have

$$y(b) \leq \left(1 + \frac{\sigma}{\nu_{\mathrm{L}}}\right)^{1+\tau\eta} \cdot \frac{\exp\left\{\eta \cdot Q_\theta^\pi(b)\right\}}{\exp\left\{\eta \cdot V_\theta^{\pi,y}\right\}}$$

$$\leq \frac{\left(1 + \frac{\sigma}{\nu_{\mathrm{L}}}\right)^{1+\tau\eta}}{\left(1 - \frac{\sigma}{\nu_{\mathrm{L}}}\right)^{1+\tau\eta}} \cdot \frac{\exp\left\{\eta \cdot Q_\theta^\pi(b)\right\}}{\exp\left\{\eta \cdot V_\theta^\pi\right\}} = \left(\frac{\nu_{\mathrm{L}} + \sigma}{\nu_{\mathrm{L}} - \sigma}\right)^{1+\tau\eta} \cdot \mu_\theta^\pi(b), \tag{80}$$

where the second inequality is by equation 79. Similarly, by equation 75, we have

$$
y(b) \geq \left(1 - \frac{\sigma}{\nu_{\mathrm{L}}}\right)^{1+\tau\eta} \cdot \frac{\exp\{\eta \cdot Q_\theta^\pi(b)\}}{\exp\{\eta \cdot V_\theta^{\pi,y}\}}
$$

$$
\geq \frac{\left(1 - \frac{\sigma}{\nu_{\mathrm{L}}}\right)^{1+\tau\eta}}{\left(1 + \frac{\sigma}{\nu_{\mathrm{L}}}\right)^{1+\tau\eta}} \cdot \frac{\exp\{\eta \cdot Q_\theta^\pi(b)\}}{\exp\{\eta \cdot V_\theta^\pi\}} = \left(\frac{\nu_{\mathrm{L}} - \sigma}{\nu_{\mathrm{L}} + \sigma}\right)^{1+\tau\eta} \cdot \mu_\theta^\pi(b), \tag{81}
$$

where the second inequality is by equation 78. Hence, we have

$$
\left[\left(\frac{\nu_{\mathrm{L}} - \sigma}{\nu_{\mathrm{L}} + \sigma}\right)^{1+\tau\eta} - 1\right] \cdot \mu_\theta^\pi(b) \leq y(b) - \mu_\theta^\pi(b) \leq \left[\left(\frac{\nu_{\mathrm{L}} + \sigma}{\nu_{\mathrm{L}} - \sigma}\right)^{1+\tau\eta} - 1\right] \cdot \mu_\theta^\pi(b), \tag{82}
$$

and consequently,

$$
|y(b) - \mu_\theta^\pi(b)| \leq \max\left\{\left|\left(\frac{\nu_{\mathrm{L}} - \sigma}{\nu_{\mathrm{L}} + \sigma}\right)^{1+\tau\eta} - 1\right|, \left|\left(\frac{\nu_{\mathrm{L}} + \sigma}{\nu_{\mathrm{L}} - \sigma}\right)^{1+\tau\eta} - 1\right|\right\} \cdot \mu_\theta^\pi(b), \tag{83}
$$

for all $b \in \mathcal{B}$.

So we have

$$
\|y - \mu_\theta^\pi\|_1 = \sum_{b \in \mathcal{B}} |y(b) - \mu_\theta^\pi(b)| \leq \max\left\{\left|\left(\frac{\nu_{\mathrm{L}} - \sigma}{\nu_{\mathrm{L}} + \sigma}\right)^{1+\tau\eta} - 1\right|, \left|\left(\frac{\nu_{\mathrm{L}} + \sigma}{\nu_{\mathrm{L}} - \sigma}\right)^{1+\tau\eta} - 1\right|\right\} \cdot \sum_{b \in \mathcal{B}} \mu_\theta^\pi(b)
$$

$$
= \max\left\{\left|\left(\frac{\nu_{\mathrm{L}} - \sigma}{\nu_{\mathrm{L}} + \sigma}\right)^{1+\tau\eta} - 1\right|, \left|\left(\frac{\nu_{\mathrm{L}} + \sigma}{\nu_{\mathrm{L}} - \sigma}\right)^{1+\tau\eta} - 1\right|\right\}
$$

$$
= \max\left\{1 - \left(\frac{\nu_{\mathrm{L}} - \sigma}{\nu_{\mathrm{L}} + \sigma}\right)^{1+\tau\eta}, \left(\frac{\nu_{\mathrm{L}} + \sigma}{\nu_{\mathrm{L}} - \sigma}\right)^{1+\tau\eta} - 1\right\}, \tag{84}
$$

where the last equality is by noticing that $0 < \frac{\nu_{\mathrm{L}} - \sigma}{\nu_{\mathrm{L}} + \sigma} \leq 1 \leq \frac{\nu_{\mathrm{L}} + \sigma}{\nu_{\mathrm{L}} - \sigma}$.

By calculus, we have that for any $\sigma \in [0, \frac{1}{2}\nu_{\mathrm{L}}]$,

$$
1 - \left(\frac{\nu_{\mathrm{L}} - \sigma}{\nu_{\mathrm{L}} + \sigma}\right)^{1+\tau\eta} \leq \frac{2(1+\tau\eta)}{\nu_{\mathrm{L}}} \cdot \sigma \tag{85}
$$

and

$$
\left(\frac{\nu_{\mathrm{L}} + \sigma}{\nu_{\mathrm{L}} - \sigma}\right)^{1+\tau\eta} - 1 \leq \frac{4 \cdot 3^{\tau\eta}(1+\tau\eta)}{\nu_{\mathrm{L}}} \cdot \sigma. \tag{86}
$$

Last, by the fact that $2 < 4 \cdot 3^{\tau\eta}$, we have

$$
\|y - \mu_\theta^\pi\|_1 \leq \frac{4 \cdot 3^{\tau\eta}(1+\tau\eta)}{\nu_{\mathrm{L}}} \cdot \sigma, \tag{87}
$$

which completes the proof. $\qquad\square$

### G.6   PROOF OF THEOREM C.6

*Proof.* Since the policy pair $(\pi, y)$ satisfies $\mathrm{TV}(y, \nu_\theta^{\pi,y}) \leq \sigma$, which is equivalently to $\|y - \nu_\theta^{\pi,y}\|_1 \leq 2\sigma$, where $\sigma \in [0, \frac{1}{4}\nu_{\mathrm{L}}]$, we have by equation 27 in Theorem C.5 that,

$$
\mathrm{TV}(y, \mu_\theta^\pi) = \frac{1}{2}\|y - \mu_\theta^\pi\|_1 \leq \frac{4 \cdot 3^{\tau\eta}(1+\tau\eta)}{\nu_{\mathrm{L}}} \cdot \sigma. \tag{88}
$$

Hence, we have

$$
\pi^\top A \nu_\theta^{\pi,y} - \pi^\top A \mu_\theta^\pi = \pi^\top A \nu_\theta^{\pi,y} - \pi^\top A y + \pi^\top A y - \pi^\top A \mu_\theta^\pi
$$

$$
= \pi^\top A(\nu_\theta^{\pi,y} - y) + \pi^\top A(y - \mu_\theta^\pi) \leq \|y - \nu_\theta^{\pi,y}\|_1 + \|y - \mu_\theta^\pi\|_1
$$

$$
\leq \left(2 + \frac{8 \cdot 3^{\tau\eta}(1+\tau\eta)}{\nu_{\mathrm{L}}}\right) \cdot \sigma, \tag{89}
$$

where the first inequality is by the boundness of the leader's reward function and the Holder's inequality, and the second inequality is by equation 88. Therefore, we have

$$\pi^\top A\nu_\theta^{\pi,y} - F_\theta(0) = \pi^\top A\nu_\theta^{\pi,y} - \max_{\text{TV}(y,\nu_\theta^{\pi,y})=0} \pi^\top A\nu_\theta^{\pi,y} = \pi^\top A\nu_\theta^{\pi,y} - \max_{\pi'\in\Pi} \pi'^\top A\mu_\theta^{\pi'}$$

$$\leq \pi^\top A\nu_\theta^{\pi,y} - \pi^\top A\mu_\theta^\pi \leq \left(2 + \frac{8\cdot 3^{\tau\eta}(1+\tau\eta)}{\nu_{\text{L}}}\right)\cdot\sigma, \tag{90}$$

where the second equality is because $\text{TV}(y,\nu_\theta^{\pi,y}) = 0$ implies $\nu_\theta^{\pi,y} = y = \mu_\theta^\pi$, as shown by Theorem C.1.

Since the last inequality holds for any policy pair $(\pi,y)$ such that $\text{TV}(y,\nu_\theta^{\pi,y}) \leq \sigma$, we can take a maximum over $(\pi,y)$ and obtain

$$F_\theta(\sigma) - F_\theta(0) = \max_{\text{TV}(y,\nu_\theta^{\pi,y})\leq\sigma} \pi^\top A\nu_\theta^{\pi,y} - F_\theta(0) \leq \left(2 + \frac{8\cdot 3^{\tau\eta}(1+\tau\eta)}{\nu_{\text{L}}}\right)\cdot\sigma, \tag{91}$$

which completes the proof. $\qquad\square$

# H  PROOFS OF AUXILIARY LEMMAS IN APPENDIX F.2

## H.1  PROOF OF THEOREM F.1 (LEADER'S REGRET FOR SCHEME 1)

*Proof.* First of all, we decompose the suboptimality $\text{SubOpt}(\pi_t, y_t)$ at time $t$ into three terms in the following way.

$$\text{SubOpt}(\pi_t, y_t) := \langle U, \pi^* \otimes \nu^* \rangle - \langle U, \pi_t \otimes \nu_{\theta^*}^{\pi_t,y_t} \rangle$$
$$= \underbrace{\langle U - \widetilde{U}_t, \pi^* \otimes \nu^* \rangle}_{=:S_t^1} + \underbrace{\langle \widetilde{U}_t - U, \pi_t \otimes \nu_{\theta^*}^{\pi_t,y_t} \rangle}_{=:S_t^2} + \underbrace{\langle \widetilde{U}_t, \pi^* \otimes \nu^* - \pi_t \otimes \nu_{\theta^*}^{\pi_t,y_t} \rangle}_{=:S_t^3}. \tag{92}$$

Next, we will establish upper bounds on the three terms $S_t^1$, $S_t^2$ and $S_t^3$ in equation 92 with high probability, respectively.

Before bounding $S_t^1$ and $S_t^2$, we first establish a high probability upper bound on the estimation error of the leader's reward function. In the construction of the estimated reward function $\widetilde{U}_t$ as in equation 14, the bonus term $\Gamma_t^1$ defined in Appendix 5.1 serves as an uncertainty quantifier. Under the condition of Theorem 6.1, we instantiate the coefficient $\alpha$ with $\alpha = C_1 d\sqrt{\log(6dT^2/p)}$ in the bonus term $\Gamma_t^1$, where $d = mn = |\mathcal{A}||\mathcal{B}|$ and $C_1$ is a universal constant. By Lemma 5.2 in Jin et al. (2021c) and Chen et al. (2023), the data compliance condition is satisfied in the online learning process, and hence when , we have with probability at least $1 - p/3$ that, for any $t \in [T]$,

$$|\widehat{U}_t(a,b) - U(a,b)| \leq \Gamma_t^1(a,b), \quad \text{for any } (a,b) \in \mathcal{A}\times\mathcal{B}, \tag{93}$$

where $\widehat{U}_t(a,b) = \phi(a,b)^\top\widehat{\omega}_t$. Then, by the construction of $\widetilde{U}_t$, we have that with probability at least $1 - p/3$, for any $t \in [T]$,

$$0 \leq \widetilde{U}_t(a,b) - U(a,b) \leq 2\Gamma_t^1(a,b), \quad \text{for any } (a,b) \in \mathcal{A}\times\mathcal{B}. \tag{94}$$

**Bounding $S_t^1$.** For $S_t^1 = \langle U - \widetilde{U}_t, \pi^* \otimes \nu^* \rangle$ in equation 92, by optimism in equation 94, we have that, with probability at least $1 - p/3$, for any $t \in [T]$,

$$U(a,b) - \widetilde{U}_t(a,b) \leq 0, \quad \forall(a,b) \in \mathcal{A}\times\mathcal{B}. \tag{95}$$

So we have by optimism that, with probability at least $1 - p/3$, for any $t \in [T]$,

$$S_t^1 \leq 0. \tag{96}$$

**Bounding $S_t^2$.** For $S_t^2 = \langle \widetilde{U}_t - U, \pi_t \otimes \nu_{\theta^*}^{\pi_t,y_t} \rangle$ in equation 92, by optimism in equation 94, we have that, with probability at least $1 - \frac{p}{3}$, for any $t \in [T]$,

$$\widetilde{U}_t(a,b) - U(a,b) \leq 2\Gamma_t^1(a,b), \quad \forall(a,b) \in \mathcal{A}\times\mathcal{B}, \tag{97}$$

So we have

$$S_t^2 \leq \langle 2\Gamma_t^1, \pi_t \otimes \nu_{\theta^*}^{\pi_t, y_t} \rangle = 2\Gamma_t^1(a_t, b_t) + \xi_t, \tag{98}$$

where

$$\xi_t := \langle 2\Gamma_t^1, \pi_t \otimes \nu_{\theta^*}^{\pi_t, y_t} \rangle - 2\Gamma_t^1(a_t, b_t). \tag{99}$$

Notice that $\{\xi_t\}$ is a martingale difference sequence satisfying $|\xi_t| \leq 2$ for all $t \in \mathbb{N}$. Therefore, by applying the Azuma-Hoeffding inequality, for any $\varepsilon > 0$, we have

$$\mathbb{P}\left(\sum_{t=1}^T \xi_t \geq \varepsilon\right) \leq \exp\left\{\frac{-2\varepsilon^2}{4T}\right\}. \tag{100}$$

Hence, with probability at least $1 - \frac{p}{3}$, we have

$$\sum_{t=1}^T \xi_t \leq \sqrt{2T \log\left(\frac{3}{p}\right)}. \tag{101}$$

This further implies that, with probability at least $1 - \frac{p}{3}$,

$$\sum_{t=1}^T S_t^2 \leq 2 \sum_{t=1}^T \Gamma_t^1(a_t, b_t) + \sqrt{2T \log\left(\frac{3}{p}\right)}. \tag{102}$$

**Bounding $S_t^3$.** For $S_t^3 = \langle \widetilde{U}_t, \pi^* \otimes \nu^* - \pi_t \otimes \nu_{\theta^*}^{\pi_t, y_t} \rangle$ in equation 92, we can further decompose it by

$$S_t^3 = \langle \widetilde{U}_t, \pi^* \otimes \nu^* \rangle - \langle \widetilde{U}_t, \pi_t \otimes \nu_{\theta^*}^{\pi_t, y_t} \rangle$$

$$= \underbrace{\langle \widetilde{U}_t, \pi^* \otimes \nu^* \rangle - \langle \widetilde{U}_t, \pi_t \otimes \nu_{\widehat{\theta}_t}^{\pi_t, y_t} \rangle}_{=:S_t^{3,1}} + \underbrace{\langle \widetilde{U}_t, \pi_t \otimes \nu_{\widehat{\theta}_t}^{\pi_t, y_t} \rangle - \langle \widetilde{U}_t, \pi_t \otimes \nu_{\theta^*}^{\pi_t, y_t} \rangle}_{=:S_t^{3,2}}. \tag{103}$$

For $S_t^{3,1}$ in equation 103, we have

$$S_t^{3,1} = \langle \widetilde{U}_t, \pi^* \otimes \nu_{\theta^*}^{\pi^*, y^*} \rangle - \langle \widetilde{U}_t, \pi_t \otimes \nu_{\widehat{\theta}_t}^{\pi_t, y_t} \rangle = \langle \widetilde{U}_t, \pi^* \otimes \mu_{\theta^*}^{\pi^*} \rangle - \langle \widetilde{U}_t, \pi_t \otimes \nu_{\widehat{\theta}_t}^{\pi_t, y_t} \rangle$$

$$\leq \max_{\theta \in \mathcal{C}_\Theta^t(\beta), \pi \in \Pi} \langle \widetilde{U}_t, \pi \otimes \mu_\theta^\pi \rangle - \langle \widetilde{U}_t, \pi_t \otimes \nu_{\widehat{\theta}_t}^{\pi_t, y_t} \rangle$$

$$= \langle \widetilde{U}_t, \pi_t \otimes \mu_{\widehat{\theta}_t}^{\pi_t} \rangle - \langle \widetilde{U}_t, \pi_t \otimes \nu_{\widehat{\theta}_t}^{\pi_t, y_t} \rangle = \langle \widetilde{U}_t, \pi_t \otimes \nu_{\widehat{\theta}_t}^{\pi_t, y_t} \rangle - \langle \widetilde{U}_t, \pi_t \otimes \nu_{\widehat{\theta}_t}^{\pi_t, y_t} \rangle = 0. \tag{104}$$

Here, the second equality is by the definition of $\nu^* := \nu_{\theta^*}^{\pi^*, y^*}$ and the third equality is by the property of the equilibrium that $y^* = \mu_{\theta^*}^{\pi^*} = \nu_{\theta^*}^{\pi^*, y^*}$. Then, the first inequality can be guaranteed by the event that $\theta^* \in \mathcal{C}_\Theta^t(\beta)$. The fourth equality is by the optimality of $\widehat{\theta}_t$ and $\pi_t$ in Scheme 1 equation 17. Last, the fifth equality is by the construction that $y_t = \mu_{\widehat{\theta}_t}^{\pi_t}$ as described in Scheme 1 equation 17, and $\mu_{\widehat{\theta}_t}^{\pi_t} = \nu_{\widehat{\theta}_t}^{\pi_t, y_t}$ as a result of Theorem C.1.

For $S_t^{3,2}$ in equation 103, we have

$$S_t^{3,2} = \langle \widetilde{U}_t, \pi_t \otimes (\nu_{\widehat{\theta}_t}^{\pi_t, y_t} - \nu_{\theta^*}^{\pi_t, y_t}) \rangle \leq \langle \mathbf{1}(\cdot, \cdot), \pi_t \otimes |\nu_{\widehat{\theta}_t}^{\pi_t, y_t} - \nu_{\theta^*}^{\pi_t, y_t}| \rangle$$

$$= \sum_{b \in \mathcal{B}} \sum_{a \in \mathcal{A}} \pi_t(a|b) |\nu_{\widehat{\theta}_t}^{\pi_t, y_t}(b) - \nu_{\theta^*}^{\pi_t, y_t}(b)| = \sum_{b \in \mathcal{B}} |\nu_{\widehat{\theta}_t}^{\pi_t, y_t}(b) - \nu_{\theta^*}^{\pi_t, y_t}(b)| \sum_{a \in \mathcal{A}} \pi_t(a|b)$$

$$= \sum_{b \in \mathcal{B}} |\nu_{\widehat{\theta}_t}^{\pi_t, y_t}(b) - \nu_{\theta^*}^{\pi_t, y_t}(b)| = \|\nu_{\widehat{\theta}_t}^{\pi_t, y_t} - \nu_{\theta^*}^{\pi_t, y_t}\|_1 = 2 \cdot \mathrm{TV}(\nu_{\widehat{\theta}_t}^{\pi_t, y_t}, \nu_{\theta^*}^{\pi_t, y_t}), \tag{105}$$

where we define $\mathbf{1}(\mathbf{a}, \mathbf{b}) = 1$ for any $a \in \mathcal{A}$ and $b \in \mathcal{B}$. Here the first inequality is by the boundedness of $\widetilde{U}_t$ within $[0, 1]$. The fourth equality is because $\nu_{\widehat{\theta}_t}^{\pi_t, y_t}(b)$ and $\nu_{\theta^*}^{\pi_t, y_t}(b)$ do not depend on $a \in \mathcal{A}$. The fifth equality is because $\pi_t(\cdot|b)$ is a probability distribution over $\mathcal{A}$ given any fixed action $b \in \mathcal{B}$. The last equality is by the relationship between the TV distance and the $\ell_1$-norm.

Therefore, by combining equation 103, equation 104 and equation 105, we have

$$S_t^3 = S_t^{3,1} + S_t^{3,2} \le 2 \cdot \mathrm{TV}(\nu_{\widehat{\theta}_t}^{\pi_t, y_t}, \nu_{\theta^*}^{\pi_t, y_t}), \tag{106}$$

conditioned on the event that $\theta^* \in \mathcal{C}_\Theta^t(\beta)$. We just need to guarantee that $\theta^* \in \mathcal{C}_\Theta^t(\beta)$ holds with high probability, which has already been shown by equation 144 in Theorem I.1.

Next, we turn to control $\mathrm{TV}(\nu_{\widehat{\theta}_t}^{\pi_t, y_t}, \nu_{\theta^*}^{\pi_t, y_t})$. Theorem J.3 provides the guarantee that

$$\mathrm{TV}(\nu_{\theta^*}^{\pi_t, y_t}, \nu_{\widehat{\theta}_t}^{\pi_t, y_t}) \le f_\varepsilon \left( \sqrt{\mathrm{Tr}\left( \left( \Sigma_t^{\theta^*} + I_d \right)^\dagger \Sigma^{\pi_t, y_t, \theta^*} \right)} \cdot \left\| \widehat{\theta}_t - \theta^* \right\|_{\Sigma_t^{\theta^*} + I_d} \right), \tag{107}$$

where the univariate function $f_\varepsilon$ is defined as $f_\varepsilon(x) = \varepsilon x + C_\varepsilon^{(3)} x^2$ with

$$C_\varepsilon^{(3)} = \varepsilon^2 \exp(2\varepsilon B_A)\left( 2 + \varepsilon B_A \cdot \exp\left( 2\varepsilon B_A \right) \right)/2. \tag{108}$$

We then need to establish an upper bound on $\left\| \widehat{\theta}_t - \theta \right\|_{\Sigma_t^\theta + I_d}$ that hopefully does not depend on $t$, where $\Sigma_t^\theta := \sum_{i=1}^{t-1} \mathrm{Cov}^{\pi_i, y_i, \theta}[\phi^{\pi_i}(b)]$ is the sum of covariance matrices. It suffices to focus on $\left\| \widehat{\theta}_t - \theta \right\|_{\Sigma_t^\theta}$. By equation 149 as the result of Theorem I.1, we have that, with probability at least $1 - \delta$,

$$\max\left\{ \left\| \widehat{\theta}_t - \theta^* \right\|_{\Sigma_t^{\theta^*}}^2, \left\| \widehat{\theta}_t - \theta^* \right\|_{\Sigma_t^{\widehat{\theta}_t}}^2 \right\} \le 8 C_\varepsilon^2 \beta, \tag{109}$$

if $\beta \ge Cd \log((1 + \varepsilon T^2 + (1 + \varepsilon)T)\delta^{-1})$ for some universal constant $C > 0$. So with probability at least $1 - \frac{p}{3}$, Equation (109) holds with $\beta \ge Cd \log((1 + \varepsilon T^2 + (1 + \varepsilon)T)3p^{-1})$.

Furthermore, we have

$$\left\| \widehat{\theta}_t - \theta^* \right\|_{\Sigma_t^\theta + I_d}^2 = \left\| \widehat{\theta}_t - \theta^* \right\|_{\Sigma_t^\theta}^2 + \left\| \widehat{\theta}_t - \theta^* \right\|_{I_d}^2 \le 8 C_\varepsilon^2 \beta + \left( \left\| \widehat{\theta}_t \right\|_2 + \left\| \theta^* \right\|_2 \right)^2$$
$$\le 8 C_\varepsilon^2 \beta + (B_\Theta + B_\Theta)^2 = 8 C_\varepsilon^2 \beta + 4 B_\Theta^2. \tag{110}$$

We define $\Gamma_t^2(\pi, y; \theta) = 2(\varepsilon \xi + C_\varepsilon^{(2)} \xi^2)$ with $C_\varepsilon^{(2)} = \varepsilon^2 \exp(2\varepsilon B_A)(2 + \varepsilon B_A \exp(2\varepsilon B_A))/2$ and,

$$\xi = \sqrt{\mathrm{Tr}\left( \left( \Sigma_t^\theta + I_d \right)^\dagger \Sigma^{\pi, y, \theta} \right)} \cdot \sqrt{8 C_\varepsilon^2 \beta + 4 B_\Theta^2}. \tag{111}$$

where $\Sigma_t^\theta = \sum_{i=1}^{t-1} \mathrm{Cov}^{\pi_i, y_i, \theta}[\phi^{\pi_i}(b)]$ is the covariance matrix, and $\Sigma^{\pi, y, \theta} = \mathrm{Cov}^{\pi, y, \theta}[\phi^\pi(b)]$ with the expectation taken by $b \sim \nu_\theta^{\pi, y}$.

Then, by equation 107 and equation 109, we have that, with probability at least $1 - \frac{p}{3}$, for any $t \in [T]$,

$$\mathrm{TV}(\nu_{\theta^*}^{\pi_t, y_t}, \nu_{\widehat{\theta}_t}^{\pi_t, y_t}) \le f_\varepsilon \left( \sqrt{\mathrm{Tr}\left( \left( \Sigma_t^{\theta^*} + I_d \right)^\dagger \Sigma^{\pi_t, y_t, \theta^*} \right)} \cdot \left\| \widehat{\theta}_t - \theta^* \right\|_{\Sigma_t^{\theta^*} + I_d} \right)$$
$$\le f_\varepsilon \left( \sqrt{\mathrm{Tr}\left( \left( \Sigma_t^{\theta^*} + I_d \right)^\dagger \Sigma^{\pi_t, y_t, \theta^*} \right)} \cdot \sqrt{8 C_\varepsilon^2 \beta + 4 B_\Theta^2} \right) = \frac{1}{2} \Gamma_t^2(\pi_t, y_t; \theta^*). \tag{112}$$

So with probability at least $1 - \frac{p}{3}$, Equation (109) holds with $\beta \ge Cd \log((1 + \varepsilon T^2 + (1 + \varepsilon)T)3p^{-1})$.

Therefore, by combining equation 106 and equation 112, we have

$$S_t^3 \le 2 \cdot \mathrm{TV}(\nu_{\widehat{\theta}_t}^{\pi_t, y_t}, \nu_{\theta^*}^{\pi_t, y_t}) \le \Gamma_t^2(\pi_t, y_t; \theta^*) \tag{113}$$

with high probability, provided that $\theta^* \in \mathcal{C}_\Theta^t(\beta)$ and equation 109 holds with $\beta \ge Cd \log((1 + \varepsilon T^2 + (1 + \varepsilon)T)3p^{-1})$. We just need to guarantee that $\theta^* \in \mathcal{C}_\Theta^t(\beta)$ holds with high probability, which has already been shown by equation 144 in Theorem I.1.

**Bounding** $\mathrm{Reg}(T)$ **for Scheme 1.** For the leader's regret in Scheme 1, by combining equation 96, equation 102 and equation 113, we have that, with high probability,

$$\mathrm{Reg}(T) = \sum_{t=1}^{T} \mathrm{SubOpt}(\pi_t, y_t) = \sum_{t=1}^{T} S_t^1 + \sum_{t=1}^{T} S_t^2 + \sum_{t=1}^{T} S_t^3$$

$$\leq 2 \sum_{t=1}^{T} \Gamma_t^1(a_t, b_t) + \sum_{t=1}^{T} \Gamma_t^2(\pi_t, y_t; \theta^*) + \sqrt{2T \log\left(\frac{3}{p}\right)}. \tag{114}$$

That above inequality is conditioned on $\theta^* \in \mathcal{C}_{\Theta}^t(\beta)$ in Theorem I.1, equation 94, equation 101, and equation 109 holds with $\beta \geq Cd \log((1 + \varepsilon T^2 + (1 + \varepsilon)T)3p^{-1})$.

Therefore, we have completed the proof of Theorem F.1. $\qquad\square$

## H.2 Proof of Theorem F.2 (leader's regret for Scheme 2)

The regret analysis of Scheme 2 in this subsection is different from that of Scheme 1 in Appendix H.1. For Scheme 2, we can first decompose the suboptimality $\mathrm{SubOpt}(\pi_t, y_t)$ by equation 92, in the same way as Scheme 2, but we will further deal with $S_t^3$ in equation 92 in a more complicated way. The analyses of $S_t^1$ and $S_t^2$ in equation 92 for Scheme 2 are the same as Scheme 1 in Appendix H.1 and hence are omitted here. So in this subsection we only develop an upper bound on $S_t^3$ in equation 92 and then derive an upper bound on the leader's regret $\mathrm{Reg}(T)$.

*Proof.* Recall that Scheme 2 is described as

$$\textbf{Scheme 2:} \quad \pi_t \leftarrow \arg\max_{\pi \in \Pi} \left\{ \left\langle \widetilde{U}_t, \pi \otimes \mu_{\widehat{\theta}_t^{\mathrm{MLE}}}^{\pi} \right\rangle_{\mathcal{A} \times \mathcal{B}} + C_{\mathrm{ii}} \cdot \Gamma_t^2(\pi, \mu_{\widehat{\theta}_t^{\mathrm{MLE}}}^{\pi}; \widehat{\theta}_t^{\mathrm{MLE}}) \right\}, \tag{115}$$

$$y_t \leftarrow \mu_{\widehat{\theta}_t^{\mathrm{MLE}}}^{\pi_t}, \quad \nu_t \leftarrow y_t. \tag{116}$$

We define

$$\widehat{y}_t^* := \mu_{\widehat{\theta}_t^{\mathrm{MLE}}}^{\pi^*}, \tag{117}$$

and also define the estimated value function

$$\widehat{W}_t := \max_{\pi \in \Pi} \left\{ \left\langle \widetilde{U}_t, \pi \otimes \mu_{\widehat{\theta}_t^{\mathrm{MLE}}}^{\pi} \right\rangle_{\mathcal{A} \times \mathcal{B}} + C_{\mathrm{ii}} \cdot \Gamma_t^2(\pi, \mu_{\widehat{\theta}_t^{\mathrm{MLE}}}^{\pi}; \widehat{\theta}_t^{\mathrm{MLE}}) \right\}. \tag{118}$$

By the optimality of $\pi_t$ in equation 18 in Scheme 2 and that $\pi^* \in \Pi$, we have that

$$\widehat{W}_t = \left\langle \widetilde{U}_t, \pi_t \otimes \mu_{\widehat{\theta}_t^{\mathrm{MLE}}}^{\pi_t} \right\rangle_{\mathcal{A} \times \mathcal{B}} + C_{\mathrm{ii}} \cdot \Gamma_t^2(\pi_t, \mu_{\widehat{\theta}_t^{\mathrm{MLE}}}^{\pi_t}; \widehat{\theta}_t^{\mathrm{MLE}})$$

$$\geq \left\langle \widetilde{U}_t, \pi^* \otimes \mu_{\widehat{\theta}_t^{\mathrm{MLE}}}^{\pi^*} \right\rangle_{\mathcal{A} \times \mathcal{B}} + C_{\mathrm{ii}} \cdot \Gamma_t^2(\pi^*, \mu_{\widehat{\theta}_t^{\mathrm{MLE}}}^{\pi^*}; \widehat{\theta}_t^{\mathrm{MLE}})$$

$$= \left\langle \widetilde{U}_t, \pi^* \otimes \widehat{y}_t^* \right\rangle_{\mathcal{A} \times \mathcal{B}} + C_{\mathrm{ii}} \cdot \Gamma_t^2(\pi^*, \widehat{y}_t^*; \widehat{\theta}_t^{\mathrm{MLE}}). \tag{119}$$

For $S_t^3$ in equation 92, we can further decompose it by

$$S_t^3 = \left\langle \widetilde{U}_t, \pi^* \otimes \nu^* \right\rangle - \left\langle \widetilde{U}_t, \pi_t \otimes \nu_{\theta^*}^{\pi_t, y_t} \right\rangle$$

$$= \underbrace{\left\langle \widetilde{U}_t, \pi^* \otimes \nu^* \right\rangle - \left\langle \widetilde{U}_t, \pi^* \otimes \widehat{y}_t^* \right\rangle}_{=:S_t^{3,3}} + \underbrace{\left\langle \widetilde{U}_t, \pi^* \otimes \widehat{y}_t^* \right\rangle - \widehat{W}_t}_{=:S_t^{3,4}}$$

$$+ \underbrace{\widehat{W}_t - \left\langle \widetilde{U}_t, \pi_t \otimes y_t \right\rangle}_{=:S_t^{3,5}} + \underbrace{\left\langle \widetilde{U}_t, \pi_t \otimes y_t \right\rangle - \left\langle \widetilde{U}_t, \pi_t \otimes \nu_{\theta^*}^{\pi_t, y_t} \right\rangle}_{=:S_t^{3,6}}. \tag{120}$$

For $S_t^{3,3}$ in equation 120, we have

$$S_t^{3,3} = \left\langle \widetilde{U}_t, \pi^* \otimes \nu^* \right\rangle - \left\langle \widetilde{U}_t, \pi^* \otimes \widehat{y}_t^* \right\rangle = \left\langle \widetilde{U}_t, \pi^* \otimes (\nu^* - \widehat{y}_t^*) \right\rangle \leq \left\langle \mathbf{1}(\cdot, \cdot), \pi^* \otimes |\nu^* - \widehat{\mathbf{y}}_{\mathbf{t}}^*| \right\rangle$$

$$= \sum_{b \in \mathcal{B}} \sum_{a \in \mathcal{A}} \pi^*(a|b) |\nu^*(b) - \widehat{y}_t^*(b)| = \sum_{b \in \mathcal{B}} |\nu^*(b) - \widehat{y}_t^*(b)| \sum_{a \in \mathcal{A}} \pi^*(a|b) = \sum_{b \in \mathcal{B}} |\nu^*(b) - \widehat{y}_t^*(b)|$$

$$= \|\nu^* - \widehat{y}_t^*\|_1 = 2 \cdot \mathrm{TV}(\nu^*, \widehat{y}_t^*). \tag{121}$$

Here the first inequality is by the boundedness of $\widetilde{U}_t$ within $[0, 1]$. The fourth equality is because $\nu^*(b)$ and $\widehat{y}_t^*(b)$ do not depend on $a \in \mathcal{A}$. The fifth equality is because $\pi^*(\cdot|b)$ is a probability distribution over $\mathcal{A}$ given any fixed action $b \in \mathcal{B}$. The last equality is by the relationship between the TV distance and the $\ell_1$-norm.

For $S_t^{3,4}$ in equation 120, we have

$$S_t^{3,4} = \langle \widetilde{U}_t, \pi^* \otimes \widehat{y}_t^* \rangle - \widehat{W}_t \leq \langle \widetilde{U}_t, \pi^* \otimes \widehat{y}_t^* \rangle - \langle \widetilde{U}_t, \pi^* \otimes \widehat{y}_t^* \rangle - C_{\mathrm{ii}} \cdot \Gamma_t^2(\pi^*, \widehat{y}_t^*; \widehat{\theta}_t^{\mathrm{MLE}})$$
$$= -C_{\mathrm{ii}} \cdot \Gamma_t^2(\pi^*, \widehat{y}_t^*; \widehat{\theta}_t^{\mathrm{MLE}}), \tag{122}$$

where the first inequality is by equation 119.

For $S_t^{3,5}$ in equation 120, we have

$$S_t^{3,5} = \widehat{W}_t - \langle \widetilde{U}_t, \pi_t \otimes y_t \rangle = \langle \widetilde{U}_t, \pi_t \otimes y_t \rangle + C_{\mathrm{ii}} \cdot \Gamma_t^2(\pi_t, y_t; \widehat{\theta}_t^{\mathrm{MLE}}) - \langle \widetilde{U}_t, \pi_t \otimes y_t \rangle$$
$$= C_{\mathrm{ii}} \cdot \Gamma_t^2(\pi_t, y_t; \widehat{\theta}_t^{\mathrm{MLE}}), \tag{123}$$

where the second equality is by the definition of $\widehat{W}_t$ in equation 118.

For $S_t^{3,6}$ in equation 120, we have

$$S_t^{3,6} = \langle \widetilde{U}_t, \pi_t \otimes y_t \rangle - \langle \widetilde{U}_t, \pi_t \otimes \nu_{\theta^*}^{\pi_t, y_t} \rangle = \langle \widetilde{U}_t, \pi_t \otimes (y_t - \nu_{\theta^*}^{\pi_t, y_t}) \rangle \leq \langle \mathbf{1}(\cdot, \cdot), \pi_{\mathbf{t}} \otimes |\mathbf{y_t} - \nu_{\theta^*}^{\pi_{\mathbf{t}}, \mathbf{y_t}}| \rangle$$
$$= \sum_{b \in \mathcal{B}} \sum_{a \in \mathcal{A}} \pi_t(a|b)|y_t(b) - \nu_{\theta^*}^{\pi_t, y_t}(b)| = \sum_{b \in \mathcal{B}} |y_t(b) - \nu_{\theta^*}^{\pi_t, y_t}(b)| \sum_{a \in \mathcal{A}} \pi^*(a|b)$$
$$= \sum_{b \in \mathcal{B}} |y_t(b) - \nu_{\theta^*}^{\pi_t, y_t}(b)| = \|y_t - \nu_{\theta^*}^{\pi_t, y_t}\|_1 = 2 \cdot \mathrm{TV}(y_t, \nu_{\theta^*}^{\pi_t, y_t}). \tag{124}$$

Here the first inequality is by the boundedness of $\widetilde{U}_t$ within $[0, 1]$. The fourth equality is because $y_t(b)$ and $\nu_{\theta^*}^{\pi_t, y_t}(b)$ do not depend on $a \in \mathcal{A}$. The fifth equality is because $\pi_t(\cdot|b)$ is a probability distribution over $\mathcal{A}$ given any fixed action $b \in \mathcal{B}$. The last equality is by the relationship between the TV distance and the $\ell_1$-norm.

By equation 121 and equation 122, we have

$$S_t^{3,3} + S_t^{3,4} \leq 2 \cdot \mathrm{TV}(\nu^*, \widehat{y}_t^*) - C_{\mathrm{ii}} \cdot \Gamma_t^2(\pi^*, \widehat{y}_t^*; \widehat{\theta}_t^{\mathrm{MLE}})$$
$$= 2 \cdot \mathrm{TV}(\mu_{\theta^*}^{\pi^*}, \mu_{\widehat{\theta}_t^{\mathrm{MLE}}}^{\pi^*}) - C_{\mathrm{ii}} \cdot \Gamma_t^2(\pi^*, \widehat{y}_t^*; \widehat{\theta}_t^{\mathrm{MLE}}), \tag{125}$$

where the first equality is because we have $\nu_{\theta^*}^{\pi^*, y^*} = y^* = \mu_{\theta^*}^{\pi^*}$ by Theorem C.1.

For simplicity, we denote $\widehat{\Sigma}_t^{\mathrm{MLE}} := \Sigma_t^{\widehat{\theta}_t^{\mathrm{MLE}}}$. By equation 168 in Theorem J.3 for the quantal responses, we have

$$\mathrm{TV}(\mu_{\theta^*}^{\pi^*}, \mu_{\widehat{\theta}_t^{\mathrm{MLE}}}^{\pi^*}) \leq C_{\mathrm{QF}} C_{\eta/\varepsilon} \cdot f_\varepsilon \left( \sqrt{\mathrm{Tr}\left( \left(\widehat{\Sigma}_t^{\mathrm{MLE}} + I_d\right)^\dagger \Sigma^{\pi^*, \widehat{y}_t^*, \widehat{\theta}^{\mathrm{MLE}}} \right)} \cdot \|\widehat{\theta}_t^{\mathrm{MLE}} - \theta^*\|_{\widehat{\Sigma}_t^{\mathrm{MLE}} + I_d} \right)$$

$$\leq C_{\mathrm{QF}} C_{\eta/\varepsilon} \cdot f_\varepsilon \left( \sqrt{\mathrm{Tr}\left( \left(\widehat{\Sigma}_t^{\mathrm{MLE}} + I_d\right)^\dagger \Sigma^{\pi^*, \widehat{y}_t^*, \widehat{\theta}^{\mathrm{MLE}}} \right)} \cdot \sqrt{8 C_\varepsilon^2 \beta + 4 B_\Theta^2} \right)$$

$$= C_{\mathrm{QF}} C_{\eta/\varepsilon} \cdot \frac{1}{2} \Gamma_t^2(\pi^*, \widehat{y}_t^*; \widehat{\theta}_t^{\mathrm{MLE}}) = \frac{1}{2} C_{\mathrm{ii}} \cdot \Gamma_t^2(\pi^*, \widehat{y}_t^*; \widehat{\theta}_t^{\mathrm{MLE}}), \tag{126}$$

where we define the constant $C_{\mathrm{ii}}$ as

$$C_{\mathrm{ii}} := C_{\mathrm{QF}} C_{\eta/\varepsilon} = \exp(2\eta)|\mathcal{B}|^{\frac{\tau}{\tau + \eta^{-1}}} \max\left\{ \frac{C_\eta^{(3)}}{C_\varepsilon^{(3)}}, \frac{\eta}{\varepsilon} \right\} \tag{127}$$

by equation 184 and equation 167, where $\varepsilon = (\tau + \eta^{-1})^{-1}$ and $C_x^{(2)} = x^2 \exp(2x B_A)(2 + x B_A \exp(2x B_A))/2$ for $x \in \{\eta, \varepsilon\}$.

So we have

$$S_t^{3,3} + S_t^{3,4} \leq 2 \cdot \mathrm{TV}(\mu_{\theta^*}^{\pi^*}, \mu_{\widehat{\theta}_t^{\mathrm{MLE}}}^{\pi^*}) - C_{\mathrm{ii}} \cdot \Gamma_t^2(\pi^*, \widehat{y}_t^*; \widehat{\theta}_t^{\mathrm{MLE}})$$
$$\leq C_{\mathrm{ii}} \cdot \Gamma_t^2(\pi^*, \widehat{y}_t^*; \widehat{\theta}_t^{\mathrm{MLE}}) - C_{\mathrm{ii}} \cdot \Gamma_t^2(\pi^*, \widehat{y}_t^*; \widehat{\theta}_t^{\mathrm{MLE}}) = 0, \tag{128}$$

by noting that $\widehat{y}_t^* = \mu_{\widehat{\theta}_t^{\mathrm{MLE}}}^{\pi^*}$ as defined in equation 117.

By combining equation 120, equation 123, equation 124 and equation 128, we can upper bound $S_t^3$ by

$$S_t^3 = S_t^{3,3} + S_t^{3,4} + S_t^{3,5} + S_t^{3,6} \leq C_{\mathrm{ii}} \cdot \Gamma_t^2(\pi_t, \mu_{\widehat{\theta}_t^{\mathrm{MLE}}}^{\pi_t}; \widehat{\theta}_t^{\mathrm{MLE}}) + 2 \cdot \mathrm{TV}(y_t, \nu_{\theta^*}^{\pi_t, y_t})$$
$$= C_{\mathrm{ii}} \cdot \Gamma_t^2(\pi_t, y_t; \widehat{\theta}_t^{\mathrm{MLE}}) + 2 \cdot \mathrm{TV}(y_t, \nu_{\theta^*}^{\pi_t, y_t}). \tag{129}$$

Then, by Theorem J.3 and equation 149 in Theorem I.1, we have that, with probability at least $1 - \frac{p}{3}$, for any $t \in [T]$,

$$2\mathrm{TV}(\nu_{\theta^*}^{\pi_t, y_t}, \nu_{\widehat{\theta}_t^{\mathrm{MLE}}}^{\pi_t, y_t}) \leq 2f_\varepsilon \left( \sqrt{\mathrm{Tr}\left( \left(\Sigma_t^{\theta^*} + I_d\right)^\dagger \Sigma^{\pi_t, y_t, \theta^*} \right)} \cdot \left\| \widehat{\theta}_t^{\mathrm{MLE}} - \theta^* \right\|_{\Sigma_t^{\theta^*} + I_d} \right)$$
$$\leq 2f_\varepsilon \left( \sqrt{\mathrm{Tr}\left( \left(\Sigma_t^{\theta^*} + I_d\right)^\dagger \Sigma^{\pi_t, y_t, \theta^*} \right)} \cdot \sqrt{8C_\varepsilon^2 \beta + 4B_\Theta^2} \right) = \Gamma_t^2(\pi_t, y_t; \theta^*). \tag{130}$$

By Theorem L.4, we have $\Gamma_t^2(\pi_t, y_t; \widehat{\theta}_t^{\mathrm{MLE}}) \leq \exp\left(4\varepsilon B_A\right) \cdot \Gamma_t^2(\pi_t, y_t; \theta^*)$. So we have

$$S_t^3 = S_t^{3,3} + S_t^{3,4} + S_t^{3,5} + S_t^{3,6} \leq C_{\mathrm{ii}} \cdot \Gamma_t^2(\pi_t, y_t; \widehat{\theta}_t^{\mathrm{MLE}}) + \Gamma_t^2(\pi_t, y_t; \theta^*)$$
$$\leq C_{\mathrm{ii}} \cdot \exp\left(4\varepsilon B_A\right) \cdot \Gamma_t^2(\pi_t, y_t; \theta^*) + \Gamma_t^2(\pi_t, y_t; \theta^*)$$
$$= (1 + C_{\mathrm{ii}} \cdot \exp\left(4\varepsilon B_A\right)) \cdot \Gamma_t^2(\pi_t, y_t; \theta^*). \tag{131}$$

By combining equation 96, equation 131 and equation 102, we have that, with high probability,

$$\mathrm{Reg}(T) = \sum_{t=1}^T \mathrm{SubOpt}(\pi_t, y_t) = \sum_{t=1}^T S_t^1 + \sum_{t=1}^T S_t^2 + \sum_{t=1}^T S_t^3$$
$$\leq 2\sum_{t=1}^T \Gamma_t^1(a_t, b_t) + (1 + C_{\mathrm{ii}} \cdot \exp\left(4\varepsilon B_A\right)) \cdot \sum_{t=1}^T \Gamma_t^2(\pi_t, y_t; \theta^*) + \sqrt{2T \log\left(\frac{3}{p}\right)}. \tag{132}$$

That above inequality is conditioned on $\left\|\widehat{\theta}_t^{\mathrm{MLE}} - \theta^*\right\|_{\Sigma_t^{\theta^*} + I_d} \leq \sqrt{8C_\varepsilon^2\beta + 4B_\Theta^2}$, equation 94 and equation 101.

Therefore, we have completed the proof of Theorem F.2. $\qquad\square$

### H.3 PROOF OF THEOREM F.3 (REGRET OF ICV FOR BOTH SCHEMES)

*Proof.* Recall that $\mathrm{ICV}(\pi, y; \theta) := \mathrm{TV}(y, \nu_\theta^{\pi, y})$ and it measures to what extent the IC condition is violated. Then, it is a natural idea to define the suboptimality of ICV as the value of ICV with the true parameter $\theta^*$, i.e.,

$$\mathrm{SubOpt}^{\mathrm{ICV}}(\pi, y) = \mathrm{ICV}(\pi, y; \theta^*) = \mathrm{TV}(y, \nu_{\theta^*}^{\pi, y}). \tag{133}$$

For Scheme 1, by the construction of $y_t$ in equation 17, we have $y_t = \mu_{\widehat{\theta}_t}^{\pi_t}$. Also, by Theorem C.1, we have $y_t = \mu_{\widehat{\theta}_t}^{\pi_t}$ implies $y_t = \nu_{\widehat{\theta}_t}^{\pi_t, y_t}$, i.e., $\mu_{\widehat{\theta}_t}^{\pi_t}$ is a fixed point of $\nu_{\widehat{\theta}_t}^{\pi_t, \cdot}$. Hence, by equation 112 we have that, with probability at least $1 - \frac{p}{3}$, for any $t \in [T]$,

$$\mathrm{SubOpt}^{\mathrm{ICV}}(\pi_t, y_t) = \mathrm{TV}(y_t, \nu_{\theta^*}^{\pi_t, y_t}) = \mathrm{TV}(\nu_{\widehat{\theta}_t}^{\pi_t, y_t}, \nu_{\theta^*}^{\pi_t, y_t}) \leq \frac{1}{2}\Gamma_t^2(\pi_t, y_t; \theta^*), \tag{134}$$

Therefore, we have the regret of ICV for the first $T$ steps, with probability at least $1 - \frac{p}{3}$,

$$\text{Reg}^{\text{ICV}}(T) = \sum_{t=1}^{T} \text{SubOpt}^{\text{ICV}}(\pi_t, y_t) \leq \frac{1}{2} \sum_{t=1}^{T} \Gamma_t^2(\pi_t, y_t; \theta^*). \tag{135}$$

The above inequality is conditioned on equation 109 in Theorem I.1.

For Scheme 2, by the construction of $y_t$ in equation 18, we have $y_t = \mu_{\widehat{\theta}_t^{\text{MLE}}}^{\pi_t}$. Note that both Theorem C.1 and equation 112 still hold when replacing $\widehat{\theta}_t$ by $\widehat{\theta}_t^{\text{MLE}}$. So we easily get an upper bound on $\text{Reg}^{\text{ICV}}(T)$ for Scheme 2 which is the same as equation 135 for Scheme 1, i.e., the upper bound in equation 135 holds for both Scheme 1 and Scheme 2.

Therefore, we have completed the proof of Theorem F.3. $\qquad\square$

### H.4 PROOF OF THEOREM F.4 (BOUNDING THE BONUS FUNCTIONS)

*Proof.* By the above analysis, in order to reach the two goals, we further need to upper bound the following bonus functions:

$$\sum_{t=1}^{T} \Gamma_t^1(a_t, b_t), \quad \sum_{t=1}^{T} \Gamma_t^2(\pi_t, y_t; \theta^*). \tag{136}$$

For $\sum \Gamma^1$, by using the elliptical potential lemma for vectors (Proposition 1 in Carpentier et al. (2020)), we can show that

$$\sum_{t=1}^{T} \Gamma_t^1(a_t, b_t) \leq C_1 d \sqrt{T d \log(T/d) \log(2dT^2/(p/3))} = \widetilde{\mathcal{O}}(|\mathcal{A}|^{\frac{3}{2}} |\mathcal{B}|^{\frac{3}{2}} \sqrt{T}). \tag{137}$$

For $\sum \Gamma^2$, we need to use the elliptical potential lemma. Recall the following definitions

$$\left(\Upsilon^{\pi,y,\theta} \phi\right)(b) = \phi^\pi(b) - \langle \phi^\pi, \nu^{\pi,y,\theta} \rangle,$$

$$\Sigma^{\pi,y,\theta} = \text{Cov}^{\pi,y,\theta} [\phi^\pi(b)] = \text{Cov}^{\pi,y,\theta} \left[(\Upsilon^{\pi,y,\theta} \phi)(b)\right],$$

$$\Sigma_t^\theta = \sum_{i=1}^{t-1} \text{Cov}^{\pi^i,\theta} \left[(\Upsilon^{\pi_i,y_i,\theta} \phi)(b)\right] = \sum_{i=1}^{t-1} \Sigma^{\pi_i,y_i,\theta}.$$

Note that $\Sigma_t^\theta$ is just a summation of $\Sigma^{\pi_i,y_i,\theta}$ over $i \in [t-1]$, which is a nonnegative definite $d$-dimensional matrix, and this gives us a self-normalized process. Therefore, we have from Theorem L.2 the elliptical potential lemma for matrices that

$$\sum_{t=1}^{T} \sqrt{\text{Tr}\left(\left(\Sigma_t^{\theta^*} + I_d\right)^\dagger \Sigma^{\pi_t,y_t,\theta^*}\right)} \leq \sqrt{C_0 dT \log\left(1 + 4B_\phi^2 T/d\right)},$$

$$\sum_{t=1}^{T} \text{Tr}\left(\left(\Sigma_t^{\theta^*} + I_d\right)^\dagger \Sigma^{\pi_t,y_t,\theta^*}\right) \leq C_0 d \log\left(1 + 4B_\phi^2 T/d\right),$$

where $C_0 = 4B_\phi^2/(\log(1 + B_\phi^2))$ and $4B_\phi^2$ upper bounds $\text{Tr}\left(\Sigma^{\pi,y,\theta^*}\right)$. To see why $B_\phi$ is a valid upper bound, we have by definition of $\Sigma^{\pi,y,\theta^*}$ that

$$\text{Tr}\left(\Sigma^{\pi,y,\theta^*}\right) = \text{Tr}\left(\text{Cov}^{\pi,y,\theta^*}\left[(\Upsilon^{\pi,y,\theta^*} \phi)(b)\right]\right) \leq \max_{b,\pi} \left\|(\Upsilon^{\pi,y,\theta^*} \phi)(b)\right\|^2 \leq 4B_\phi^2.$$

Therefore, the sum of $\Gamma_t^2(\pi_t, y_t; \theta^*)$ over $t \in [T]$ becomes

$$
\begin{aligned}
\sum_{t=1}^{T} \Gamma_t^2(\pi_t, y_t; \theta^*) &= \varepsilon\sqrt{8C_\varepsilon^2\beta + 4B_\Theta^2} \sum_{t=1}^{T} \sqrt{\mathrm{Tr}\left(\left(\Sigma_t^\theta + I_d\right)^\dagger \Sigma^{\pi_t, y_t, \theta}\right)} \\
&\quad + C_\varepsilon^{(3)}(8C_\varepsilon^2\beta + 4B_\Theta^2) \sum_{t=1}^{T} \mathrm{Tr}\left(\left(\Sigma_t^\theta + I_d\right)^\dagger \Sigma^{\pi_t, y_t, \theta}\right) \\
&\leq \varepsilon\sqrt{8C_\varepsilon^2\beta + 4B_\Theta^2}\sqrt{C_0 dT \log\left(1 + 4B_\phi^2 T/d\right)} \\
&\quad + C_\varepsilon^{(3)}(8C_\varepsilon^2\beta + 4B_\Theta^2)C_0 d \log\left(1 + 4B_\phi^2 T/d\right).
\end{aligned}
\tag{138}
$$

Therefore, we have completed the proof of Theorem F.4. $\qquad\square$

# I  CONFIDENCE SET ON THE FOLLOWER'S SIDE

**Lemma I.1** (Confidence set). *We define a distance $\rho$ on $\Theta$ by letting*

$$
\rho(\theta, \widetilde{\theta}) := \max_{\pi \in \Pi, y} \left\{ D_{\mathrm{H}}(\nu_\theta^{\pi, y}, \nu_{\widetilde{\theta}}^{\pi, y}), \ (1 + \varepsilon) \cdot \left\| Q_\theta^\pi - Q_{\widetilde{\theta}}^\pi \right\|_\infty \right\},
\tag{139}
$$

*where $B_A$ upper bounds the follower's A-, Q-, and V-functions and is specified in equation 204. Let $\mathcal{N}_\rho(\Theta, \epsilon)$ be the $\epsilon$-covering number of $\Theta$ with respect to the distance $\rho$. That is, $\mathcal{N}_\rho(\Theta, \epsilon)$ is the smallest $N \geq 1$ with the following property: there exists $\{\theta^i\}_{i \in [N]} \subseteq \Theta$ such that, for any $\theta \in \Theta$, there exists $\theta^i$ such that $\rho(\theta, \theta^i) \leq \epsilon$. For any $\delta \in (0, 1)$, we set $\beta \geq 2\log(e^3 \cdot \mathcal{N}_\rho(\Theta, T^{-1})/\delta)$. Then, with probability at least $1 - \delta$, the following properties hold for $\mathcal{C}_\Theta^t(\beta)$ defined in equation 16:*

*(i) (Validity) $\theta^* \in \mathcal{C}_\Theta^t(\beta)$;*

*(ii) (Accuracy) For any $\theta \in \Theta, t \in [T]$, it holds that*

$$
\sum_{i=1}^{t-1} D_{\mathrm{H}}^2\left(\nu_\theta^{\pi_i, y_i}, \nu_{\theta^*}^{\pi_i, y_i}\right) \leq \frac{1}{2}\left(\mathcal{L}_t(\theta) - \mathcal{L}_t(\theta^*) + \beta\right).
\tag{140}
$$

*Furthermore, for $\forall \theta' \in \{\theta^*, \theta\}, \theta \in \Theta$,*

$$
\sum_{i=1}^{t-1} \mathrm{Var}^{\pi_i, y_i, \theta'}\left[Q_\theta^{\pi_i}(b) - Q_{\theta^*}^{\pi_i}(b)\right] \leq 4C_\varepsilon^2 \cdot \left(\mathcal{L}_t(\theta) - \mathcal{L}_t(\theta^*) + \beta\right),
\tag{141}
$$

*where $\mathrm{Var}^{\pi, y, \theta}[Z] = \mathbb{E}^{\pi, y, \theta}[(Z - \mathbb{E}^{\pi, y, \theta}[Z])^2]$ and $C_\varepsilon = \varepsilon^{-1} + B_A$.*

*Proof.* The proof is similar to that of Lemma B.5 in Chen et al. (2023), with the main difference due to the divergence term in the follower's response model. We choose filtration $\mathcal{F}_{t-1} = \sigma(\tau^{1:t-1})$ where $\sigma(X)$ denotes the $\sigma$-algebra generated by $X$ and $\tau^{1:t-1}$ is just the history up to $t - 1$. Let $\mathcal{N}_\rho(\Theta, \epsilon)$ be the covering number for the $\epsilon$-covering net of $\Theta$ with respect to norm $\rho$ defined in equation 139. Let $\Theta_\epsilon$ be the $\epsilon$-covering net of $\Theta$. To simplify the notation, we define $\iota = \log(\mathcal{N}_\rho(\Theta, \epsilon)/\delta)$. Note that $\mathcal{L}_t(\theta) = -\sum_{i=1}^{t-1} \varepsilon A_\theta^{\pi_i, y_i}(b_i) - \tau\varepsilon\sum_{i=1}^{t-1} \log[y_i(b_i)]$. Then, for any $t \in [T]$, we have that, with

probability at least $1 - \delta$, the following inequality holds for all $\theta \in \Theta_\epsilon$,

$$\frac{1}{2}\left(-\mathcal{L}^t(\theta) + \mathcal{L}^t(\theta^*)\right) = \frac{1}{2}\sum_{i=1}^{t-1}\varepsilon A_\theta^{\pi_i,y_i}(b_i) + \frac{\tau\varepsilon}{2}\sum_{i=1}^{t-1}\log[y_i(b_i)] - \frac{1}{2}\sum_{i=1}^{t-1}\varepsilon A_{\theta^*}^{\pi_i,y_i}(b_i) - \frac{\tau\varepsilon}{2}\sum_{i=1}^{t-1}\log[y_i(b_i)]$$

$$= \sum_{i=1}^{t-1}\frac{\varepsilon}{2}(A_\theta^{\pi_i,y_i} - A_{\theta^*}^{\pi_i,y_i})(b_i) \leq \sum_{i=1}^{t-1}\log\mathbb{E}^{\pi_i,y_i}\left[\exp\left\{\frac{\varepsilon}{2}(A_\theta^{\pi_i,y_i} - A_{\theta^*}^{\pi_i,y_i})\right\}\right] + \log\left(\mathcal{N}_\rho(\Theta,\epsilon)/\delta\right)$$

$$= \sum_{i=1}^{t-1}\log\mathbb{E}^{\pi_i,y_i}\left[\sqrt{\frac{y_i^{\tau\varepsilon}\exp(\varepsilon A_\theta^{\pi_i,y_i})}{y_i^{\tau\varepsilon}\exp(\varepsilon A_{\theta^*}^{\pi_i,y_i})}}\right] + \log\left(\mathcal{N}_\rho(\Theta,\epsilon)/\delta\right)$$

$$= \sum_{i=1}^{t-1}\log\mathbb{E}^{\pi_i,y_i}\left[\sqrt{\nu_\theta^{\pi_i,y_i}/\nu_{\theta^*}^{\pi_i,y_i}}\right] + \log\left(\mathcal{N}_\rho(\Theta,\epsilon)/\delta\right)$$

$$\leq -\sum_{i=1}^{t-1}D_H^2\left(\nu_\theta^{\pi_i,y_i}, \nu_{\theta^*}^{\pi_i,y_i}\right) + \log\left(\mathcal{N}_\rho(\Theta,\epsilon)/\delta\right), \tag{142}$$

where the expectation is taken with respect to the true model. Here, the first inequality holds by applying Theorem L.1 and taking a union bound over the $\epsilon$-covering net. The second inequality holds by noting that $\log(x) \leq x - 1$ with $x = \mathbb{E}\left[\sqrt{\nu_\theta/\nu_{\theta^*}}\right]$ and $D_H^2(\nu_\theta, \nu_{\theta^*}) = 1 - \mathbb{E}\left[\sqrt{\nu_\theta/\nu_{\theta^*}}\right]$ by calculation from the definition of the Hellinger distance.

Meanwhile, by the definition of the distance $\rho$ in equation 139, for any $\theta, \widetilde{\theta} \in \Theta$, we have

$$|D_H^2(\nu_\theta^{\pi,y}, \nu_{\theta^*}^{\pi,y}) - D_H^2(\nu_{\widetilde{\theta}}^{\pi,y}, \nu_{\theta^*}^{\pi,y})|$$

$$\leq (D_H(\nu_\theta^{\pi,y}, \nu_{\theta^*}^{\pi,y}) + D_H(\nu_{\widetilde{\theta}}^{\pi,y}, \nu_{\theta^*}^{\pi,y})) \cdot \left|D_H(\nu_\theta^{\pi,y}, \nu_{\theta^*}^{\pi,y}) - D_H(\nu_{\widetilde{\theta}}^{\pi,y}, \nu_{\theta^*}^{\pi,y})\right|$$

$$\leq 2D_H(\nu_{\widetilde{\theta}}^{\pi,y}, \nu_\theta^{\pi,y}) \leq 2\rho(\theta,\widetilde{\theta}),$$

where the second inequality holds by noting that the Hellinger distance is no greater than 1, and that the Hellinger distance satisfies the triangle inequality as a norm, and the last inequality holds by definition of $\rho$. Moreover, by noting that $\mathcal{L}_t(\theta) = -\sum_{i=1}^{t-1}\varepsilon A_\theta^{\pi_i,y_i}(b_i) - \tau\varepsilon\sum_{i=1}^{t-1}\log[y_i(b_i)]$, we have

$$\left|\mathcal{L}_t(\theta) - \mathcal{L}_t(\widetilde{\theta})\right| \leq \varepsilon T\max_{i\in[t-1]}\left|A_\theta^{\pi_i,y_i}(b^i) - A_{\widetilde{\theta}}^{\pi_i,y_i}(b^i)\right| \leq 2\varepsilon T\max_{i\in[t-1]}\left\|Q_\theta^{\pi_i} - Q_{\widetilde{\theta}}^{\pi_i}\right\|_\infty \leq 2T\cdot\rho(\theta,\widetilde{\theta}),$$

where the second inequality holds by noting that $|V_\theta^{\pi,y} - V_{\widetilde{\theta}}^{\pi,y}| \leq \|Q_\theta^\pi - Q_{\widetilde{\theta}}^\pi\|_\infty$ by Theorem K.2, and the last inequality holds by the definition of $\rho$. Therefore, adding an extra term $3T\epsilon$ to the right-hand side of equation 142 extends the result to any $\theta \in \Theta$ by definition of the covering net $\Theta_\epsilon$. We thus obtain for all $\theta \in \Theta, t \in [T]$ with probability $1 - \delta$,

$$\frac{1}{2}\left(-\mathcal{L}_t(\theta) + \mathcal{L}_t(\theta^*)\right) \leq -\sum_{i=1}^{t-1}D_H^2\left(\nu_\theta^{\pi_i,y_i}, \nu_{\theta^*}^{\pi_i,y_i}\right) + \log\left(\mathcal{N}_\rho(\Theta,\epsilon)/\delta\right) + 3T\epsilon. \tag{143}$$

In the sequel, we take $\epsilon = T^{-1}$ and let $\iota = \log\left(\mathcal{N}_\rho(\Theta, T^{-1})/\delta\right) + 3$. Now, we plug in $\widehat{\theta}_{\mathrm{MLE}} = \arg\min_{\theta'\in\Theta}\mathcal{L}_t(\theta')$ in equation 143 and obtain by the nonnegativity of the Hellinger distance that

$$\mathcal{L}_t(\theta^*) \leq \inf_{\theta'\in\Theta}\mathcal{L}_t(\theta') + 2\iota \leq \mathcal{L}_t(\widehat{\theta}_{\mathrm{MLE}}) + 2\iota, \tag{144}$$

which guarantees that our confidence set is indeed valid by letting

$$\beta \geq 2\iota = 2\log(e^3 \cdot \mathcal{N}_\rho(\Theta, T^{-1})/\delta).$$

Next, we show that our confidence set is also accurate. For equation 143, we have that

$$\sum_{i=1}^{t-1}D_H^2\left(\nu_\theta^{\pi_i,y_i}, \nu_{\theta^*}^{\pi_i,y_i}\right) \leq \frac{1}{2}\left(\mathcal{L}_t(\theta) - \mathcal{L}_t(\theta^*)\right) + \iota \leq \frac{1}{2}\left(\mathcal{L}_t(\theta) - \mathcal{L}_t(\theta^*) + \beta\right), \tag{145}$$

Now, if $\theta \in \mathcal{C}_\Theta^t(\beta)$, it follows directly from equation 145 that

$$\sum_{i=1}^{t-1} D_{\mathrm{H}}^2\left(\nu_\theta^{\pi_i, y_i}, \nu_{\theta^*}^{\pi_i, y_i}\right) \le \beta,$$

which shows that our confidence set is also valid and gives equation 140.

We next show how to derive the bound for the Q function. Invoking Theorem J.6, we have that

$$8\left(\frac{1 + \varepsilon B_A}{\varepsilon}\right)^2 \cdot D_{\mathrm{H}}^2\left(\nu_{\widehat{\theta}}^{\pi, y}, \nu_{\theta^*}^{\pi, y}\right) \ge \left\langle \nu_{\theta^*}^{\pi, y}, (A_{\widehat{\theta}}^{\pi, y} - A_{\theta^*}^{\pi, y})^2 \right\rangle$$

$$\ge \mathbb{E}^{\pi, y, \theta^*}\left(\left(\mathbb{E}_b^{\pi, y, \theta^*} - \mathbb{E}^{\pi, y, \theta^*}\right)[A_{\widehat{\theta}}^{\pi, y} - A_{\theta^*}^{\pi, y}]\right)^2 = \mathbb{E}^{\pi, y, \theta^*}\left(\left(\mathbb{E}_b^{\pi, y, \theta^*} - \mathbb{E}^{\pi, y, \theta^*}\right)[Q_{\widehat{\theta}}^\pi - Q_{\theta^*}^\pi]\right)^2, \tag{146}$$

where the second inequality follows from direct calculation (expansion), and the last equality holds by invoking equation 181 in Theorem J.5. Swapping the positions of $\theta^*$ and $\widehat{\theta}$ by the exchangeability of the Hellinger distance leads to another version of the result.

Plugging equation 146 into the previous accuracy guarantees with $C_\varepsilon = \varepsilon^{-1} + B_A$, we have

$$\sum_{i=1}^{t-1} \mathrm{Var}^{\pi_i, y_i, \theta'}\left[Q_\theta^{\pi_i}(b) - Q_{\theta^*}^{\pi_i}(b)\right] = \sum_{i=1}^{t-1} \mathbb{E}^{\pi_i, y_i, \theta^*}\left(\left(\mathbb{E}_b^{\pi_i, y_i, \theta^*} - \mathbb{E}^{\pi_i, y_i, \theta^*}\right)[Q_{\widehat{\theta}}^{\pi_i} - Q_{\theta^*}^{\pi_i}]\right)^2$$

$$\le 8\left(\frac{1 + \varepsilon B_A}{\varepsilon}\right)^2 \cdot \sum_{i=1}^{t-1} D_{\mathrm{H}}^2\left(\nu_{\widehat{\theta}}^{\pi_i, y_i}, \nu_{\theta^*}^{\pi_i, y_i}\right) \le 4C_\varepsilon^2 \cdot \left(\mathcal{L}_t(\theta) - \mathcal{L}_t(\theta^*) + \beta\right), \tag{147}$$

which proves equation 141. Therefore, we have proved equation 140 and equation 141. $\qquad\square$

We remark that if $\theta \in \mathcal{C}_\Theta^t(\beta)$, the guarantees in the accuracy result equation 141 is just $8C_\varepsilon^2\beta$ since $\mathcal{L}_t(\theta) - \mathcal{L}_t(\theta^*) \le \mathcal{L}_t(\theta) - \inf_{\theta' \in \Theta} \mathcal{L}_t(\theta') \le \beta$.

Next, following the analysis of (A.17) in Chen et al. (2023), we have a similar result for the covering number of $\Theta$, which is given by

$$\log \mathcal{N}_\rho(\Theta, \epsilon) \le d \log\left(1 + \frac{2B_\Theta B_\phi}{\min\{\epsilon^2/(8\varepsilon), \epsilon/(1+\varepsilon)\}}\right) \lesssim d \log\left(1 + \varepsilon/\epsilon^2 + (1+\varepsilon)/\epsilon\right), \tag{148}$$

where $B_\Theta$ bounds $\Theta$ in 2-norm and $B_\phi$ bounds the feature mapping $\phi$ in 2-norm for each action pair $(a, b)$. Using this result, we have the following remark.

*Remark* I.2. (myopic and linear case) Suppose that $\beta \ge Cd \log((1 + \varepsilon T^2 + (1+\varepsilon)T)\delta^{-1})$ for some universal constant $C > 0$, equation 141 further implies that

$$\max\left\{\left\|\widehat{\theta} - \theta^*\right\|_{\Sigma_t^{\theta^*}}^2, \left\|\widehat{\theta} - \theta^*\right\|_{\Sigma_t^{\widehat{\theta}}}^2\right\} \le 8C_\varepsilon^2\beta, \tag{149}$$

where $C_\varepsilon = \varepsilon^{-1} + B_A$, $B_A$ is specified in equation 204, and $\Sigma_t^\theta$ is a data-dependent covariance matrix defined as

$$\Sigma_t^\theta = \sum_{i=1}^{t-1} \mathrm{Cov}^{\pi_i, y_i, \theta}\left[\phi^{\pi_i}(b)\right], \tag{150}$$

where $\mathrm{Cov}^{\pi, y, \theta}[\phi^\pi(b)]$ represents the covariance matrix of the feature $\phi^\pi$ with respect to $\nu_\theta^{\pi, y}$. We then use equation 141 and note that $(Q_\theta^{\pi_i} - Q_{\theta^*}^{\pi_i})(b) = \phi^{\pi_i}(b)^\top(\theta - \theta^*)$, which leads to equation 149.

## J    ERRORS OF THE FOLLOWER'S RESPONSE MODELS

The results in this chapter are mostly inspired by Chen et al. (2023), while here we introduce policy recommendation when the horizon length $H = 1$. The main difference is due to the introduced recommendation policy as well as the divergence term in the follower's response model.

For leader's policy pair $(\pi, y)$ and any parameter $\theta \in \Theta$, we define a matrix $\Sigma^{\pi,y,\theta}$ as

$$\Sigma^{\pi,y,\theta} := \mathrm{Cov}^{\pi,y,\theta}[\phi^\pi(b)] = \mathbb{E}\big[(\phi^\pi(b) - \mathbb{E}[\phi^\pi(b)])(\phi^\pi(b) - \mathbb{E}[\phi^\pi(b)])^\top\big], \qquad (151)$$

where the expectation is taken with respect to $b \sim \nu_\theta^{\pi,y}(\cdot)$.

In the following, we denote $\varepsilon := (\tau + \eta^{-1})^{-1}$ for simplicity of notation. Then, by equation 5, we have the follower's response policy to $(\pi, y)$ with parameter $\theta \in \Theta$, to be

$$\nu_\theta^{\pi,y}(b) = y(b)^{\tau\varepsilon} \cdot \exp\left\{\varepsilon A_\theta^{\pi,y}(b)\right\}. \qquad (152)$$

To simplify the notation, in the rest of this subsection, we let $\mathbb{E} = \mathbb{E}^{\pi,M^*}$ and $\mathbb{E}_z[\cdot] = \mathbb{E}^{\pi,M^*}[\cdot \mid z]$ for any variable $z$.

**Lemma J.1** (Errors of response models). *Fix the leader's policy pair $(\pi, y) \in \Pi \times \mathcal{Y}_{\mathrm{QR}}$. Let $\theta^* \in \Theta$ be the true parameter for the follower's reward function, and let $\widehat{\theta} \in \Theta$ be any estimate of $\theta^* \in \Theta$.*

*Let $\nu_{\theta^*}^{\pi,y}$ and $\nu_{\widehat{\theta}}^{\pi,y}$ be the follower's response policy against $(\pi, y)$ based on $\theta^*$ and $\widehat{\theta}$, respectively, as defined in equation 5. Then, we can upper bound the TV distance between $\nu_{\widehat{\theta}}^{\pi,y}$ and $\nu_{\theta^*}^{\pi,y}$ by*

$$\mathrm{TV}\left(\nu_{\widehat{\theta}}^{\pi,y}, \nu_{\theta^*}^{\pi,y}\right) \leq \varepsilon \mathbb{E}\left[\left|(Q_{\widehat{\theta}}^\pi(b) - Q_{\theta^*}^\pi(b)) - \mathbb{E}[Q_{\widehat{\theta}}^\pi(b) - Q_{\theta^*}^\pi(b)]\right|\right]$$

$$+ C_\varepsilon^{(3)} \mathbb{E}\left[\left(\left(Q_{\widehat{\theta}}^\pi(b) - Q_{\theta^*}^\pi(b)\right) - \mathbb{E}\left[Q_{\widehat{\theta}}^\pi(b) - Q_{\theta^*}^\pi(b)\right]\right)^2\right]. \qquad (153)$$

*Also, let $\mu_{\theta^*}^\pi$ and $\mu_{\widehat{\theta}}^\pi$ be the quantal response policy against $\pi$ based on $\theta^*$ and $\widehat{\theta}$, respectively, as defined in equation 6. Then, we can upper bound the TV distance between $\mu_{\theta^*}^\pi$ and $\mu_{\widehat{\theta}}^\pi$ by*

$$\mathrm{TV}(\mu_{\widehat{\theta}}^\pi, \mu_{\theta^*}^\pi) \leq \eta C_{\mathrm{QF}} \cdot \mathbb{E}\left[\left|(Q_{\widehat{\theta}}^\pi(b) - Q_{\theta^*}^\pi(b)) - \mathbb{E}[Q_{\widehat{\theta}}^\pi(b) - Q_{\theta^*}^\pi(b)]\right|\right]$$

$$+ C_{\mathrm{QF}} \cdot C_\eta^{(3)} \cdot \mathbb{E}\left[\left(\left(Q_{\widehat{\theta}}^\pi(b) - Q_{\theta^*}^\pi(b)\right) - \mathbb{E}\left[Q_{\widehat{\theta}}^\pi(b) - Q_{\theta^*}^\pi(b)\right]\right)^2\right]. \qquad (154)$$

*Here we define the constant*

$$C_\sigma^{(3)} = \frac{1}{2}\sigma^2 \exp(2\sigma B_A)\big(2 + \sigma B_A \cdot \exp\left(2\sigma B_A\right)\big) \qquad (155)$$

*for $\sigma \in \{\eta, \varepsilon\}$. The constant $B_A$ defined in equation 204 is an upper bound on the value functions of the follower. The constant $C_{\mathrm{QF}}$ defined in equation 184 characterizes the relationship between the quantal response and follower's response.*

*The expectations $\mathbb{E} = \mathbb{E}_{\nu_{\theta^*}^{\pi,y}}$ in equation 153 and equation 154 are both taken with respect to the follower's response policy based on the true parameter $\theta^*$, i.e., $b \sim \nu_{\theta^*}^{\pi,y}(\cdot)$.*

*Proof.* The proof is similar to that of Corollary B.3 in Chen et al. (2023), with the main difference due to the divergence term in the follower's response model. Since the leader's policy pair $(\pi, y)$ is fixed, for simplicity, we denote $Q = Q_{\theta^*}^\pi$, $\widetilde{Q} = Q_{\widehat{\theta}}^\pi$, $V = V_{\theta^*}^{\pi,y}$, $\widetilde{V} = V_{\widehat{\theta}}^{\pi,y}$, $A = A_{\theta^*}^{\pi,y}$, $\widetilde{A} = A_{\widehat{\theta}}^{\pi,y}$, $\nu = \nu_{\theta^*}^{\pi,y}$ and $\widetilde{\nu} = \nu_{\widehat{\theta}}^{\pi,y}$ in the sequel.

To begin with, we invoke equation 171 in Theorem J.4 and obtain that

$$\mathrm{TV}(\nu, \widetilde{\nu}) \leq \varepsilon \langle \nu, |\widetilde{A} - A| + \frac{\varepsilon}{2}\exp\left(\varepsilon|\widetilde{A} - A|\right)(\widetilde{A} - A)^2\rangle_\mathcal{B} \leq \varepsilon\langle\nu, |\widetilde{A} - A| + \frac{\varepsilon}{2}\exp(2\varepsilon B_A)(\widetilde{A} - A)^2\rangle_\mathcal{B}$$

$$= \varepsilon\mathbb{E}_\nu\big[|\widetilde{A} - A|\big] + \frac{\varepsilon^2}{2}\exp\left(2\varepsilon B_A\right)\mathbb{E}_\nu\big[(\widetilde{A} - A)^2\big]$$

$$= \varepsilon\mathbb{E}_\nu\big[|\widetilde{A} - A|\big] + \frac{\varepsilon^2}{2}\exp\left(2\varepsilon B_A\right)\big(\mathrm{Var}_\nu(\widetilde{A} - A) + \big(\mathbb{E}_\nu[\widetilde{A} - A]\big)^2\big),$$

where the last equality holds by the variance-mean decomposition. Here the expectation and variance are taken with respect to $b \sim \nu(\cdot)$.

Now, by using Theorem J.5, we have

$$\mathbb{E}\big[|\widetilde{A} - A|\big] \le \mathbb{E}\big[|(\widetilde{Q} - Q) - \mathbb{E}[\widetilde{Q} - Q]|\big] + \varepsilon^{-1}\mathrm{KL}(\nu \parallel \widetilde{\nu}). \tag{156}$$

For the variance term, we have

$$\mathrm{Var}(\widetilde{A} - A) = \mathbb{E}\big[\big((\widetilde{A} - A) - \mathbb{E}(\widetilde{A} - A)\big)^2\big] = \mathbb{E}\big[\big((\widetilde{Q} - Q) - \mathbb{E}[\widetilde{Q} - Q]\big)^2\big], \tag{157}$$

where the last equality holds because $V$ and $\widetilde{V}$ do not involve $b$. Furthermore, we note that

$$\varepsilon\mathbb{E}_\nu[A - \widetilde{A}] = \mathrm{KL}(\nu \parallel \widetilde{\nu}) \le 2\varepsilon B_A. \tag{158}$$

Thus, combining the inequalities above, we have we have

$$\mathrm{TV}(\nu, \widetilde{\nu}) \le \varepsilon \cdot \mathbb{E}\big[|(\widetilde{Q} - Q) - \mathbb{E}[\widetilde{Q} - Q]|\big] + \frac{\varepsilon^2}{2} \cdot \exp\left(2\varepsilon B_A\right) \cdot \mathbb{E}\big[\big((\widetilde{Q} - Q) - \mathbb{E}[\widetilde{Q} - Q]\big)^2\big]$$
$$+ \mathrm{KL}(\nu \parallel \widetilde{\nu}) + \exp\left(2\varepsilon B_A\right)/2 \cdot \big(\mathrm{KL}(\nu \parallel \widetilde{\nu})\big)^2$$
$$\le \varepsilon \cdot \mathbb{E}\big[|(\widetilde{Q} - Q) - \mathbb{E}[\widetilde{Q} - Q]|\big] + \frac{\varepsilon^2}{2} \cdot \exp\left(2\varepsilon B_A\right) \cdot \mathbb{E}\big[\big((\widetilde{Q} - Q) - \mathbb{E}[\widetilde{Q} - Q]\big)^2\big]$$
$$+ \big(1 + \varepsilon B_A \exp\left(2\varepsilon B_A\right)\big) \cdot \mathrm{KL}(\nu \parallel \widetilde{\nu}), \tag{159}$$

where the last inequality holds by $\mathrm{KL}(\nu \parallel \widetilde{\nu}) \le 2\varepsilon B_A$.

In the following, we handle the KL divergence term. We calculate the derivative of $\varepsilon^{-2}\mathrm{KL}(\nu \parallel \widetilde{\nu})$ with respect to $\widetilde{Q}$ and obtain

$$\partial_{\widetilde{Q}}\big(\varepsilon^{-2}\mathrm{KL}(\nu \parallel \widetilde{\nu})\big) = \varepsilon^{-1}\partial_{\widetilde{Q}}\big(\mathbb{E}[A - \widetilde{A}]\big) = \varepsilon^{-1}\big(\partial_{\widetilde{Q}}\widetilde{V} - \nu\big) = \varepsilon^{-1}\left(\widetilde{\nu} - \nu\right).$$

Here the first equality follows from $\varepsilon\mathbb{E}[A - \widetilde{A}] = \mathrm{KL}(\nu \parallel \widetilde{\nu})$, the second equality holds because $\nu$ and $A$ do not depend on $\widetilde{Q}$, and $\widetilde{A} = \widetilde{Q} - \widetilde{V}$. Moreover, the last equality holds because

$$\widetilde{V} = \varepsilon^{-1}\log\left(\sum_{b\in\mathcal{B}}[y(b)]^{\tau\varepsilon} \cdot \exp\big(\varepsilon \cdot \widetilde{Q}(b)\big)\right),$$

and also $\partial_{\widetilde{Q}}\mathbb{E}[\widetilde{Q}] = \nu$. We further take a second-order derivative and obtain

$$\partial^2_{\widetilde{Q}\widetilde{Q}}\big(\varepsilon^{-2}\mathrm{KL}(\nu \parallel \widetilde{\nu})\big) = \varepsilon^{-1}\partial_{\widetilde{Q}}\widetilde{\nu} = \mathrm{diag}(\widetilde{\nu}) - \widetilde{\nu}\widetilde{\nu}^\top =: \mathrm{H},$$

where the last equality holds for the vector case. Note that the Hessian is upper and lower bounded by L where $\mathrm{L} = \mathrm{diag}(\nu) - \nu\nu^\top$, which is proved by the following Theorem J.2.

Using the upper bound in equation 162 in Theorem J.2, we have for the KL divergence that

$$\varepsilon^{-2}\mathrm{KL}(\nu \parallel \widetilde{\nu}) \le 1/2 \cdot (\widetilde{Q} - Q)^\top \mathrm{H}(\widetilde{Q} - Q) \le \exp\left(2\varepsilon B_A\right)/2 \cdot (\widetilde{Q} - Q)^\top \mathrm{L}(\widetilde{Q} - Q)$$
$$= \frac{\exp\left(2\varepsilon B_A\right)}{2} \cdot (\widetilde{Q} - Q)^\top\big(\mathrm{diag}(\nu) - \nu\nu^\top\big)(\widetilde{Q} - Q)$$
$$= \frac{\exp\left(2\varepsilon B_A\right)}{2} \cdot \mathbb{E}\big[\big((\widetilde{Q} - Q) - \mathbb{E}[\widetilde{Q} - Q]\big)^2\big],$$

where the first inequality holds by noting that the derivative of the KL-divergence at $\widetilde{\nu} = \nu$ is zero, and we upper bound the KL-divergence only by the second order term. Furthermore, the second inequality holds because $\mathrm{H} \preceq \exp(2\varepsilon B_A) \cdot \mathrm{L}$, which is proved by Theorem J.2. Therefore, we conclude for equation 159 that

$$\mathrm{TV}\left(\nu, \widetilde{\nu}\right) \le \varepsilon \cdot \mathbb{E}\big[|(\widetilde{Q} - Q) - \mathbb{E}[\widetilde{Q} - Q]|\big] + \frac{\varepsilon^2}{2}\exp\left(2\varepsilon B_A\right) \cdot \mathbb{E}\big[\big((\widetilde{Q} - Q) - \mathbb{E}[\widetilde{Q} - Q]\big)^2\big]$$
$$+ \big(1 + \varepsilon B_A \cdot \exp\left(2\varepsilon B_A\right)\big) \cdot \mathrm{KL}(\nu \parallel \widetilde{\nu})$$
$$\le \varepsilon \cdot \mathbb{E}\big[|(\widetilde{Q} - Q) - \mathbb{E}[\widetilde{Q} - Q]|\big]$$
$$+ \frac{\varepsilon^2\exp(2\varepsilon B_A)}{2}\big(2 + \varepsilon B_A \cdot \exp\left(2\varepsilon B_A\right)\big) \cdot \mathbb{E}\big[\big((\widetilde{Q} - Q) - \mathbb{E}[\widetilde{Q} - Q]\big)^2\big].$$

We now turn to upper bounding $\mathrm{TV}(\mu_{\theta^*}^\pi, \mu_{\widetilde{\theta}}^\pi)$. By equation 172 in Theorem J.4, we have for the TV distance between the corresponding quantal responses,

$$\mathrm{TV}(\mu_{\theta^*}^\pi, \mu_{\widetilde{\theta}}^\pi) \leq \eta \cdot \left\langle \mu_{\theta^*}^\pi, |\widetilde{A} - A| + \frac{\eta}{2} \exp\left(\eta|\widetilde{A} - A|\right) \cdot (\widetilde{A} - A)^2 \right\rangle_{\mathcal{B}}$$

$$\leq \eta C_{\mathrm{QF}} \cdot \left\langle \nu_{\theta^*}^{\pi,y}, |\widetilde{A} - A| + \frac{\eta}{2} \exp\left(\eta|\widetilde{A} - A|\right) \cdot (\widetilde{A} - A)^2 \right\rangle_{\mathcal{B}}$$

$$\leq \eta C_{\mathrm{QF}} \cdot \left\{ \mathbb{E}_\nu\left[|\widetilde{A} - A|\right] + \frac{\eta}{2} \exp\left(2\eta B_A\right) \cdot \mathbb{E}_\nu\left[(\widetilde{A} - A)^2\right] \right\}$$

$$\leq \eta C_{\mathrm{QF}} \cdot \left\{ \mathbb{E}_\nu\left[|\widetilde{A} - A|\right] + \frac{\eta}{2} \exp\left(2\eta B_A\right) \cdot \left(\mathrm{Var}_\nu(\widetilde{A} - A) + \left(\mathbb{E}_\nu[\widetilde{A} - A]\right)^2\right) \right\}, \tag{160}$$

where the second inequality is by equation 183 in Theorem K.1 and the constant $C_{\mathrm{QF}} := \exp(2\eta) \cdot |\mathcal{B}|^{\frac{\tau}{\tau+\eta-1}}$ is defined in equation 184.

Finally, by plugging equation 156, equation 157 and equation 158 into equation 160, we have

$$\mathrm{TV}(\mu_{\theta^*}^\pi, \mu_{\widetilde{\theta}}^\pi) \leq \eta C_{\mathrm{QF}} \cdot \frac{\eta}{2} \exp\left(2\eta B_A\right) \cdot \left(\mathrm{Var}_\nu(\widetilde{A} - A) + \left(\mathbb{E}_\nu[\widetilde{A} - A]\right)^2\right) + \eta C_{\mathrm{QF}} \cdot \mathbb{E}_\nu\left[|\widetilde{A} - A|\right]$$

$$= \eta C_{\mathrm{QF}} \left(\frac{\eta}{2} \exp\left(2\eta B_A\right) + \frac{\varepsilon}{2} \exp\left(2\varepsilon B_A\right) + \frac{\eta \varepsilon B_A}{2} \exp\left(2(\eta + \varepsilon) B_A\right)\right) \cdot \mathbb{E}\left[\left((\widetilde{Q} - Q) - \mathbb{E}[\widetilde{Q} - Q]\right)^2\right]$$

$$+ \eta C_{\mathrm{QF}} \cdot \mathbb{E}\left[\left|(\widetilde{Q} - Q) - \mathbb{E}[\widetilde{Q} - Q]\right|\right]$$

$$\leq C_{\mathrm{QF}} \cdot \frac{\eta^2 \exp(2\eta B_A)}{2} \left(2 + \eta B_A \exp\left(2\eta B_A\right)\right) \cdot \mathbb{E}\left[\left((\widetilde{Q} - Q) - \mathbb{E}[\widetilde{Q} - Q]\right)^2\right]$$

$$+ \eta C_{\mathrm{QF}} \cdot \mathbb{E}\left[\left|(\widetilde{Q} - Q) - \mathbb{E}[\widetilde{Q} - Q]\right|\right], \tag{161}$$

where the last inequality uses the fact that $\varepsilon \leq \eta$. This finishes the proof of Theorem J.1. $\qquad\square$

**Proposition J.2.** *Let* $\mathrm{H} = \mathrm{diag}(\widetilde{\nu}) - \widetilde{\nu}\widetilde{\nu}^\top$ *and* $\mathrm{L} = \mathrm{diag}(\nu) - \nu\nu^\top$ *where* $\nu = y^{\tau\varepsilon} \exp(\varepsilon A)$ *and* $\widetilde{\nu} = y^{\tau\varepsilon} \exp(\varepsilon \widetilde{A})$ *are follower's responses over* $\mathcal{B}$ *with* $\|A\|_\infty \leq B_A, \|\widetilde{A}\|_\infty \leq B_A$. *Then for any vector* $g \in \mathbb{R}^{|\mathcal{B}|}$, *we have*

$$\exp\left(2\varepsilon B_A\right) \cdot g^\top \mathrm{L} g \geq g^\top \mathrm{H} g \geq \exp\left(-2\varepsilon B_A\right) \cdot g^\top \mathrm{L} g. \tag{162}$$

*Proof.* The proof is similar to that of Proposition F.1 in Chen et al. (2023), with the main difference due to the divergence term in the follower's response model.

Note that $\exp\left(-2\varepsilon B_A\right) \leq \widetilde{\nu}(b)/\nu(b) \leq \exp\left(2\varepsilon B_A\right)$ for any $b \in \mathcal{B}$. Let $\mathbb{E}^\nu$ and $\mathrm{Var}^\nu$ denote the expectation and variance under distribution $\nu$. Then we have

$$g^\top \mathrm{L} g = \mathrm{Var}^\nu[g(b)] = \mathbb{E}^\nu\left[\left(g(b) - \mathbb{E}^\nu[g(b)]\right)^2\right],$$

$$g^\top \mathrm{H} g = \mathrm{Var}^{\widetilde{\nu}}[g(b)] = \mathbb{E}^{\widetilde{\nu}}\left[\left(g(b) - \mathbb{E}^{\widetilde{\nu}}[g(b)]\right)^2\right].$$

By direct computation, we have

$$\exp\left(-2\varepsilon B_A\right) \cdot \mathbb{E}^{\widetilde{\nu}}\left[\left(g(b) - \mathbb{E}^{\widetilde{\nu}}[g(b)]\right)^2\right] \leq \exp\left(-2\varepsilon B_A\right) \cdot \mathbb{E}^{\widetilde{\nu}}\left[\left(g(b) - \mathbb{E}^\nu[g(b)]\right)^2\right]$$

$$\leq \mathbb{E}^\nu\left[\left(g(b) - \mathbb{E}^\nu[g(b)]\right)^2\right] = g^\top \mathrm{L} g$$

where the first inequality is true because changing $\mathbb{E}^{\widetilde{\nu}}[g(b)]$ to $\mathbb{E}^\nu[g(b)]$ incurs additional bias, and the second inequality is true because $\widetilde{\nu}(b)/\nu(b) \leq \exp\left(2\varepsilon B_A\right)$. Similarly, we have

$$g^\top \mathrm{L} g = \mathbb{E}^\nu\left[\left(g(b) - \mathbb{E}^\nu[g(b)]\right)^2\right] \leq \mathbb{E}^\nu\left[\left(g(b) - \mathbb{E}^{\widetilde{\nu}}[g(b)]\right)^2\right] \leq \exp\left(2\varepsilon B_A\right) \cdot \mathbb{E}^{\widetilde{\nu}}\left[\left(g(b) - \mathbb{E}^{\widetilde{\nu}}[g(b)]\right)^2\right].$$

Therefore, we conclude that equation 162 holds. $\qquad\square$

**Lemma J.3** (Errors of response models, further). *Under the setting of Theorem J.1, we assume the reward function of the follower is a linear function of $\phi$. Let $\theta^*$ be the parameter of the true reward function and let $\widetilde{\theta} \in \Theta$ be another parameter. Fix any policy pair $(\pi, y)$. We define a matrix $\Sigma^{\pi, y, \theta}$ as $\mathrm{Cov}^{\pi, y, \theta}[\phi^\pi(b)]$, i.e.,*

$$\Sigma^{\pi, y, \theta} = \mathbb{E}\big[\big(\phi^\pi(b) - \mathbb{E}[\phi^\pi(b)]\big)\big(\phi^\pi(b) - \mathbb{E}[\phi^\pi(b)]\big)^\top\big], \tag{163}$$

*where the expectation is taken with respect to $\nu_\theta^{\pi, y}(\cdot)$.*

*We define the univariate function $f_\sigma$ for $x \geq 0$ and $\sigma \in \{\eta, \varepsilon\}$ as*

$$f_\sigma(x) = \sigma x + C_\sigma^{(3)} x^2. \tag{164}$$

*Then, for the TV distance between $\nu_{\theta^*}^{\pi, y}$ and $\nu_{\widehat{\theta}}^{\pi, y}$, we have*

$$\mathrm{TV}\left(\nu_{\theta^*}^{\pi, y}, \nu_{\widetilde{\theta}}^{\pi, y}\right) \leq \min\left\{f_\varepsilon\left(\sqrt{\mathrm{Tr}\left(\Psi^\dagger \Sigma^{\pi, y, \widetilde{\theta}}\right)} \cdot \left\|\theta^* - \widetilde{\theta}\right\|_\Psi\right), f_\varepsilon\left(\sqrt{\mathrm{Tr}\left(\Psi^\dagger \Sigma^{\pi, y, \theta^*}\right)} \cdot \left\|\theta^* - \widetilde{\theta}\right\|_\Psi\right)\right\}, \tag{165}$$

*and for the TV distance between $\mu_{\theta^*}^\pi$ and $\mu_{\widehat{\theta}}^\pi$, we have*

$$\mathrm{TV}(\mu_{\theta^*}^\pi, \mu_{\widehat{\theta}}^\pi) \leq C_{\mathrm{QF}} \cdot \min\left\{f_\eta\left(\sqrt{\mathrm{Tr}\left(\Psi^\dagger \Sigma^{\pi, y, \widetilde{\theta}}\right)} \cdot \left\|\theta^* - \widetilde{\theta}\right\|_\Psi\right), f_\eta\left(\sqrt{\mathrm{Tr}\left(\Psi^\dagger \Sigma^{\pi, y, \theta^*}\right)} \cdot \left\|\theta^* - \widetilde{\theta}\right\|_\Psi\right)\right\}, \tag{166}$$

*where $\Psi \in \mathbb{S}_+^d$ can be any fixed nonnegative definite matrix, $\Psi^\dagger$ is the pseudo-inverse of $\Psi$.*

*Also recall that $C_{\mathrm{QF}}$ is defined in equation 184. By defining*

$$C_{\eta/\varepsilon} := \max\left\{\frac{C_\eta^{(3)}}{C_\varepsilon^{(3)}}, \frac{\eta}{\varepsilon}\right\}, \tag{167}$$

*we have that, for any $x \geq 0$,*

$$f_\eta(x) \leq C_{\eta/\varepsilon} \cdot f_\varepsilon(x),$$

*and hence we further have*

$$\mathrm{TV}(\mu_{\theta^*}^\pi, \mu_{\widehat{\theta}}^\pi) \leq C_{\mathrm{QF}} C_{\eta/\varepsilon} \min\left\{f_\varepsilon\left(\sqrt{\mathrm{Tr}\left(\Psi^\dagger \Sigma^{\pi, y, \widetilde{\theta}}\right)} \cdot \left\|\theta^* - \widetilde{\theta}\right\|_\Psi\right), f_\varepsilon\left(\sqrt{\mathrm{Tr}\left(\Psi^\dagger \Sigma^{\pi, y, \theta^*}\right)} \cdot \left\|\theta^* - \widetilde{\theta}\right\|_\Psi\right)\right\}. \tag{168}$$

*This implies that the upper bounds on $\mathrm{TV}\left(\nu_{\theta^*}^{\pi, y}, \nu_{\widetilde{\theta}}^{\pi, y}\right)$ and $\mathrm{TV}(\mu_{\theta^*}^\pi, \mu_{\widehat{\theta}}^\pi)$ are the same up to only a constant factor $C_{\mathrm{QF}} \cdot C_{\eta/\varepsilon}$.*

*Proof.* The proof is similar to that of Corollary B.4 in Chen et al. (2023), with the main difference due to the divergence term in the follower's response model. Note that $\widetilde{Q}$ and $Q$ in Theorem J.1 correspond to $\widetilde{r}^\pi = \langle \phi^\pi, \widetilde{\theta}\rangle$a and $r^\pi = \langle \phi^\pi, \theta^*\rangle$a, respectively. Let $\mathbb{E}$ denote the expectation taken with respect to $\nu_{\theta^*}^{\pi, y}(\cdot)$. Then we have

$$\mathbb{E}\left[\Big(\big(\widetilde{r}^\pi(b) - r^\pi(b)\big) - \mathbb{E}\left[\widetilde{r}^\pi(b) - r^\pi(b)\right]\Big)^2\right] = \left\|\theta^* - \widetilde{\theta}\right\|_{\Sigma^{\pi, y, \theta^*}}^2$$

$$= \mathrm{Tr}\left(\Sigma^{\pi, y, \theta^*} \cdot (\theta^* - \widetilde{\theta})(\theta^* - \widetilde{\theta})^\top\right) \leq \mathrm{Tr}\left(\Psi^\dagger \Sigma^{\pi, y, \theta^*}\right) \cdot \left\|\theta^* - \widetilde{\theta}\right\|_\Psi^2, \tag{169}$$

where $\Psi^\dagger$ is the pseudo-inverse of $\Psi$. Moreover, by Jensen's inequality, we have

$$\mathbb{E}\left[\left|\big(\widetilde{r}^\pi(b) - r^\pi(b)\big) - \mathbb{E}\left[\widetilde{r}^\pi(b) - r^\pi(b)\right]\right|\right] \leq \sqrt{\left(\mathbb{E}\left[\Big(\big(\widetilde{r}^\pi(b) - r^\pi(b)\big) - \mathbb{E}\left[\widetilde{r}^\pi(b) - r^\pi(b)\right]\Big)^2\right]\right)}$$

$$\leq \sqrt{\mathrm{Tr}\left(\Psi^\dagger \Sigma^{\pi, y, \theta^*}\right)} \cdot \left\|\theta^* - \widetilde{\theta}\right\|_\Psi. \tag{170}$$

Finally, combining equation 169, equation 170, and Theorem J.1, we conclude the proof for $\mathrm{TV}\left(\nu_{\theta^*}^{\pi, y}, \nu_{\widetilde{\theta}}^{\pi, y}\right)$. We note that a similar result also holds if we swap $\theta^*$ and $\widetilde{\theta}$.

Similarly, for $\text{TV}(\mu_{\theta^*}^{\pi}, \mu_{\widetilde{\theta}}^{\pi})$, by plugging the above results into equation 154 in Theorem J.1, we have

$$\text{TV}(\mu_{\theta^*}^{\pi}, \mu_{\widetilde{\theta}}^{\pi}) \leq C_{\text{QF}} \cdot \min \left\{ f_{\eta}\left(\sqrt{\text{Tr}\left(\Psi^{\dagger} \Sigma^{\pi, y, \widetilde{\theta}}\right)} \cdot \left\|\theta^* - \widetilde{\theta}\right\|_{\Psi}\right), f_{\eta}\left(\sqrt{\text{Tr}\left(\Psi^{\dagger} \Sigma^{\pi, y, \theta^*}\right)} \cdot \left\|\theta^* - \widetilde{\theta}\right\|_{\Psi}\right) \right\},$$

which completes the proof. $\qquad\square$

The following lemma establishes upper bounds and lower bounds on the TV distance between $\nu_{\theta}^{\pi, y}$ and $\nu_{\widetilde{\theta}}^{\pi, y}$, for any $\theta, \widetilde{\theta} \in \Theta$. Since we fix the leader's policy pair $\pi, y$ in the analysis, we drop the superscript $\pi, y$ for simplicity, and denote $\nu$ and $\widetilde{\nu}$ as $\nu_{\theta}^{\pi, y}$ and $\nu_{\theta'}^{\pi, y}$, respectively.

**Lemma J.4** (TV-distance between follower's responses). *Let $\nu, \widetilde{\nu} \in \Delta(\mathcal{B})$ be defined in equation 152, where $\widetilde{A} = \widetilde{Q} - \widetilde{V}$ and $A = Q - V$. The TV distance between $\nu$ and $\widetilde{\nu}$ enjoys the following upper bound:*

$$\text{TV}(\nu, \widetilde{\nu}) \leq \varepsilon \cdot \left\langle \nu, \left|\widetilde{Q} - Q - \xi \cdot \mathbf{1}\right| + \frac{\varepsilon}{2} \exp\left(\varepsilon |\widetilde{Q} - Q - \xi \cdot \mathbf{1}|\right) \cdot \left(\widetilde{Q} - Q - \xi \cdot \mathbf{1}\right)^2 \right\rangle_{\mathcal{B}}, \quad \forall \xi \in \mathbb{R}.$$

*Here $\mathbf{1} \in \mathbb{R}^{|\mathcal{B}|}$ is the all-one vector. In particular, setting $\xi = \widetilde{V} - V$, we have*

$$\text{TV}(\nu, \widetilde{\nu}) \leq \varepsilon \cdot \left\langle \nu, |\widetilde{A} - A| + \frac{\varepsilon}{2} \exp\left(\varepsilon |\widetilde{A} - A|\right) \cdot (\widetilde{A} - A)^2 \right\rangle_{\mathcal{B}}. \tag{171}$$

*Note that when $\tau = 0$ (and so $\varepsilon = \eta$), the follower's response $\nu_{\theta}^{\pi, y}$ is equal to the corresponding quantal response $\mu_{\theta}^{\pi}$. Hence, the TV distance between the corresponding quantal responses $\mu = \mu_{\theta^*}^{\pi}$ and $\widetilde{\mu} = \mu_{\widetilde{\theta}}^{\pi}$ enjoys the following upper bound for any $\xi \in \mathbb{R}$:*

$$\text{TV}(\mu_{\theta^*}^{\pi}, \mu_{\widetilde{\theta}}^{\pi}) \leq \eta \cdot \left\langle \mu_{\theta^*}^{\pi}, \left|\widetilde{Q} - Q - \xi \cdot \mathbf{1}\right| + \frac{\eta}{2} \exp\left(\eta |\widetilde{Q} - Q - \xi \cdot \mathbf{1}|\right) \cdot \left(\widetilde{Q} - Q - \xi \cdot \mathbf{1}\right)^2 \right\rangle_{\mathcal{B}}.$$

*By setting $\xi = V_{\widetilde{\theta}}^{\pi, y} - V_{\theta^*}^{\pi, y}$, we have*

$$\text{TV}(\mu_{\theta^*}^{\pi}, \mu_{\widetilde{\theta}}^{\pi}) \leq \eta \cdot \left\langle \mu_{\theta^*}^{\pi}, |A_{\widetilde{\theta}}^{\pi, y} - A_{\theta^*}^{\pi, y}| + \frac{\eta}{2} \exp\left(\eta |A_{\widetilde{\theta}}^{\pi, y} - A_{\theta^*}^{\pi, y}|\right) \cdot (A_{\widetilde{\theta}}^{\pi, y} - A_{\theta^*}^{\pi, y})^2 \right\rangle_{\mathcal{B}}$$

$$\leq \eta \cdot \left\langle \mu_{\theta^*}^{\pi}, |\widetilde{A} - A| + \frac{\eta}{2} \exp\left(\eta |\widetilde{A} - A|\right) \cdot (\widetilde{A} - A)^2 \right\rangle_{\mathcal{B}}. \tag{172}$$

*Besides, the following lower bounds hold:*

$$\text{TV}(\nu, \widetilde{\nu}) \geq \frac{\varepsilon}{2(1 + 2\varepsilon B_A)} \cdot \left\langle \nu, |\widetilde{A} - A| \right\rangle_{\mathcal{B}}, \tag{173}$$

$$\text{TV}(\nu, \widetilde{\nu}) \geq \frac{1}{2} \left\langle \nu, \varepsilon \exp\left(-\varepsilon |\widetilde{A} - A|\right) \cdot |\widetilde{A} - A| \right\rangle_{\mathcal{B}}. \tag{174}$$

*Here $B_A$ is an upper bound on $\max\{\|A\|_{\infty}, \|\widetilde{A}\|_{\infty}\}$, which, for example, can be set as in equation 204.*

*Proof.* The proof is similar to that of Lemma G.1 in Chen et al. (2023), with the main difference due to the divergence term in the follower's response model. In this proof, to simplify the notation, we omit the subscript in $\langle \cdot, \cdot \rangle_{\mathcal{B}}$ without causing confusion. We first prove the lower bounds. Using the relation between TV distance and $\ell_1$-norm, we have

$$\text{TV}(\nu, \widetilde{\nu}) = \frac{1}{2} \|\nu - \widetilde{\nu}\|_1 = \frac{1}{2} \left\langle \nu, \left|\mathbf{1} - \frac{\widetilde{\nu}}{\nu}\right| \right\rangle = \frac{1}{2} \left\langle \nu, \left|\mathbf{1} - \exp\left(\varepsilon(\widetilde{A} - A)\right)\right| \right\rangle. \tag{175}$$

We note that $|\exp(x) - 1| \geq (1 - \exp(-B))/B \cdot |x|$ holds for all $x \in [-B, B]$. Thus, for any $b \in \mathcal{B}$, we have

$$\left|1 - \exp\left(\varepsilon\left(\widetilde{A}(b) - A(b)\right)\right)\right| \geq \frac{1 - \exp(-2\varepsilon B_A)}{2\varepsilon B_A} \cdot \varepsilon \cdot |\widetilde{A}(b) - A(b)|. \tag{176}$$

Hence, combining equation 175 and equation 176, we have

$$\frac{1}{2} \cdot \left\langle \nu, \left|\mathbf{1} - \exp\left(\varepsilon(\widetilde{A} - A)\right)\right| \right\rangle \geq \frac{\varepsilon}{2} \cdot \frac{1 - \exp\left(-2\varepsilon B_A\right)}{2\varepsilon B_A} \cdot \left\langle \nu, |\widetilde{A} - A| \right\rangle$$

$$\geq \frac{\varepsilon}{2} \cdot \frac{1}{1 + 2\varepsilon B_A} \cdot \left\langle \nu, |\widetilde{A} - A| \right\rangle.$$

where the last inequality holds by noting that

$$1 - \exp(-x) = \frac{\exp(x) - 1}{\exp(x) - 1 + 1} \geq \frac{x}{1 + x}, \quad \forall x > 0.$$

Therefore, we establish equation 173.

Meanwhile, we note that $|\exp(x) - 1| \geq \exp(-|x|) \cdot x$ holds for all $x \in \mathbb{R}$, which indicates that

$$\frac{1}{2} \cdot \langle \nu, | \mathbf{1} - \exp\left(\varepsilon(\widetilde{A} - A)\right)| \rangle \geq \frac{1}{2} \langle \nu, \varepsilon \exp\left(-\varepsilon|\widetilde{A} - A|\right) \cdot |\widetilde{A} - A| \rangle.$$

Thus, we establish equation 174.

It remains to establish an upper bound on the right hand side of equation 175. Note that $A = Q - V$ and $\widetilde{A} = \widetilde{Q} - \widetilde{V}$. Then, we have

$$\frac{1}{2}\langle \nu, | \mathbf{1} - \exp\left(\varepsilon(\widetilde{A} - A)\right)| \rangle = \frac{1}{2}\langle \nu, | \mathbf{1} - \exp\left(\varepsilon(\widetilde{Q} - Q - \widetilde{V} + V)\right)| \rangle$$

$$\leq \frac{1}{2}\langle \nu, | \mathbf{1} - \exp\left(\varepsilon(\widetilde{Q} - Q)\right)| \rangle + \frac{1}{2}\langle \nu, | \exp\left(\varepsilon(\widetilde{Q} - Q)\right) - \exp\left(\varepsilon(\widetilde{Q} - Q - \widetilde{V} + V)\right)| \rangle$$

$$= \frac{1}{2}\langle \nu, | \mathbf{1} - \exp\left(\varepsilon(\widetilde{Q} - Q)\right)| \rangle + \frac{1}{2}\langle y^{\tau\varepsilon}, | \exp\left(\varepsilon(\widetilde{Q} - V)\right) - \exp\left(\varepsilon(\widetilde{Q} - \widetilde{V})\right)| \rangle,$$

where the inequality is just a split of terms and the last equality holds by the definition $\nu = y^{\tau\varepsilon}\exp\left(\varepsilon(Q - V)\right)$. With a slight abuse of notation, we let $y^{\tau\varepsilon} \in \mathbb{R}^{|\mathcal{B}|}$ be a vector with $(y^{\tau\varepsilon})(b) = (y(b))^{\tau\varepsilon}$ for $\forall b \in \mathcal{B}$. Using the equality $\langle y^{\tau\varepsilon}, \exp(\varepsilon\widetilde{Q}) \rangle = \exp(\varepsilon\widetilde{V})$ by definition of $\widetilde{V}$, and noting that $V$ and $\widetilde{V}$ does not depend on $b \in \mathcal{B}$, we further obtain

$$\frac{1}{2}\langle \nu, | \mathbf{1} - \exp\left(\varepsilon(\widetilde{A} - A)\right)| \rangle$$

$$\leq \frac{1}{2}\langle \nu, | \mathbf{1} - \exp\left(\varepsilon(\widetilde{Q} - Q)\right)| \rangle + \frac{1}{2}\langle \exp\left(\varepsilon\widetilde{Q}\right), y^{\tau\varepsilon} | \exp\left(\varepsilon(-V)\right) - \exp\left(\varepsilon(-\widetilde{V})\right)| \rangle$$

$$= \frac{1}{2}\langle \nu, | \mathbf{1} - \exp\left(\varepsilon(\widetilde{Q} - Q)\right)| \rangle + \frac{1}{2}\exp\left(\varepsilon\widetilde{V}\right) \cdot | \exp\left(-\varepsilon V\right) - \exp\left(-\varepsilon\widetilde{V}\right)|, \tag{177}$$

Moreover, the second term on the right hand side of equation 177 can be bounded by the same trick,

$$\frac{1}{2}\exp\left(\varepsilon\widetilde{V}\right) \cdot | \exp\left(-\varepsilon V\right) - \exp\left(-\varepsilon\widetilde{V}\right)| = \frac{1}{2}\exp\left(-\varepsilon V\right) \cdot | \exp\left(\varepsilon\widetilde{V}\right) - \exp\left(\varepsilon V\right)|$$

$$= \frac{1}{2}\exp\left(-\varepsilon V\right) \cdot |\langle y^{\tau\varepsilon}, \exp\left(\varepsilon\widetilde{Q}\right) - \exp\left(\varepsilon Q\right) \rangle| = \frac{1}{2}|\langle \nu, \exp\left(\varepsilon\widetilde{Q} - \varepsilon Q\right) - \mathbf{1} \rangle|, \tag{178}$$

where the last equality uses the fact that $\nu = y^{\tau\varepsilon}\exp(\varepsilon Q - \varepsilon V)$. Plugging equation 178 into equation 177 gives

$$\frac{1}{2}\langle \nu, | \mathbf{1} - \exp\left(\varepsilon(\widetilde{A} - A)\right)| \rangle \leq \langle \nu, | \mathbf{1} - \exp\left(\varepsilon(\widetilde{Q} - Q)\right)| \rangle$$

$$\leq \varepsilon \cdot \langle \nu, |\widetilde{Q} - Q| + \frac{\varepsilon}{2} \cdot \exp(\varepsilon|\widetilde{Q} - Q|)|\widetilde{Q} - Q|^2 \rangle,$$

where the last inequality holds by a Taylor expansion of $\exp(x) - 1$ at $x = 0$. Moreover, note that adding a constant shift $\xi \mathbf{1}$ to both $\widetilde{Q} - Q$ does not change $\nu$ nor $\widetilde{\nu}$. Hence, we finish the proof of the upper bound. $\qquad \square$

**Lemma J.5** (Difference in A-functions). *For any $b \in \mathcal{B}$, we have*

$$A(b) - \widetilde{A}(b) = (\mathbb{E}_b - \mathbb{E})\left[(r^\pi - \widetilde{r}^\pi)(b)\right] + \varepsilon^{-1}D(\nu||\widetilde{\nu}),$$

*where $D(\nu||\widetilde{\nu}) = \varepsilon \cdot \mathbb{E}_\nu\left[A - \widetilde{A}\right]$, and $\widetilde{r}$ is an estimate of $r$ that generates $\widetilde{\nu}$.*

*Proof.* The proof is similar to that of Lemma G.4 in Chen et al. (2023), with the main difference due to the divergence term in the follower's response model. To simplify the notation, we omit $b$ in the functions and omit the subscript in $\langle \cdot, \cdot_{\mathcal{B}} \rangle$.

We first show that $D(\nu||\widetilde{\nu}) = \varepsilon \cdot \mathbb{E}_\nu\big[A - \widetilde{A}\big]$.

$$D(\nu||\widetilde{\nu}) = \langle \nu, \log \nu \rangle - \langle \nu, \log \widetilde{\nu} \rangle = \langle \nu, \log \frac{\nu}{\widetilde{\nu}} \rangle = \langle \nu, \varepsilon \cdot (A - \widetilde{A}) \rangle = \varepsilon \cdot \mathbb{E}_\nu\big[A - \widetilde{A}\big]. \qquad (179)$$

Fix the policy pair $(\pi, y)$, and by the optimality of $\nu$ and $\widetilde{\nu}$, we can write the difference of V-functions as follows:

$$V - \widetilde{V} = \langle \nu, Q \rangle + \eta^{-1}\mathcal{H}(\nu) - \tau D(\nu||y) - \langle \widetilde{\nu}, \widetilde{Q} \rangle - \eta^{-1}\mathcal{H}(\widetilde{\nu}) + \tau D(\widetilde{\nu}||y)$$
$$= \langle \nu, Q - \widetilde{Q} \rangle + \langle \nu - \widetilde{\nu}, \widetilde{Q} \rangle + \eta^{-1}\mathcal{H}(\nu) - \tau D(\nu||y) - \eta^{-1}\mathcal{H}(\widetilde{\nu}) + \tau D(\widetilde{\nu}||y),$$

where

$$\eta^{-1}\mathcal{H}(\nu) - \tau D(\nu||y) = -\eta^{-1}\langle \nu, \log \nu \rangle - \tau \langle \nu, \log \nu \rangle + \tau \langle \nu, \log y \rangle$$
$$= -(\eta^{-1} + \tau)\langle \nu, \log \nu \rangle + \tau \langle \nu, \log y \rangle = -(\eta^{-1} + \tau)\langle \nu, \tau\varepsilon \log y + \varepsilon(Q - V) \rangle + \tau \langle \nu, \log y \rangle$$
$$= -\varepsilon(\eta^{-1} + \tau)\langle \nu, Q - V \rangle + \tau(1 - \varepsilon(\eta^{-1} + \tau))\langle \nu, \log y \rangle = -\langle \nu, Q - V \rangle$$

and similarly

$$\eta^{-1}\mathcal{H}(\widetilde{\nu}) - \tau D(\widetilde{\nu}||y) = -\left\langle \widetilde{\nu}, \widetilde{Q} - \widetilde{V} \right\rangle.$$

So we have

$$V - \widetilde{V} = \langle \nu, Q - \widetilde{Q} \rangle + \langle \nu - \widetilde{\nu}, \widetilde{Q} \rangle - \langle \nu, Q - V \rangle + \langle \widetilde{\nu}, \widetilde{Q} - \widetilde{V} \rangle.$$

Note that here both $\widetilde{V}$ and $V$ are real numbers. Then, by direct calculation, we have

$$\langle \nu - \widetilde{\nu}, \widetilde{Q} \rangle - \langle \nu, Q - V \rangle + \langle \widetilde{\nu}, \widetilde{Q} - \widetilde{V} \rangle = -\langle \nu, Q - V - (\widetilde{Q} - \widetilde{V}) \rangle,$$

where we use the fact that $\langle \widetilde{\nu} - \nu, \widetilde{V} \rangle = 0$ since here $\widetilde{V}$ represents a scalar multiple of $\mathbf{1}$. Hence, we can write $V - \widetilde{V}$ as

$$V - \widetilde{V} = \langle \nu, Q - \widetilde{Q} \rangle - \langle \nu, Q - V - (\widetilde{Q} - \widetilde{V}) \rangle$$
$$= \langle \nu, Q - \widetilde{Q} \rangle - \langle \nu, A - \widetilde{A} \rangle = \langle \nu, Q - \widetilde{Q} \rangle - \varepsilon^{-1}D(\nu||\widetilde{\nu}), \qquad (180)$$

where the equality holds by noting that $D(\nu||\widetilde{\nu}) = \varepsilon\langle \nu, A - \widetilde{A} \rangle$. Thus, by equation 180, For the A-function, we have

$$A - \widetilde{A} = Q - V - (\widetilde{Q} - \widetilde{V}) = (\mathbb{E}_b - \mathbb{E})\big[Q - \widetilde{Q}\big] + \varepsilon^{-1}D(\nu||\widetilde{\nu}). \qquad (181)$$

Meanwhile, by $Q = r^\pi$, we have

$$Q - \widetilde{Q} = \mathbb{E}_b\big[r^\pi - \widetilde{Q}\big]. \qquad (182)$$

Then, we have

$$A - \widetilde{A} = (\mathbb{E}_b - \mathbb{E})\big[r^\pi - \widetilde{Q}\big] + \varepsilon^{-1}D(\nu||\widetilde{\nu}),$$

which completes the proof. $\qquad\square$

The next lemma establishes a lower bound on the Hellinger distance between $\widetilde{\nu}$ and $\nu$.

**Lemma J.6** (Hellinger distance between quantal responses). *Let $\nu$ and $\widetilde{\nu}$ be defined in equation 152. We let $D_{\mathrm{H}}(\cdot, \cdot)$ denote the Hellinger distance between two probability distributions. Then, we have*

$$D_{\mathrm{H}}^2(\nu, \widetilde{\nu}) \geq \frac{\varepsilon^2}{8(1 + \varepsilon B_A)^2} \cdot \big\langle \nu, (\widetilde{A} - A)^2 \big\rangle_{\mathcal{B}}.$$

*Proof.* The proof is similar to that of Lemma G.2 in Chen et al. (2023), with the main difference due to the divergence term in the follower's response model. In this proof, to simplify the notation, we omit the subscript in $\langle \cdot, \cdot a_{\mathcal{B}} \rangle$. By the definition of the Hellinger distance, we have

$$D_{\mathrm{H}}^2(\nu, \widetilde{\nu}) = \frac{1}{2}\big\langle \nu, (1 - \sqrt{\widetilde{\nu}/\nu})^2 \big\rangle = \frac{1}{2}\big\langle \nu, |\mathbf{1} - \exp(\varepsilon/2 \cdot (\widetilde{A} - A))|^2 \big\rangle$$
$$\geq \frac{1}{8}\left(\frac{1 - \exp(-\varepsilon B_A)}{B_A}\right)^2 \cdot \big\langle \nu, (\widetilde{A} - A)^2 \big\rangle \geq \frac{\varepsilon^2}{8(1 + \varepsilon B_A)^2} \cdot \big\langle \nu, (\widetilde{A} - A)^2 \big\rangle.$$

where the first inequality follows from $|\exp(x) - 1| \geq (1 - \exp(-B))|x|/B$ for any $|x| \leq B$, and the last inequality holds by noting that

$$1 - \exp(-x) = \frac{\exp(x) - 1}{\exp(x) - 1 + 1} \geq \frac{x}{1 + x}, \quad \forall x > 0,$$

which concludes the proof. $\qquad\square$

## K   OTHER THEORETICAL RESULTS FOR LEADER-FOLLOWER GAMES WITH POLICY RECOMMENDATION

**Lemma K.1.** *Fix the parameter $\theta \in \Theta$ and the policy pair $(\pi, y)$ where $\pi \in \Pi$ and $y \in \mathcal{Y}_{\mathrm{QR}}$. Then, the follower's response $\nu_\theta^{\pi,y}$ can upper bound the corresponding quantal response $\mu_\theta^\pi$ up to a constant. Specially, for any $b \in \mathcal{B}$, we have*

$$\mu_\theta^\pi(b) \leq C_{\mathrm{QF}} \cdot \nu_\theta^{\pi,y}(b), \tag{183}$$

*where we define the constant factor*

$$C_{\mathrm{QF}} := \exp(2\eta) \cdot |\mathcal{B}|^{\frac{\tau}{\tau + \eta^{-1}}} \tag{184}$$

*Proof.* Recall that the follower's response defined in equation 5 can be written as

$$\nu_\theta^{\pi,y}(b) = \frac{[y(b)]^{\frac{\tau}{\tau + \eta^{-1}}} \cdot \exp\left\{\frac{1}{\tau + \eta^{-1}} \cdot Q_\theta^\pi(b)\right\}}{\sum_{b' \in \mathcal{B}} [y(b')]^{\frac{\tau}{\tau + \eta^{-1}}} \cdot \exp\left\{\frac{1}{\tau + \eta^{-1}} \cdot Q_\theta^\pi(b')\right\}}$$

$$= \left(\sum_{b' \in \mathcal{B}} \left[\frac{y(b')}{y(b)}\right]^{\frac{\tau}{\tau + \eta^{-1}}} \cdot \exp\left\{\frac{1}{\tau + \eta^{-1}} \cdot (Q_\theta^\pi(b') - Q_\theta^\pi(b))\right\}\right)^{-1}, \tag{185}$$

for any $b \in \mathcal{B}$. Taking $\tau = 0$ leads to the corresponding quantal response $\mu_\theta^\pi$

$$\mu_\theta^\pi(b) = \left(\sum_{b' \in \mathcal{B}} \exp\left\{\eta \cdot (Q_\theta^\pi(b') - Q_\theta^\pi(b))\right\}\right)^{-1}. \tag{186}$$

So we have

$$\frac{\mu_\theta^\pi(b)}{\nu_\theta^{\pi,y}(b)} = \frac{\sum_{b' \in \mathcal{B}} \left[\frac{y(b')}{y(b)}\right]^{\frac{\tau}{\tau + \eta^{-1}}} \cdot \exp\left\{\frac{1}{\tau + \eta^{-1}} \cdot (Q_\theta^\pi(b') - Q_\theta^\pi(b))\right\}}{\sum_{b' \in \mathcal{B}} \exp\left\{\eta \cdot (Q_\theta^\pi(b') - Q_\theta^\pi(b))\right\}}. \tag{187}$$

Since $y \in \mathcal{Y}_{\mathrm{QR}}$, by Theorem C.3 we have that, for any $b \in \mathcal{B}$,

$$y(b) \geq \mu_{\mathrm{L}} := |\mathcal{B}|^{-1} \cdot \exp(-\eta) > 0. \tag{188}$$

For the numerator of equation 187, we have

$$\sum_{b' \in \mathcal{B}} \left[\frac{y(b')}{y(b)}\right]^{\frac{\tau}{\tau + \eta^{-1}}} \exp\left\{\frac{1}{\tau + \eta^{-1}}(Q_\theta^\pi(b') - Q_\theta^\pi(b))\right\}$$

$$\leq \sum_{b' \in \mathcal{B}} \left[\frac{1}{\mu_{\mathrm{L}}}\right]^{\frac{\tau}{\tau + \eta^{-1}}} \exp\left\{\frac{1}{\tau + \eta^{-1}} \cdot 1\right\} = |\mathcal{B}| \cdot \mu_{\mathrm{L}}^{\frac{-\tau}{\tau + \eta^{-1}}} \cdot \exp\left\{\frac{1}{\tau + \eta^{-1}}\right\}, \tag{189}$$

where the inequality is by equation 188, $y(b') \leq 1$ and the fact that $Q_\theta^\pi(b') - Q_\theta^\pi(b) \leq 1$ for any $b, b' \in \mathcal{B}$. Similarly, for the denominator of equation 187, we have

$$\sum_{b' \in \mathcal{B}} \exp\left\{\eta \cdot (Q_\theta^\pi(b') - Q_\theta^\pi(b))\right\} \geq \sum_{b' \in \mathcal{B}} \exp\left\{\eta \cdot (-1)\right\} = |\mathcal{B}| \cdot \exp\left\{-\eta\right\}, \tag{190}$$

where the inequality is because $Q_\theta^\pi(b') - Q_\theta^\pi(b) \geq -1$ for any $b, b' \in \mathcal{B}$. Finally, combining equation 187, equation 189 and equation 190, we have

$$\frac{\mu_\theta^\pi(b)}{\nu_\theta^{\pi,y}(b)} \leq \frac{|\mathcal{B}| \cdot \mu_L^{\frac{-\tau}{\tau+\eta^{-1}}} \cdot \exp\left\{\frac{1}{\tau+\eta^{-1}}\right\}}{|\mathcal{B}| \cdot \exp\{-\eta\}} = \frac{\mu_L^{\frac{-\tau}{\tau+\eta^{-1}}} \cdot \exp\left\{\frac{1}{\tau+\eta^{-1}}\right\}}{\exp\{-\eta\}}$$

$$= \frac{|\mathcal{B}|^{\frac{\tau}{\tau+\eta^{-1}}} \cdot \exp(\eta)^{\frac{\tau}{\tau+\eta^{-1}}} \cdot \exp\left\{\frac{1}{\tau+\eta^{-1}}\right\}}{\exp\{-\eta\}} = \exp(2\eta) \cdot |\mathcal{B}|^{\frac{\tau}{\tau+\eta^{-1}}}, \quad (191)$$

where we use $\mu_L := \frac{1}{|\mathcal{B}|\cdot\exp(\eta)}$ in equation 188. We then immediately have

$$\mu_\theta^\pi(b) \leq \exp(2\eta) \cdot |\mathcal{B}|^{\frac{\tau}{\tau+\eta^{-1}}} \cdot \nu_\theta^{\pi,y}(b), \quad (192)$$

which completes the proof. $\qquad\square$

Note that by Theorem C.3 and Theorem C.4 we can easily derive such a constant that plays the same role as $C_{QF}$ in Theorem K.1. However, the dependency of $C_{QF}$ on $|\mathcal{B}|$ in Theorem K.1 is better than such a constant.

**Lemma K.2.** *Fix the policy pair $(\pi, y)$. Then, for any $\theta, \widetilde{\theta} \in \Theta$, we have*

$$|V_\theta^{\pi,y} - V_{\widetilde{\theta}}^{\pi,y}| \leq \|Q_\theta^\pi - Q_{\widetilde{\theta}}^\pi\|_\infty. \quad (193)$$

*Proof.* Since the policy pair $(\pi, y)$ is fixed, we denote $V = V_\theta^{\pi,y}$ and $\widetilde{V} = V_{\widetilde{\theta}}^{\pi,y}$. By definition and equation 3, we have

$$V = \max_{\nu'\in\Delta(\mathcal{B})}\left\{\langle\nu', Q\rangle_\mathcal{B} + \eta^{-1}\mathcal{H}(\nu') - \tau D(\nu'||y)\right\},$$
$$\widetilde{V} = \max_{\nu'\in\Delta(\mathcal{B})}\left\{\langle\nu', \widetilde{Q}\rangle_\mathcal{B} + \eta^{-1}\mathcal{H}(\nu') - \tau D(\nu'||y)\right\}, \quad (194)$$

where the maximizers are $\nu$ and $\widetilde{\nu}$, respectively. Then, by the optimality condition of equation 194, we have

$$V = \langle\nu, Q\rangle_\mathcal{B} + \eta^{-1}\mathcal{H}(\nu) - \tau D(\nu||y) \geq \langle\widetilde{\nu}, Q\rangle_\mathcal{B} + \eta^{-1}\mathcal{H}(\widetilde{\nu}) - \tau D(\widetilde{\nu}||y), \quad (195)$$
$$\widetilde{V} = \langle\widetilde{\nu}, \widetilde{Q}\rangle_\mathcal{B} + \eta^{-1}\mathcal{H}(\widetilde{\nu}) - \tau D(\widetilde{\nu}||y) \geq \langle\nu, \widetilde{Q}\rangle_\mathcal{B} + \eta^{-1}\mathcal{H}(\nu) - \tau D(\nu||y). \quad (196)$$

So we have

$$V - \widetilde{V} \leq \langle\nu, Q\rangle_\mathcal{B} + \eta^{-1}\mathcal{H}(\nu) - \tau D(\nu||y) - \langle\nu, \widetilde{Q}\rangle_\mathcal{B} - \eta^{-1}\mathcal{H}(\nu) + \tau D(\nu||y) = \langle\nu, Q - \widetilde{Q}\rangle_\mathcal{B}, \quad (197)$$

and

$$V - \widetilde{V} \geq \langle\widetilde{\nu}, Q\rangle_\mathcal{B} + \eta^{-1}\mathcal{H}(\widetilde{\nu}) - \tau D(\widetilde{\nu}||y) - \langle\widetilde{\nu}, \widetilde{Q}\rangle_\mathcal{B} - \eta^{-1}\mathcal{H}(\widetilde{\nu}) + \tau D(\widetilde{\nu}||y) = \langle\widetilde{\nu}, Q - \widetilde{Q}\rangle_\mathcal{B}. \quad (198)$$

Therefore, we have

$$|V - \widetilde{V}| \leq \max\left\{|\langle\nu, Q-\widetilde{Q}\rangle_\mathcal{B}|, |\langle\widetilde{\nu}, Q-\widetilde{Q}\rangle_\mathcal{B}|\right\} \leq \max\left\{\langle\nu, |Q-\widetilde{Q}|\rangle_\mathcal{B}, \langle\widetilde{\nu}, |Q-\widetilde{Q}|\rangle_\mathcal{B}\right\}$$
$$\leq \max\left\{\|Q-\widetilde{Q}\|_\infty, \|Q-\widetilde{Q}\|_\infty\right\} = \|Q-\widetilde{Q}\|_\infty, \quad (199)$$

which completes the proof. $\qquad\square$

**Lemma K.3.** *Let $p = \{p_i\}_{i\in[n]}$ be any probability distribution over $[n]$, and the constant $\alpha \in [0, 1)$. Then, we have*

$$1 \leq \sum_{i\in[n]} p_i^\alpha \leq n^{1-\alpha}. \quad (200)$$

*Furthermore, the lower bound can be achieved by a degenerate distribution at any $i \in [n]$, and the upper bound can be achieved by the uniform distribution over $[n]$.*

*Proof.* This can be simply proved by the method of Lagrange multipliers, and hence the proof is omitted here. □

**Lemma K.4** (Ranges of Q-, V-, and A- functions)**.** *For any fixed policy pair $(\pi, y)$, let $Q$, $V$ and $A$ denote the follower's Q-, V-, and A-functions, respectively. Then, for any $b \in \mathcal{B}$, we have*

$$0 \leq Q(b) \leq 1, \tag{201}$$

$$0 \leq V \leq 1 + \eta^{-1} \log |\mathcal{B}|, \tag{202}$$

$$-1 \leq A(b) \leq 1 + \eta^{-1} \log |\mathcal{B}|. \tag{203}$$

*Therefore,*

$$B_A := 1 + \eta^{-1} \log |\mathcal{B}| \tag{204}$$

*upper bounds the follower's Q-, V-, and A-functions, i.e.,*

$$\max\{\|Q\|_\infty, |V|, \|A\|_\infty\} \leq B_A. \tag{205}$$

*Proof.* Under the assumption of the reward functions, we have $0 \leq Q_\theta^\pi(b) \leq 1$ for any $b \in \mathcal{B}$, $\pi \in \Pi$ and $\theta \in \Theta$.

By the definition of $V_\theta^{\pi,y}$ in equation 4, we have

$$V_\theta^{\pi,y} = (\tau + \eta^{-1}) \log \left( \sum_{b \in \mathcal{B}} [y(b)]^{\frac{\tau}{\tau + \eta^{-1}}} \exp \left\{ \frac{1}{\tau + \eta^{-1}} \cdot Q_\theta^\pi(b) \right\} \right)$$

$$= \varepsilon^{-1} \log \left( \sum_{b \in \mathcal{B}} [y(b)]^{\tau \varepsilon} \exp \left\{ \varepsilon \cdot Q_\theta^\pi(b) \right\} \right).$$

For the upper bound of $V_\theta^{\pi,y}$, we have

$$V_\theta^{\pi,y} \leq \varepsilon^{-1} \log \left( \sum_{b \in \mathcal{B}} [y(b)]^{\tau \varepsilon} \exp \left\{ \varepsilon \right\} \right) \leq \varepsilon^{-1} \log \left( \exp \left\{ \varepsilon \right\} |\mathcal{B}|^{1 - \tau \varepsilon} \right)$$

$$= 1 + \frac{1 - \tau \varepsilon}{\varepsilon} \log(|\mathcal{B}|) = 1 + \eta^{-1} \log |\mathcal{B}|.$$

Here, the first inequality is by $Q_\theta^\pi(b) \leq 1$ for any $b \in \mathcal{B}$, $\pi \in \Pi$ and $\theta \in \Theta$. The second inequality is by using the second inequality in Theorem K.3 with $n = |\mathcal{B}|$, $p = y$ and $\alpha = \tau \varepsilon$.

For the lower bound of $V_\theta^{\pi,y}$, we have

$$V_\theta^{\pi,y} \geq \varepsilon^{-1} \log \left( \sum_{b \in \mathcal{B}} [y(b)]^{\tau \varepsilon} \exp \left\{ \varepsilon \cdot 0 \right\} \right) = \varepsilon^{-1} \log \left( \sum_{b \in \mathcal{B}} [y(b)]^{\tau \varepsilon} \right)$$

$$\geq \varepsilon^{-1} \log (1) = 0.$$

Here, the first inequality is by $0 \leq Q_\theta^\pi(b)$ for any $b \in \mathcal{B}$, $\pi \in \Pi$ and $\theta \in \Theta$. The second inequality is by using the first inequality in Theorem K.3 with $n = |\mathcal{B}|$, $p = y$ and $\alpha = \tau \varepsilon$.

Therefore, we have

$$0 \leq V_\theta^{\pi,y} \leq 1 + \eta^{-1} \log |\mathcal{B}|$$

and hence $|V_\theta^{\pi,y}| \leq 1 + \eta^{-1} \log |\mathcal{B}|$.

For $\|A\|_\infty$, we can simply have

$$\|A\|_\infty = \|Q - V\|_\infty \leq \|Q\|_\infty + \|V\|_\infty \leq 2 + \eta^{-1} \log |\mathcal{B}|.$$

However, we will show that we can obtain a sharper upper bound on $\|A\|_\infty$. By the definition of $V_\theta^{\pi,y}$ in equation 4, we have

$$A_\theta^{\pi,y}(b) = V_\theta^{\pi,y} - Q_\theta^\pi(b) = \varepsilon^{-1} \log \left( \sum_{b' \in \mathcal{B}} [y(b')]^{\tau \varepsilon} \exp \left\{ \varepsilon \cdot Q_\theta^\pi(b') \right\} \right) - \varepsilon^{-1} \log \left( \exp \left\{ \varepsilon \cdot Q_\theta^\pi(b) \right\} \right)$$

$$= \varepsilon^{-1} \log \left( \frac{\sum_{b' \in \mathcal{B}} [y(b')]^{\tau \varepsilon} \exp \left\{ \varepsilon \cdot Q_\theta^\pi(b') \right\}}{\exp \left\{ \varepsilon \cdot Q_\theta^\pi(b) \right\}} \right)$$

$$= \varepsilon^{-1} \log \left( \sum_{b' \in \mathcal{B}} [y(b')]^{\tau \varepsilon} \exp \left\{ \varepsilon \cdot (Q_\theta^\pi(b') - Q_\theta^\pi(b)) \right\} \right).$$

For the upper bound of $A_\theta^{\pi,y}(b)$, we have

$$A_\theta^{\pi,y}(b) = \varepsilon^{-1} \log \left( \sum_{b' \in \mathcal{B}} [y(b')]^{\tau \varepsilon} \exp \left\{ \varepsilon \cdot (Q_\theta^\pi(b') - Q_\theta^\pi(b)) \right\} \right)$$

$$\leq \varepsilon^{-1} \log \left( \sum_{b' \in \mathcal{B}} [y(b')]^{\tau \varepsilon} \exp \left\{ \varepsilon \cdot 1 \right\} \right) \leq \varepsilon^{-1} \log \left( \exp \left\{ \varepsilon \right\} |\mathcal{B}|^{1 - \tau \varepsilon} \right)$$

$$= 1 + \frac{1 - \tau \varepsilon}{\varepsilon} \log(|\mathcal{B}|) = 1 + \eta^{-1} \log |\mathcal{B}|.$$

Here, the first inequality is by $Q_\theta^\pi(b') - Q_\theta^\pi(b) \leq 1$ for any $b, b' \in \mathcal{B}$, $\pi \in \Pi$ and $\theta \in \Theta$. The second inequality is by using the second inequality in Theorem K.3 with $n = |\mathcal{B}|$, $p = y$ and $\alpha = \tau\varepsilon$.

For the lower bound of $A_\theta^{\pi,y}(b)$, we have

$$A_\theta^{\pi,y}(b) = \varepsilon^{-1} \log \left( \sum_{b' \in \mathcal{B}} [y(b')]^{\tau\varepsilon} \exp\left\{\varepsilon \cdot (Q_\theta^\pi(b') - Q_\theta^\pi(b))\right\}\right)$$
$$\geq \varepsilon^{-1} \log \left( \sum_{b' \in \mathcal{B}} [y(b')]^{\tau\varepsilon} \exp\left\{\varepsilon \cdot (-1)\right\}\right) \geq \varepsilon^{-1} \log\left(\exp\left\{-\varepsilon\right\} \cdot 1\right) = -1.$$

Here, the first inequality is by $Q_\theta^\pi(b') - Q_\theta^\pi(b) \geq -1$ for any $b, b' \in \mathcal{B}$, $\pi \in \Pi$ and $\theta \in \Theta$. The second inequality is by using the first inequality in Theorem K.3 with $n = |\mathcal{B}|$, $p = y$ and $\alpha = \tau\varepsilon$.

Therefore, for any $b \in \mathcal{B}$, we have

$$-1 \leq A_\theta^{\pi,y}(b) \leq 1 + \eta^{-1} \log|\mathcal{B}|,$$

and hence $\|A_\theta^{\pi,y}\|_\infty \leq \max\{|-1|, |1 + \eta^{-1}\log|\mathcal{B}||\} = 1 + \eta^{-1}\log|\mathcal{B}|$.

Therefore, we have shown that

$$B_A := 1 + \eta^{-1}\log|\mathcal{B}|$$

upper bounds the follower's Q-, V-, and A-functions, i.e.,

$$\max\{\|Q\|_\infty, |V|, \|A\|_\infty\} \leq B_A,$$

which completes the proof. $\qquad\square$

A remarkable observation from this result is that the upper bound $B_A := 1 + \eta^{-1}\log|\mathcal{B}|$ on the follower's Q-, V-, and A-functions does not depend on the temperature $\tau$ of the divergence term in the follower's response model. Moreover, this upper bound is half less than the upper bound (A.12) of Chen et al. (2023) when the length of horizon is $H = 1$.

The following Theorem K.5 characterizes how the follower's response policy $\nu_\theta^{\pi,y}$ changes by the change of the leader's recommendation policy $y$ which is a quantal response, given any fixed leader's policy $\pi$ and the parameter $\theta$ are fixed. The result aligns with the intuition that a small difference between $y_1$ and $y_2$ corresponds to a small difference between $\nu_\theta^{\pi,y_1}$ and $\nu_\theta^{\pi,y_2}$.

**Proposition K.5.** *Fix the leader's policy $\pi \in \Pi$ and the parameter $\theta \in \Theta$. Let the recommendation policies $y_1$ and $y_2$ be any two quantal responses, i.e., $y_1, y_2 \in \mathcal{Y}_{\mathrm{QR}}$, where $\mathcal{Y}_{\mathrm{QR}}$ is the class of all quantal responses as defined in equation 25. Then, there exists a constant $C > 0$ such that*

$$\mathrm{TV}(\nu_\theta^{\pi,y_1}, \nu_\theta^{\pi,y_2}) \leq C \cdot \left(\sqrt{\mathrm{TV}(y_1, y_2)} + \mathrm{TV}(y_1, y_2)\right). \tag{206}$$

*Specially, we can take*

$$C := \max\left\{\frac{\sqrt{2}}{2}\exp(\varepsilon(1 + B_A))\sqrt{8\tau\varepsilon\exp(\varepsilon)|\mathcal{B}|^{2-\tau\varepsilon}}\ ,\ \tau\varepsilon\exp(3\varepsilon)|\mathcal{B}|^{2-\tau\varepsilon}\right\},$$

*which is independent of $y_1, y_2, \pi, \theta$.*

*Proof.* By the expression of the follower's response $\nu_\theta^{\pi,y}$ as presented in equation 5, we have

$$\nu_\theta^{\pi,y}(b) = [y(b)]^{\frac{\tau}{\tau+\eta^{-1}}} \cdot \exp\left\{\frac{1}{\tau+\eta^{-1}} \cdot A_\theta^{\pi,y}(b)\right\}, \quad \text{where} \quad A_\theta^{\pi,y}(b) = Q_\theta^\pi(b) - V_\theta^{\pi,y}, \tag{207}$$

$$Q_\theta^\pi(b) = r_\theta^\pi(b) = \pi^\top B_\theta e_b, \tag{208}$$

$$V_\theta^{\pi,y} = (\tau + \eta^{-1}) \log\left(\sum_{b \in \mathcal{B}} [y(b)]^{\frac{\tau}{\tau+\eta^{-1}}} \exp\left\{\frac{1}{\tau+\eta^{-1}} \cdot Q_\theta^\pi(b)\right\}\right). \tag{209}$$

For simplicity of notation, we denote

$$g_\theta^\pi(b) := \exp\left\{\varepsilon \cdot Q_\theta^\pi(b)\right\} \in [1, \exp(\varepsilon)], \qquad h_\theta^\pi(y) := \exp\left\{\varepsilon \cdot V_\theta^{\pi,y}\right\} \in [1, \exp(\varepsilon B_A)],$$

and then we have the follower's response policy to be

$$
\nu_\theta^{\pi,y}(b) = \frac{[y(b)]^{\tau\varepsilon} \cdot \exp\left\{\varepsilon \cdot Q_\theta^\pi(b)\right\}}{\sum_{b'\in\mathcal{B}} [y(b')]^{\tau\varepsilon} \cdot \exp\left\{\varepsilon \cdot Q_\theta^\pi(b')\right\}}
$$
$$
= \frac{[y(b)]^{\tau\varepsilon} \cdot g_\theta^\pi(b')}{\sum_{b'\in\mathcal{B}} [y(b')]^{\tau\varepsilon} \cdot g_\theta^\pi(b')} = \frac{[y(b)]^{\tau\varepsilon} \cdot g_\theta^\pi(b)}{\exp\left\{\varepsilon \cdot V_\theta^{\pi,y}\right\}} = \frac{[y(b)]^{\tau\varepsilon} \cdot g_\theta^\pi(b)}{h_\theta^\pi(y)}. \tag{210}
$$

Then, we have

$$
|\nu_\theta^{\pi,y_1}(b) - \nu_\theta^{\pi,y_2}(b)| = \left| \frac{[y_1(b)]^{\tau\varepsilon} \cdot g_\theta^\pi(b)}{h_\theta^\pi(y_1)} - \frac{[y_2(b)]^{\tau\varepsilon} \cdot g_\theta^\pi(b)}{h_\theta^\pi(y_2)} \right|
$$
$$
= g_\theta^\pi(b) \cdot \left| \frac{[y_1(b)]^{\tau\varepsilon} \cdot h_\theta^\pi(y_2) - [y_2(b)]^{\tau\varepsilon} \cdot h_\theta^\pi(y_1)}{h_\theta^\pi(y_1) \cdot h_\theta^\pi(y_2)} \right|
$$
$$
= \frac{g_\theta^\pi(b)}{h_\theta^\pi(y_1) \cdot h_\theta^\pi(y_2)} \cdot |[y_1(b)]^{\tau\varepsilon} \cdot h_\theta^\pi(y_2) - [y_2(b)]^{\tau\varepsilon} \cdot h_\theta^\pi(y_1)|
$$
$$
\leq \exp(\varepsilon) \cdot |[y_1(b)]^{\tau\varepsilon} \cdot h_\theta^\pi(y_2) - [y_2(b)]^{\tau\varepsilon} \cdot h_\theta^\pi(y_1)| = \exp(\varepsilon) \cdot f_\theta^\pi(b; y_1, y_2), \tag{211}
$$

where we use the fact that $g_\theta^\pi(b) \in [1, \exp(\varepsilon)]$ and $h_\theta^\pi(y) \in [1, \exp(\varepsilon B_A)]$. Here we denote

$$
f_\theta^\pi(b; y_1, y_2) := |[y_1(b)]^{\tau\varepsilon} \cdot h_\theta^\pi(y_2) - [y_2(b)]^{\tau\varepsilon} \cdot h_\theta^\pi(y_1)|. \tag{212}
$$

Since $h_\theta^\pi(y) := \exp\left\{\varepsilon \cdot V_\theta^{\pi,y}\right\} = \sum_{b\in\mathcal{B}} [y(b)]^{\tau\varepsilon} \cdot g_\theta^\pi(b)$, we have

$$
|h_\theta^\pi(y_1) - h_\theta^\pi(y_2)| = \left| \sum_{b\in\mathcal{B}} [y_1(b)]^{\tau\varepsilon} \cdot g_\theta^\pi(b) - \sum_{b\in\mathcal{B}} [y_2(b)]^{\tau\varepsilon} \cdot g_\theta^\pi(b) \right| = \left| \sum_{b\in\mathcal{B}} g_\theta^\pi(b) \cdot ([y_1(b)]^{\tau\varepsilon} - [y_2(b)]^{\tau\varepsilon}) \right|
$$
$$
\leq \sum_{b\in\mathcal{B}} |g_\theta^\pi(b) \cdot ([y_1(b)]^{\tau\varepsilon} - [y_2(b)]^{\tau\varepsilon})| \leq \sum_{b\in\mathcal{B}} \sup_{b\in\mathcal{B}} |g_\theta^\pi(b)| \cdot |([y_1(b)]^{\tau\varepsilon} - [y_2(b)]^{\tau\varepsilon})|
$$
$$
\leq \exp(\varepsilon) \cdot \sum_{b\in\mathcal{B}} |[y_1(b)]^{\tau\varepsilon} - [y_2(b)]^{\tau\varepsilon}| \leq \exp(\varepsilon) \cdot d_{\tau\varepsilon}(y_1, y_2), \tag{213}
$$

where we denote the $\tau\varepsilon$-distance between $y_1$ and $y_2$ as

$$
d_{\tau\varepsilon}(y_1, y_2) := \sum_{b\in\mathcal{B}} |[y_1(b)]^{\tau\varepsilon} - [y_2(b)]^{\tau\varepsilon}|. \tag{214}
$$

For any $\sigma \in [0, 1]$ and the given recommendation policies $y_1, y_2$, we have a corresponding separation of the follower's action space $\mathcal{B}$ as follows:

$$
\mathcal{B}_<^\sigma(y_1, y_2) := \{b \in \mathcal{B} : |[y_1(b)]^{\tau\varepsilon} - [y_2(b)]^{\tau\varepsilon}| < \sigma\}, \tag{215}
$$
$$
\mathcal{B}_\geq^\sigma(y_1, y_2) := \{b \in \mathcal{B} : |[y_1(b)]^{\tau\varepsilon} - [y_2(b)]^{\tau\varepsilon}| \geq \sigma\}, \tag{216}
$$

and it is evident that $\mathcal{B} = \mathcal{B}_<^\sigma(y_1, y_2) \cup \mathcal{B}_\geq^\sigma(y_1, y_2)$ and $\mathcal{B}_<^\sigma(y_1, y_2) \cap \mathcal{B}_\geq^\sigma(y_1, y_2) = \varnothing$.

(i) For any $b \in \mathcal{B}_\geq^\sigma(y_1, y_2)$, we turn to show that, there exists a finite constant $C_\sigma > 0$ such that

$$
f_\theta^\pi(b; y_1, y_2) := |[y_1(b)]^{\tau\varepsilon} \cdot h_\theta^\pi(y_2) - [y_2(b)]^{\tau\varepsilon} \cdot h_\theta^\pi(y_1)| \leq C_\sigma \cdot |[y_1(b)]^{\tau\varepsilon} - [y_2(b)]^{\tau\varepsilon}|. \tag{217}
$$

Since $b \in \mathcal{B}_\geq^\sigma(y_1, y_2)$, we have $|[y_1(b)]^{\tau\varepsilon} - [y_2(b)]^{\tau\varepsilon}| \geq \sigma$, and hence

$$
\frac{|[y_1(b)]^{\tau\varepsilon} \cdot h_\theta^\pi(y_2) - [y_2(b)]^{\tau\varepsilon} \cdot h_\theta^\pi(y_1)|}{|[y_1(b)]^{\tau\varepsilon} - [y_2(b)]^{\tau\varepsilon}|} \leq \frac{|[y_1(b)]^{\tau\varepsilon} \cdot h_\theta^\pi(y_2) - [y_2(b)]^{\tau\varepsilon} \cdot h_\theta^\pi(y_1)|}{\sigma}
$$
$$
\leq \frac{|[y_1(b)]^{\tau\varepsilon} \cdot h_\theta^\pi(y_2)| + |[y_2(b)]^{\tau\varepsilon} \cdot h_\theta^\pi(y_1)|}{\sigma} \leq \frac{2\exp(\varepsilon B_A)}{\sigma} < \infty, \tag{218}
$$

where we use the fact that $h_\theta^\pi(y) \in [1, \exp(\varepsilon B_A)]$ for any $y \in \mathcal{Y}$. So we have

$$
|[y_1(b)]^{\tau\varepsilon} \cdot h_\theta^\pi(y_2) - [y_2(b)]^{\tau\varepsilon} \cdot h_\theta^\pi(y_1)| \leq \frac{2\exp(\varepsilon B_A)}{\sigma} \cdot |[y_1(b)]^{\tau\varepsilon} - [y_2(b)]^{\tau\varepsilon}|
$$
$$
= C_\sigma \cdot |[y_1(b)]^{\tau\varepsilon} - [y_2(b)]^{\tau\varepsilon}|, \tag{219}
$$

where we take $C_\sigma = \frac{2\exp(\varepsilon B_A)}{\sigma}$.

Therefore, we have

$$
\begin{aligned}
\sum_{b \in \mathcal{B}^\sigma_\geq(y_1,y_2)} f^\pi_\theta(b; y_1, y_2) &= \sum_{b \in \mathcal{B}^\sigma_\geq(y_1,y_2)} |[y_1(b)]^{\tau\varepsilon} \cdot h^\pi_\theta(y_2) - [y_2(b)]^{\tau\varepsilon} \cdot h^\pi_\theta(y_1)| \\
&\leq C_\sigma \cdot \sum_{b \in \mathcal{B}^\sigma_\geq(y_1,y_2)} |[y_1(b)]^{\tau\varepsilon} - [y_2(b)]^{\tau\varepsilon}| \\
&\leq C_\sigma \cdot \sum_{b \in \mathcal{B}} |[y_1(b)]^{\tau\varepsilon} - [y_2(b)]^{\tau\varepsilon}| = C_\sigma \cdot d_{\tau\varepsilon}(y_1, y_2).
\end{aligned}
\tag{220}
$$

(ii) For any $b \in \mathcal{B}^\sigma_<(y_1, y_2)$, we turn to establish an upper bound on

$$
f^\pi_\theta(b; y_1, y_2) := |[y_1(b)]^{\tau\varepsilon} \cdot h^\pi_\theta(y_2) - [y_2(b)]^{\tau\varepsilon} \cdot h^\pi_\theta(y_1)|.
$$

Since $b \in \mathcal{B}^\sigma_<(y_1, y_2)$, we have $|[y_1(b)]^{\tau\varepsilon} - [y_2(b)]^{\tau\varepsilon}| < \sigma$, and hence

$$
-\sigma < [y_1(b)]^{\tau\varepsilon} - [y_2(b)]^{\tau\varepsilon} < \sigma.
\tag{221}
$$

If $[y_1(b)]^{\tau\varepsilon} \cdot h^\pi_\theta(y_2) - [y_2(b)]^{\tau\varepsilon} \cdot h^\pi_\theta(y_1) \geq 0$, then we have

$$
\begin{aligned}
f^\pi_\theta(b; y_1, y_2) := |[y_1(b)]^{\tau\varepsilon} \cdot h^\pi_\theta(y_2) - [y_2(b)]^{\tau\varepsilon} \cdot h^\pi_\theta(y_1)| &= [y_1(b)]^{\tau\varepsilon} \cdot h^\pi_\theta(y_2) - [y_2(b)]^{\tau\varepsilon} \cdot h^\pi_\theta(y_1) \\
&\leq ([y_2(b)]^{\tau\varepsilon} + \sigma) \cdot h^\pi_\theta(y_2) - [y_2(b)]^{\tau\varepsilon} \cdot h^\pi_\theta(y_1) = [y_2(b)]^{\tau\varepsilon} \cdot (h^\pi_\theta(y_2) - h^\pi_\theta(y_1)) + \sigma \cdot h^\pi_\theta(y_2) \\
&\leq [y_2(b)]^{\tau\varepsilon} \cdot |h^\pi_\theta(y_2) - h^\pi_\theta(y_1)| + \sigma \cdot h^\pi_\theta(y_2) \leq \exp(\varepsilon) \cdot d_{\tau\varepsilon}(y_1, y_2) + \sigma \cdot h^\pi_\theta(y_2),
\end{aligned}
\tag{222}
$$

where the inequalities are by equation 213, equation 221 and the fact that $[y_2(b)]^{\tau\varepsilon} \leq 1$.

If $[y_1(b)]^{\tau\varepsilon} \cdot h^\pi_\theta(y_2) - [y_2(b)]^{\tau\varepsilon} \cdot h^\pi_\theta(y_1) < 0$, then we have

$$
\begin{aligned}
f^\pi_\theta(b; y_1, y_2) := |[y_1(b)]^{\tau\varepsilon} \cdot h^\pi_\theta(y_2) - [y_2(b)]^{\tau\varepsilon} \cdot h^\pi_\theta(y_1)| &= [y_2(b)]^{\tau\varepsilon} \cdot h^\pi_\theta(y_1) - [y_1(b)]^{\tau\varepsilon} \cdot h^\pi_\theta(y_2) \\
&\leq ([y_1(b)]^{\tau\varepsilon} + \sigma) \cdot h^\pi_\theta(y_1) - [y_1(b)]^{\tau\varepsilon} \cdot h^\pi_\theta(y_2) = [y_1(b)]^{\tau\varepsilon} \cdot (h^\pi_\theta(y_1) - h^\pi_\theta(y_2)) + \sigma \cdot h^\pi_\theta(y_1) \\
&\leq [y_1(b)]^{\tau\varepsilon} \cdot |h^\pi_\theta(y_1) - h^\pi_\theta(y_2)| + \sigma \cdot h^\pi_\theta(y_1) \leq \exp(\varepsilon) \cdot d_{\tau\varepsilon}(y_1, y_2) + \sigma \cdot h^\pi_\theta(y_1),
\end{aligned}
\tag{223}
$$

where the inequalities are by equation 213, equation 221 and the fact that $[y_1(b)]^{\tau\varepsilon} \leq 1$.

By equation 222 and equation 223, we have

$$
\begin{aligned}
f^\pi_\theta(b; y_1, y_2) &\leq \exp(\varepsilon) \cdot d_{\tau\varepsilon}(y_1, y_2) + \sigma \cdot \max\{h^\pi_\theta(y_1), h^\pi_\theta(y_2)\} \\
&\leq \exp(\varepsilon) \cdot d_{\tau\varepsilon}(y_1, y_2) + \sigma \cdot \exp(\varepsilon B_A).
\end{aligned}
\tag{224}
$$

Therefore, we have

$$
\begin{aligned}
\sum_{b \in \mathcal{B}^\sigma_<(y_1,y_2)} f^\pi_\theta(b; y_1, y_2) &= \sum_{b \in \mathcal{B}^\sigma_<(y_1,y_2)} |[y_1(b)]^{\tau\varepsilon} \cdot h^\pi_\theta(y_2) - [y_2(b)]^{\tau\varepsilon} \cdot h^\pi_\theta(y_1)| \\
&\leq \sum_{b \in \mathcal{B}^\sigma_<(y_1,y_2)} \{\exp(\varepsilon) \cdot d_{\tau\varepsilon}(y_1, y_2) + \sigma \cdot \exp(\varepsilon B_A)\} \leq \sum_{b \in \mathcal{B}} \{\exp(\varepsilon) \cdot d_{\tau\varepsilon}(y_1, y_2) + \sigma \cdot \exp(\varepsilon B_A)\} \\
&= |\mathcal{B}| \cdot \{\exp(\varepsilon) \cdot d_{\tau\varepsilon}(y_1, y_2) + \sigma \cdot \exp(\varepsilon B_A)\}.
\end{aligned}
\tag{225}
$$

Combining equation 220 and equation 225, we have

$$
\begin{aligned}
\sum_{b \in \mathcal{B}} f^\pi_\theta(b; y_1, y_2) &= \sum_{b \in \mathcal{B}^\sigma_\geq(y_1,y_2)} f^\pi_\theta(b; y_1, y_2) + \sum_{b \in \mathcal{B}^\sigma_<(y_1,y_2)} f^\pi_\theta(b; y_1, y_2) \\
&\leq C_\sigma \cdot d_{\tau\varepsilon}(y_1, y_2) + |\mathcal{B}| \cdot \{\exp(\varepsilon) \cdot d_{\tau\varepsilon}(y_1, y_2) + \sigma \cdot \exp(\varepsilon B_A)\} \\
&= \frac{2\exp(\varepsilon B_A) \cdot d_{\tau\varepsilon}(y_1, y_2)}{\sigma} + |\mathcal{B}|\exp(\varepsilon B_A) \cdot \sigma + |\mathcal{B}|\exp(\varepsilon) \cdot d_{\tau\varepsilon}(y_1, y_2),
\end{aligned}
\tag{226}
$$

which holds for any $\sigma \in [0, 1]$ and the given recommendation policies $y_1, y_2$. By calculus, we have that the RHS of equation 226 reaches its minimum when

$$\sigma = \sqrt{\frac{2 \exp(\varepsilon B_A) \cdot d_{\tau\varepsilon}(y_1, y_2)}{|\mathcal{B}| \exp(\varepsilon B_A)}} = \sqrt{\frac{2 \cdot d_{\tau\varepsilon}(y_1, y_2)}{|\mathcal{B}|}} =: \sigma_{\min}. \tag{227}$$

Plugging equation 227 into equation 226, we have

$$\sum_{b \in \mathcal{B}} f_\theta^\pi(b; y_1, y_2) \leq \frac{2 \exp(\varepsilon B_A) \cdot d_{\tau\varepsilon}(y_1, y_2)}{\sigma_{\min}} + |\mathcal{B}| \exp(\varepsilon B_A) \cdot \sigma_{\min} + |\mathcal{B}| \exp(\varepsilon) \cdot d_{\tau\varepsilon}(y_1, y_2)$$

$$= \frac{2 \exp(\varepsilon B_A) \cdot d_{\tau\varepsilon}(y_1, y_2)}{\sqrt{\frac{2 \cdot d_{\tau\varepsilon}(y_1, y_2)}{|\mathcal{B}|}}} + |\mathcal{B}| \exp(\varepsilon B_A) \sqrt{\frac{2 \cdot d_{\tau\varepsilon}(y_1, y_2)}{|\mathcal{B}|}} + |\mathcal{B}| \exp(\varepsilon) \cdot d_{\tau\varepsilon}(y_1, y_2)$$

$$= \sqrt{8|\mathcal{B}|} \exp(\varepsilon B_A) \sqrt{d_{\tau\varepsilon}(y_1, y_2)} + |\mathcal{B}| \exp(\varepsilon) \cdot d_{\tau\varepsilon}(y_1, y_2). \tag{228}$$

Finally, we turn to show that there exists a constant $C$ such that $d_{\tau\varepsilon}(y_1, y_2)$ defined in equation 214 satisfies

$$d_{\tau\varepsilon}(y_1, y_2) := \sum_{b \in \mathcal{B}} |[y_1(b)]^{\tau\varepsilon} - [y_2(b)]^{\tau\varepsilon}| \leq C \cdot \|y_1 - y_2\|_1.$$

By Lagrange's mean value theorem, we have that

$$\frac{|[y_1(b)]^{\tau\varepsilon} - [y_2(b)]^{\tau\varepsilon}|}{|[y_1(b)] - [y_2(b)]|} \leq \frac{\tau\varepsilon}{(\min\{y_1(b), y_2(b)\})^{1-\tau\varepsilon}}$$

$$\leq \frac{\tau\varepsilon}{(\min_{y \in \{y_1, y_2\}}\{y(b)\})^{1-\tau\varepsilon}} \leq \frac{\tau\varepsilon}{(\min_{y \in \mathcal{Y}_{QR}}\{y(b)\})^{1-\tau\varepsilon}} \leq \frac{\tau\varepsilon}{\mu_L^{1-\tau\varepsilon}}, \tag{229}$$

where $\mu_L := \frac{1}{|\mathcal{B}| \cdot \exp(\eta)}$. The inequalities holds by $y_1, y_2 \in \mathcal{Y}_{QR}$ and equation 24 in Theorem C.3.

Therefore, we have

$$d_{\tau\varepsilon}(y_1, y_2) := \sum_{b \in \mathcal{B}} |[y_1(b)]^{\tau\varepsilon} - [y_2(b)]^{\tau\varepsilon}| \leq \sum_{b \in \mathcal{B}} \frac{\tau\varepsilon}{\mu_L^{1-\tau\varepsilon}} |[y_1(b)] - [y_2(b)]|$$

$$= \frac{\tau\varepsilon}{\mu_L^{1-\tau\varepsilon}} \sum_{b \in \mathcal{B}} |[y_1(b)] - [y_2(b)]| = \frac{\tau\varepsilon}{\mu_L^{1-\tau\varepsilon}} \cdot \|y_1 - y_2\|_1, \tag{230}$$

where the inequality holds by equation 229.

Finally, combining equation 228 and equation 230, we have

$$\sum_{b \in \mathcal{B}} f_\theta^\pi(b; y_1, y_2) \leq \sqrt{8|\mathcal{B}|} \exp(\varepsilon B_A) \sqrt{d_{\tau\varepsilon}(y_1, y_2)} + |\mathcal{B}| \exp(\varepsilon) \cdot d_{\tau\varepsilon}(y_1, y_2)$$

$$\leq \sqrt{8|\mathcal{B}|} \exp(\varepsilon B_A) \sqrt{\frac{\tau\varepsilon}{\mu_L^{1-\tau\varepsilon}} \cdot \|y_1 - y_2\|_1} + |\mathcal{B}| \exp(\varepsilon) \cdot \frac{\tau\varepsilon}{\mu_L^{1-\tau\varepsilon}} \cdot \|y_1 - y_2\|_1$$

$$\leq \exp(\varepsilon B_A) \sqrt{8\tau\varepsilon \exp(\varepsilon)|\mathcal{B}|^{2-\tau\varepsilon}} \sqrt{\|y_1 - y_2\|_1} + \tau\varepsilon \exp(2\varepsilon)|\mathcal{B}|^{2-\tau\varepsilon}\|y_1 - y_2\|_1. \tag{231}$$

Combining equation 211 and equation 231, we have

$$\|\nu_\theta^{\pi, y_1} - \nu_\theta^{\pi, y_2}\|_1 = \sum_{b \in \mathcal{B}} |\nu_\theta^{\pi, y_1}(b) - \nu_\theta^{\pi, y_2}(b)| \leq \exp(\varepsilon) \cdot \sum_{b \in \mathcal{B}} f_\theta^\pi(b; y_1, y_2)$$

$$\leq \exp(\varepsilon(1 + B_A)) \sqrt{8\tau\varepsilon \exp(\varepsilon)|\mathcal{B}|^{2-\tau\varepsilon}} \sqrt{\|y_1 - y_2\|_1} + \tau\varepsilon \exp(3\varepsilon)|\mathcal{B}|^{2-\tau\varepsilon}\|y_1 - y_2\|_1. \tag{232}$$

Finally, we have

$$\mathrm{TV}(\nu_\theta^{\pi,y_1}, \nu_\theta^{\pi,y_2}) = \frac{1}{2}\|\nu_\theta^{\pi,y_1} - \nu_\theta^{\pi,y_2}\|_1$$

$$\leq \frac{\sqrt{2}}{2}\exp(\varepsilon(1+B_A))\sqrt{8\tau\varepsilon\exp(\varepsilon)|\mathcal{B}|^{2-\tau\varepsilon}}\sqrt{\frac{1}{2}\|y_1-y_2\|_1} + \frac{1}{2}\tau\varepsilon\exp(3\varepsilon)|\mathcal{B}|^{2-\tau\varepsilon}\|y_1-y_2\|_1$$

$$\leq \frac{\sqrt{2}}{2}\exp(\varepsilon(1+B_A))\sqrt{8\tau\varepsilon\exp(\varepsilon)|\mathcal{B}|^{2-\tau\varepsilon}}\sqrt{\mathrm{TV}(y_1,y_2)} + \tau\varepsilon\exp(3\varepsilon)|\mathcal{B}|^{2-\tau\varepsilon}\mathrm{TV}(y_1,y_2)$$

$$\leq C \cdot \left(\sqrt{\mathrm{TV}(y_1,y_2)} + \mathrm{TV}(y_1,y_2)\right), \tag{233}$$

where the constant

$$C := \max\left\{\frac{\sqrt{2}}{2}\exp(\varepsilon(1+B_A))\sqrt{8\tau\varepsilon\exp(\varepsilon)|\mathcal{B}|^{2-\tau\varepsilon}} , \ \tau\varepsilon\exp(3\varepsilon)|\mathcal{B}|^{2-\tau\varepsilon}\right\},$$

which completes the proof. $\qquad\square$

## L  OTHER AUXILIARY PROPOSITIONS

**Lemma L.1** (Lemma A.4 in Foster et al. (2021)). *For any sequence of real-valued random variables $(X_t)_{t\in[T]}$ adapted to a filtration $(\mathcal{F}_t)_{t\in[T]}$, it holds with probability at least $1-\delta$ for all $t\in[T]$ that*

$$\sum_{i=1}^{t} X_i \leq \sum_{i=1}^{t}\log\left(\mathbb{E}\left[e^{X_i}\mid\mathcal{F}_{i-1}\right]\right) + \log(\delta^{-1}).$$

**Lemma L.2** (Elliptical potential lemma for matrices, Lemma G.10 in Chen et al. (2023)). *Suppose $U_0 = \lambda I_d$, $U_t = U_{t-1} + X_t$ for $X_t \in \mathbb{S}_+^d$, and $\mathrm{Tr}(X_t) \leq L$, then*

$$\sum_{t=1}^{T}\sqrt{\mathrm{Tr}\left(U_{t-1}^{-1}X_t\right)} \leq \sqrt{\frac{LT/\lambda}{\log(1+L/\lambda)}\cdot d\log\left(1+\frac{LT}{\lambda d}\right)},$$

*and also*

$$\sum_{t=1}^{T}\mathrm{Tr}\left(U_{t-1}^{-1}X_t\right) \leq \frac{L/\lambda}{\log(1+L/\lambda)}\cdot d\log\left(1+\frac{LT}{\lambda d}\right).$$

**Lemma L.3.** *Under the conditions of Theorem L.2, we further have*

$$\sum_{t=1}^{T}\left[\mathrm{Tr}\left(U_{t-1}^{-1}X_t\right)\right]^2 \leq \frac{L^2/\lambda^2}{\log^2(1+L/\lambda)}\cdot d^2\log^2\left(1+\frac{LT}{\lambda d}\right).$$

*Proof.* Since $X_t \in \mathbb{S}_+^d$, we have that $X_t$ and $U_{t-1}$ are both symmetric nonnegative definite matrices. Hence, $U_{t-1}^{-1}X_t$ is also a symmetric nonnegative definite matrix. So we have $\mathrm{Tr}\left(U_{t-1}^{-1}X_t\right) \geq 0$ for each $t \in [T]$. Therefore, by the result of Theorem L.2, we have

$$\sum_{t=1}^{T}\left[\mathrm{Tr}\left(U_{t-1}^{-1}X_t\right)\right]^2 \leq \left[\sum_{t=1}^{T}\mathrm{Tr}\left(U_{t-1}^{-1}X_t\right)\right]^2 \leq \frac{L^2/\lambda^2}{\log^2(1+L/\lambda)}\cdot d^2\log^2\left(1+\frac{LT}{\lambda d}\right), \tag{234}$$

which completes the proof. $\qquad\square$

**Proposition L.4.** *For any $\theta,\widetilde{\theta}\in\Theta$, $\pi\in\Pi$, $y\in\mathcal{Y}$, we have*

$$\Sigma^{\pi,y,\theta} \leq \exp\left(2\varepsilon B_A\right)\cdot\Sigma^{\pi,y,\widetilde{\theta}},$$

*and also*

$$\Gamma_t^2(\pi,y,\theta) \leq \exp\left(4\varepsilon B_A\right)\cdot\Gamma_t^2(\pi,y,\widetilde{\theta}).$$

*Proof.* This is similar to Proposition D.1 in Chen et al. (2023), but with difference parameters.

By Theorem K.4, we have $\|A_\theta^{\pi,y}(\cdot)\| \le B_A$ and $\|A_{\widetilde{\theta}}^{\pi,y}(\cdot)\| \le B_A$.

To see how we derive the upper bound, we note that by definition $\Gamma_t^2(\pi,y,\theta) = 2(\varepsilon\xi + C_\varepsilon^{(3)}\xi^2)$ with

$$\xi = \sqrt{\mathrm{Tr}\left(\left(\Sigma_t^\theta + I_d\right)^\dagger \Sigma^{\pi,y,\theta}\right)} \cdot \sqrt{8C_\varepsilon^2\beta + 4B_\Theta^2}.$$

Furthermore, by definition of $\Sigma^{\pi,y,\theta}$ and $\Sigma_t^\theta$, we have

$$\Sigma^{\pi_i,y_i,\theta} = \mathrm{Cov}^{\pi_i,y_i,\theta}\left[\phi^{\pi_i}(b)\right] = \sum_{b',b''} \phi^{\pi_i}(b') \cdot \mathrm{H}^{\pi_i,y_i,\theta}(b',b'') \cdot \phi^{\pi_i}(b'')^\top$$

$$\le \exp\left(2\varepsilon B_A\right) \sum_{b',b''} \phi^{\pi_i}(b') \cdot \mathrm{H}^{\pi_i,y_i,\widetilde{\theta}}(b',b'') \cdot \phi^{\pi_i}(b'')^\top = \exp\left(2\varepsilon B_A\right) \Sigma^{\pi_i,y_i,\widetilde{\theta}},$$

and similarly,

$$\Sigma_t^\theta = \sum_{i=1}^{t-1} \mathrm{Cov}^{\pi_i,y_i,\theta}\left[\phi^{\pi_i}(b)\right] = \sum_{i=1}^{t-1}\sum_{b',b''} \phi^{\pi_i}(b') \cdot \mathrm{H}^{\pi_i,y_i,\theta}(b',b'') \cdot \phi^{\pi_i}(b'')^\top$$

$$\le \exp\left(2\varepsilon B_A\right) \sum_{i=1}^{t-1}\sum_{b',b''} \phi^{\pi_i}(b') \cdot \mathrm{H}^{\pi_i,y_i,\widetilde{\theta}}(b',b'') \cdot \phi^{\pi_i}(b'')^\top = \exp\left(2\varepsilon B_A\right) \Sigma_t^{\widetilde{\theta}},$$

where we define $\mathrm{H}^{\pi,y,\theta}(b',b'') = \mathrm{diag}(\nu^{\pi,y,\theta}) - \nu^{\pi,y,\theta}\nu^{\pi,y,\theta\top}$ as the Hessian matrix corresponding to $\nu^{\pi,y,\theta}$. Here, the inequalities hold by invoking Theorem J.2 and a lower bound follows similarly by invoking the lower bound in Theorem J.2.

Therefore, we have for $\Gamma_t^2$ that

$$\Gamma_t^2(\pi,y,\theta) = 2\varepsilon\sqrt{\mathrm{Tr}\left(\left(\Sigma_t^\theta + I_d\right)^\dagger \Sigma^{\pi,y,\theta}\right)}\sqrt{8C_\varepsilon^2\beta + 4B_\Theta^2}$$

$$+ 2C_\varepsilon^{(3)}\left(\sqrt{\mathrm{Tr}\left(\left(\Sigma_t^\theta + I_d\right)^\dagger \Sigma^{\pi,y,\theta}\right)}\sqrt{8C_\varepsilon^2\beta + 4B_\Theta^2}\right)^2$$

$$\le 2\varepsilon\exp\left(2\varepsilon B_A\right)\sqrt{\mathrm{Tr}\left(\left(\Sigma_t^{\widetilde{\theta}} + I_d\right)^\dagger \Sigma^{\pi,y,\widetilde{\theta}}\right)} \cdot \sqrt{8C_\varepsilon^2\beta + 4B_\Theta^2}$$

$$+ 2C_\varepsilon^{(3)}\exp\left(4\varepsilon B_A\right)\left(\sqrt{\mathrm{Tr}\left(\left(\Sigma_t^{\widetilde{\theta}} + I_d\right)^\dagger \Sigma^{\pi,y,\widetilde{\theta}}\right)} \cdot \sqrt{8C_\varepsilon^2\beta + 4B_\Theta^2}\right)^2$$

$$\le \exp\left(4\varepsilon B_A\right) \cdot \Gamma_t^2(\pi,y,\widetilde{\theta}),$$

which completes the proof. $\square$

