# OpenReview forum: "Learning to Incentivize on the Fly: Leader-Follower Games with Policy Recommendation"
_ICLR.cc/2026/Conference — Submitted to ICLR 2026_

### Official Review · Reviewer_CLjs · 2025-10-22

**Soundness:** 3
**Presentation:** 3
**Contribution:** 2
**Rating:** 4
**Confidence:** 4

**Summary:**

This paper studies online learning for leader–follower matrix games when the leader commits to a policy and also recommends a policy to the follower, under an "IC" (credibility) constraint. The follower is modeled as boundedly rational and computes a one-step mirror-descent update that trades off reward, entropy, and a KL penalty from the leader’s recommendation. The authors formalize ICV via a TV between the recommendation and the follower’s response, and show that QRE is the unique fixed point of the recommendation–response map. This paper proposes PROM, and under linear rewards and finite follower actions they prove \(\tilde O(d^2\sqrt{T})\) bounds on standard regret, IC-violation regret and absolute regret.

**Strengths:**

1. The paper introduces a new formulation of the objective, where the leader is constrained to a credibility constraint and the follower updates according to the leader's recommendation policy . The ICV metric and its fixed-point connection to the quantal response are interesting and provide a well-defined policy set.
2. The proposed PROM algorithm achieves $\tilde{\mathcal{O}}(d^2 \sqrt{T})$ standard regret and ICV regret, providing a sample-efficient learning method.

**Weaknesses:**

1. The paper assumes the leader knows\(\eta,\tau\);This can be unrealistic and learning them is nontrivial. It’s unclear how sensitive and robust the results are under noisy or approximate $\alpha, \beta$.
2. The paper’s “IC” is not the incentive compatibility in the mechanism-design sense. What the paper calls “IC” is closer to a credibility requirement than the truth-telling incentive compatibility in classical mechanism design. And generally the leader has no incentive to add this “IC” constraint and the follower has no clear incentive to add a KL term according to the leader’s policy; these settings and assumptions lack motivation.
3. The technical contribution is incremental to prior work on Stackelberg/QSE learning and entropy-regularized responses.

**Questions:**

1. How sensitive are the results to the choices of $\eta$ and $\tau$? In particular, how do estimation errors in these parameters influence the results?
2. Suggestion: It is more accurate to use “online learning” rather than “online RL” throughout.  “online RL” typically implies results in Markov decision processes/Markov games, which your current setting (matrix Stackelberg games) does not cover and readers may anticipate analyses beyond the scope of this work.
3. In your analysis and proofs, which aspects of the analysis differ from the $H=1$ case in Chen et al.? In particular, which techniques are newly introduced due to the need to control \textbf{ICV} and distribution shift compared to previous work?

---

> ### Author Response · Authors · 2025-11-22
> **Response to Reviewer CLjs**
>
> Dear Reviewer CLjs,
>
> We sincerely thank you for your constructive feedback. We appreciate your recognition of the new objective formulation, the interesting fixed-point connection between the ICV metric and the quantal response, and the sample efficiency of our proposed algorithm. We will address your concerns and questions below, particularly highlighting the technical novelty compared to prior work.
>
> **1. Sensitivity to Parameters $\eta$ and $\tau$ (Response to Weakness 1 and Question 1)**
>
> Note: We assume your mention of “$\alpha, \beta$” is a typo referring to the rationality parameter $\eta$ and recommendation weight $\tau$.
>
> You are correct that we currently assume $\eta$ and $\tau$ are known to the leader. While learning these parameters is an interesting direction for future work, assuming the follower's behavioral parameters known is a standard starting point in the Stackelberg learning and optimization literature. For instance, in the literature on computing optimal strategies against quantal response followers, such as Yang et al. (2012) and Chen et al. (2023), the rationality parameter $\eta$ is treated as a fixed known input. Moreover, our setting covers a more general spectrum than works like Zhao et al. (2023) and Zhong et al. (2023), which assume the follower is perfectly rational, a setting mathematically equivalent to assuming a known $\eta \to \infty$ in our model, making our setting a more general framework. Therefore, our model actually covers a more general framework with respect to rationality than works limited to the perfect rationality case.
>
> ***Robustness and Misspecification***:
>
> Crucially, our framework is mathematically robust to estimation errors. Due to the Lipschitz continuity of the response model (Eq. 5), a small fixed parameter error $\delta$ satisfying $|\hat{\eta} - \eta| \leq \delta$ and $|\hat{\tau} - \tau| \leq \delta$ results in a bounded deviation of order $\mathcal{O}(\delta)$ in the estimated response distribution.
> In terms of regret, this parameter mismatch (model misspecification) introduces an additive term scaling as $\mathcal{O}(\delta \cdot T)$. While this term is linear in $T$, this type of guarantee is standard and widely accepted in the theoretical RL literature when dealing with misspecification. A seminal example is Jin et al. (2020), which proves that for $\zeta$-approximate linear MDPs, the regret bound includes an additional $\mathcal{O}(\zeta \cdot T)$ term (Theorem 3.2), compared with linear MDPs. As they note, this term reflects the intrinsic bias of the approximation, yet the algorithm still enjoys good theoretical guarantees because performance degrades gracefully.
>
> Similarly, in our setting, this $\mathcal{O}(\delta \cdot T)$ term confirms that our algorithm does not fail catastrophically but converges to an $\mathcal{O}(\delta)$-optimal policy. This demonstrates that performance degrades gracefully rather than catastrophically. The sublinear rate $\tilde{\mathcal{O}}(\sqrt{T})$ is recovered for the learning component, ensuring sample efficiency up to the limit imposed by parameter precision. We will add a discussion on this sensitivity analysis in the revised version.
>
>
> **2. Motivation for IC and the KL Term (Response to Weakness 2)**
>
> As for the IC definition, you are right that our usage of IC differs from the classical definition of truth-telling in mechanism design. However, it aligns precisely with the obedience constraint found in the Bayesian Persuasion and Information Design literature (e.g., Kamenica & Gentzkow, 2011; Bergemann & Morris, 2019). In our setting, IC represents a credibility or obedience requirement: it ensures that the recommended policy $y$ is the follower's best response to itself. This distinction is standard in persuasion games where, by the revelation principle, one can restrict attention to recommendations that the receiver finds optimal to follow. Crucially, Proposition 4.1 confirms that satisfying this constraint allows the leader to induce the quantal response equilibrium behavior, thereby bridging the gap between our recommendation framework and standard Stackelberg games.

---

> ### Author Response · Authors · 2025-11-22
> **Response to Reviewer CLjs (continued)**
>
> ***Motivations for the Agents***:
>
> (1) **Follower's Incentive (Why the KL term?)**: The inclusion of the KL divergence term (Eq. 3) models the anchoring effect arising from computational constraints. While the entropy term captures the intrinsic stochasticity of bounded rationality (standard quantal response), the KL term $D(\nu||y)$ specifically addresses the resource asymmetry discussed in our motivating example (Appendix B.1). Lacking the computational capacity for global optimization, the follower adopts a heuristic strategy: treating the leader's recommendation $y$ as a trusted “anchor” and performing a single-step optimization (mirror descent) from that reference point. The KL term is the mathematical consequence of this local update rule. Without it, the recommendation $y$ would effectively disappear from the follower's objective, rendering the recommendation mechanism void.
>
> (2) **Leader's Incentive (Why enforce IC?)**: While standard Stackelberg leaders simply commit to a policy $\pi$, a leader in our online learning setting has strong incentives to enforce IC on the recommendation $y$ to ensure learnability and control. Since the leader does not know the follower's reward function $r$, recommending an arbitrary, non-IC $y$ would cause the follower to play a complex “off-equilibrium” response $\nu_{\theta}^{\pi, y}$. Inferring the underlying parameter $\theta$ from such transient, unstable responses is statistically difficult. By enforcing IC, the leader ensures the interaction stabilizes on the quantal response manifold (where $\nu \approx y$), making the follower's behavior consistent and the underlying parameters learnable. Furthermore, this constraint converts an indirect influence problem into a direct control problem (obedience), reducing strategic uncertainty and rendering the high-dimensional, non-convex optimization landscape computationally tractable.
>
> **3. Technical Novelty vs. Chen et al. (Response to Weakness 3 & Question 3)**
>
> Our work is not merely an incremental extension of Chen et al. (2023), even when $H=1$. The introduction of policy recommendation introduces specific, non-trivial technical challenges that Chen et al. (2023) did not face. Below, we highlight two critical challenges that are unique to our setting. For a comprehensive discussion of the full range of technical hurdles and contributions, we respectfully refer you to Appendix C.
>
> ***(1) Distribution Shifts***:
>
> The most significant technical divergence arises from the distribution shift caused by the recommendation mechanism. In Chen et al. (2023), the follower always plays the quantal response, meaning the observed data distribution matches the learning target. In our setting, however, the follower plays her response $\nu_{\theta}^{\pi, y}$ (Eq. 3), which is biased by the recommendation $y$. Yet, the leader's ultimate goal is to learn the quantal response $\mu_{\theta}^{\pi}$ (Eq. 2), because the optimal IC recommendation must be a QR. This leads to a mismatch that we observe data from distribution $\nu$ but must bound the error of $\mu$.
>
> Standard MLE analysis used in Chen et al. (2023) only bounds the error for the observed distribution ($\nu$). We cannot trivially bound the error of the target quantal response $\mu$ using the error of the observed follower’s response $\nu$ because the mapping is not identifiable. To solve this, we developed a novel analysis leveraging the invariance property of the follower’s response under reward shifts (see Appendix C.4). This allows us to fundamentally bridge the gap between the observed follower’s response error and the target quantal response error (Eq. 126), a technique that is both absent in and unnecessary for the framework of Chen et al. (2023).
>
> ***(2) Absolute Regret Analysis***:
>
> Furthermore, while Chen et al. (2023) only analyze standard regret, this metric is fundamentally insufficient in our constrained setting. In constrained optimization, a policy that violates the constraint can theoretically yield “super-optimal” rewards higher than the equilibrium value, resulting in negative suboptimality. Consequently, a vanishing standard regret could misleadingly mask large oscillations between valid, low-reward policies and invalid, high-reward policies.

---

> ### Author Response · Authors · 2025-11-22
> **Response to Reviewer CLjs (continued 2)**
>
> To strictly guarantee convergence to the equilibrium, we introduce and analyze absolute regret in Theorem 6.2. Establishing a sublinear bound for absolute regret is technically non-trivial, as it requires proving that controlling the cumulative constraint violation (ICV regret) effectively bounds the magnitude of these negative regret terms. To achieve this, we developed a delicate and rigorous perturbation analysis of the optimization landscape in Proposition B.6. Specifically, we proved that the maximum potential utility gain from relaxing the IC constraint decays linearly with the violation magnitude. This geometric insight allows us to quantitatively link ICV regret to absolute regret, a theoretical innovation completely absent in Chen et al. (2023) and other prior work as their unconstrained setting does not require such analysis.
>
>
> **4. Terminology of Online Learning (Response to Question 2)**
>
> We fully agree with your suggestion. The use of “online RL” was actually a legacy of our broader research agenda, where we view this work as the foundational step towards incentive-compatible RL in general Markov games. We initially focused on the matrix game setting to isolate and rigorously resolve the novel theoretical complexities introduced by policy recommendation and the IC constraint, ensuring the framework is solid before introducing temporal dynamics. We appreciate you pointing out this potential ambiguity. To ensure precision and avoid creating false expectations about multi-step dynamics, we will update the terminology to “online learning” in the revised version.
>
>
>
> **Final Remarks**
>
> We thank you again for your time and insightful feedback. We hope these clarifications can fully address your concerns and highlight the novelty of our technical and theoretical contributions. If you think our responses have resolved your concerns and questions, we would be truly grateful if you would consider raising your assessment score and confidence. Thank you very much!
>
>
> **Reference**
>
> Chen et al. (2023). Actions speak what you want: Provably sampleefficient reinforcement learning of the quantal stackelberg equilibrium from strategic feedbacks.
>
> Yang et al. (2012). Computing Optimal Strategy against Quantal Response in Security Games.
>
> Zhong et al. (2023). Can reinforcement learning find stackelberg-nash equilibria in general-sum markov games with myopically rational followers?
>
> Zhao et al. (2023). Online learning in stackelberg games with an omniscient follower.
>
> Kamenica and Gentzkow (2011). Bayesian Persuasion.
>
> Bergemann & Morris (2019). Information Design: A Unified Perspective.
>
> Jin et al. (2020). Provably efficient reinforcement learning with linear function approximation.

---

> ### Author Response · Authors · 2025-12-03
> **Detailed Response on Parameter Robustness (New Section 6.2)**
>
> Dear Reviewer CLjs,
>
> We are writing to provide a detailed update specifically addressing your concerns regarding the assumption of known behavioral parameters $\eta$ and $\tau$. We fully agree that analyzing the impact of parameter estimation errors is crucial for the practical applicability and theoretical completeness of our framework.
>
> In the revised manuscript, we have added a new **Section 6.2: Robustness to Parameter Misspecification** to rigorously address this issue. We are happy to report that our analysis confirms our algorithm exhibits graceful degradation under parameter misspecification.
>
> Below we summarize the new theoretical results and their implications:
>
> **1. New Theoretical Guarantee (Theorem 6.4)**
>
> We consider a setting where the leader uses estimated parameters $\hat{\eta}, \hat{\tau}$ such that $|\hat{\eta} - \eta| \le \zeta$ and $|\hat{\tau} - \tau| \le \zeta$. We prove that with probability at least $1-p$, both the cumulative regret and ICV regret satisfy:
> $$\text{Reg}(T) \le \tilde{\mathcal{O}}(d^2\sqrt{T}) + \tilde{\mathcal{O}}(\zeta T) \quad \text{and} \quad \text{Reg}^{\text{ICV}}(T) \le \tilde{\mathcal{O}}(d^2\sqrt{T}) + \tilde{\mathcal{O}}(\zeta T).$$
> Detailed proofs, which leverage the Lipschitz continuity of the follower's response model, are provided in the new **Appendix F.5**.
>
> **2. Interpretation: Graceful Degradation & Intrinsic Bias**
> * **Controlled Bias:** The additional term $\tilde{\mathcal{O}}(\zeta T)$ is linear in $T$, implying that the average regret converges to $\mathcal{O}(\zeta)$. This theoretically confirms that the algorithm is **robust**: small estimation errors do not cause catastrophic failure but instead lead to a policy that is $\mathcal{O}(\zeta)$-optimal.
> * **Alignment with Standard Theory:** This result parallels the robustness guarantees for misspecified linear MDPs (e.g., **Jin et al., 2020**), where the regret bound naturally includes an inevitable $\mathcal{O}(\zeta T)$ term representing the intrinsic modeling bias. Our result shows that our algorithm achieves the best possible performance allowed by the parameter precision.
>
> **3. Advantage: Robustness in High Dimensions**
>
> A key insight from our new analysis is that, unlike the misspecification term in linear MDPs which typically scales with the feature dimension $d$, our linear bias term $\tilde{\mathcal{O}}(\zeta T)$ is **independent of $d$**.
> * This is because $\zeta$ here characterizes the error of scalar behavioral parameters, and the sensitivity of the follower's response is bounded by the reward magnitude rather than the feature space dimension.
> * **Implication:** This property renders our algorithm particularly robust in **large-scale and high-dimensional problems**, as the system bias remains controlled even as the feature dimension increases.
>
> We hope this rigorous analysis fully resolves your concerns regarding the realism and sensitivity of the parameter assumptions.
>
> Sincerely,
>
> The Authors

---

### Official Review · Reviewer_coiB · 2025-10-23

**Soundness:** 3
**Presentation:** 3
**Contribution:** 2
**Rating:** 2
**Confidence:** 3

**Summary:**

The paper proposes an algorithm to achieve sublinear regret in a variant of the Stackelberg game, where the leader can recommend to the follower, and the follower will take one step of mirror descent from the recommendation.

**Strengths:**

- The writing of the paper is clear and easy to follow.
- The paper discussed its relevance to prior work clearly.

**Weaknesses:**

My primary concern with this paper is the lack of motivation. The paper introduces a new setting where players’ objective functions include an additional entropy regularizer. However, this assumption seems questionable, as such regularizers are typically used as a tool to accelerate convergence (see last-iterate convergence) / encouraging exploration (see learning in MDP) rather than being inherent components of the game itself.

Moreover, the paper assumes that the follower performs mirror descent updates from the recommendation due to limited computational power. While this may seem reasonable, the choice of mirror descent is not well justified. More critically, the proposed algorithm requires the leader to know the follower’s learning rate, which is an unrealistic assumption.

In my view, a more natural assumption would be that the follower computes a best response within a neighborhood of the recommendation, constrained by a radius $r$.

**Questions:**

See weaknesses.

---

> ### Author Response · Authors · 2025-11-22
> **Response to Reviewer coiB**
>
> Dear Reviewer coiB,
>
> We sincerely thank you for your review. We appreciate your positive comments on the clarity of our writing and the relevance to prior work.
> We understand your concerns regarding the motivation of the model settings. However, we believe these concerns stem from viewing the problem through a pure optimization lens rather than a behavioral game-theoretic one. We provide detailed clarifications below to justify our modelling choices.
>
> **1. The Motivation of Entropy Regularization**
>
> You questioned whether entropy regularization is an inherent component of the game, noting it is typically used for acceleration or exploration. However, we respectfully clarify that entropy regularization is not merely an algorithmic tool for convergence in this context, but rather the foundational mathematical representation of bounded rationality in game theory. Specifically, the objective in Eq. (2) corresponds exactly to the quantal response model established in the seminal work of McKelvey & Palfrey (1995). This model posits that agents are not perfectly rational but choose better strategies with higher probabilities, where the entropy term captures the intrinsic stochasticity or calculation error of human decision-making. Consequently, this formulation is an inherent component of the behavioral model we study, which is widely adopted in Stackelberg learning literature like Chen et al. (2023), rather than an auxiliary optimization trick.
>
> **2. Justification for Mirror Descent Updates**
>
> Regarding the choice of one-step mirror descent (OMD), we employ the OMD update (Eq. 3) specifically to model the mechanism of policy recommendation under conditions of resource asymmetry. As discussed in our motivating examples in Appendix B.1, resource-constrained followers often rely on the leader's recommendation $y$ as a trusted “anchor” or reference point. OMD is the mathematically rigorous way to model such a local improvement step from a reference point, and crucially, it naturally induces the KL divergence term $D(\nu || y)$. Without this mechanism, the recommendation $y$ would have no mathematical influence on the follower's decision. Furthermore, as shown in Proposition 4.1, this dynamic is consistent in the sense that, if the leader recommends the true equilibrium strategy, the follower's OMD response coincides with the optimal quantal response.
>
> **3. Assumption of Known Parameters $\eta, \tau$**
>
> You stated that requiring the leader to know the follower’s learning rate is unrealistic. However, we respectfully point out that treating behavioral parameters (like $\eta$ and $\tau$) as known parameters is a standard starting point in the theoretical RL literature.
> For instance, in the literature on computing optimal strategies against quantal response followers, such as Yang et al. (2012) and Chen et al. (2023), the rationality parameter $\eta$ is treated as a fixed known input. Moreover, our setting covers a more general spectrum than works like Zhao et al. (2023) and Zhong et al. (2023), which assume the follower is perfectly rational, a setting mathematically equivalent to assuming a known $\eta \to \infty$ in our model, making our setting a more general framework. Therefore, our model actually covers a more general framework with respect to rationality than works limited to the perfect rationality case.
>
> ***Robustness and Misspecification***:
>
> Crucially, our framework is mathematically robust to estimation errors. Due to the Lipschitz continuity of the response model (Eq. 5), a small fixed parameter error $\delta$ satisfying $|\hat{\eta} - \eta| \leq \delta$ and $|\hat{\tau} - \tau| \leq \delta$ results in a bounded deviation of order $\mathcal{O}(\delta)$ in the estimated response distribution.
> In terms of regret, this parameter mismatch (model misspecification) introduces an additive term scaling as $\mathcal{O}(\delta \cdot T)$. While this term is linear in $T$, this type of guarantee is standard and widely accepted in the theoretical RL literature when dealing with misspecification. A seminal example is Jin et al. (2020), which proves that for $\zeta$-approximate linear MDPs, the regret bound includes an additional $\mathcal{O}(\zeta \cdot T)$ term (Theorem 3.2), compared with linear MDPs. As they note, this term reflects the intrinsic bias of the approximation, yet the algorithm still enjoys good theoretical guarantees because performance degrades gracefully.
>
> Similarly, in our setting, this $\mathcal{O}(\delta \cdot T)$ term confirms that our algorithm does not fail catastrophically but converges to an $\mathcal{O}(\delta)$-optimal policy. This demonstrates that performance degrades gracefully rather than catastrophically. The sublinear rate $\tilde{\mathcal{O}}(\sqrt{T})$ is recovered for the learning component, ensuring sample efficiency up to the limit imposed by parameter precision. We will add a discussion on this sensitivity analysis in the revised version.

---

> ### Author Response · Authors · 2025-11-22
> **Response to Reviewer coiB (continued)**
>
> **4. Alternative Model: Recommendation Neighborhood of Radius $r$**
>
> We find your suggestion of a “neighborhood” constraint insightful and note that our model is effectively a “soft”, differentiable generalization of exactly this idea. While a radius constraint $r$ imposes a “hard” boundary, our model uses the KL divergence term $-\tau D(\nu || y)$ as a “soft” regularization. Mathematically, this soft regularization acts similarly to the Lagrangian relaxation of a hard constraint, penalizing the follower for deviating too far from the recommendation $y$ and thereby constraining the response to a probability neighborhood of $y$. The advantage of our OMD formulation is that it remains analytically smooth and allows for gradient-based analysis while capturing the same behavioral intuition of staying close to the recommendation.
>
>
> **Final Remarks**
>
> We thank you again for your time and insightful feedback. We hope these clarifications demonstrate that our model is deeply rooted in established behavioral game theory and that the assumptions are consistent with standard theoretical frameworks. If you feel that our responses have resolved your concerns and questions, we would be truly grateful if you would consider raising your assessment score and confidence. Thank you very much!
>
> **Reference**
>
> McKelvey and Palfrey (1995). Quantal response equilibria for normal form games.
>
> Chen et al. (2023). Actions speak what you want: Provably sampleefficient reinforcement learning of the quantal stackelberg equilibrium from strategic feedbacks.
>
> Yang et al. (2012). Computing Optimal Strategy against Quantal Response in Security Games.
>
> Zhao et al. (2023). Online learning in stackelberg games with an omniscient follower.
>
> Kamenica and Gentzkow (2011). Bayesian Persuasion.
>
> Jin et al. (2020). Provably efficient reinforcement learning with linear function approximation.

---

> > ### Comment · Reviewer_coiB · 2025-11-24
> >
> > Thank you for your response. I would discuss with other reviewers to decide my final rating.

---

> ### Author Response · Authors · 2025-12-03
> **Detailed Response on Parameter Robustness (New Section 6.2)**
>
> Dear Reviewer coiB,
>
> We are writing to provide a detailed update specifically addressing your concerns regarding the assumption of known behavioral parameters $\eta$ and $\tau$. We fully agree that analyzing the impact of parameter estimation errors is crucial for the practical applicability and theoretical completeness of our framework.
>
> In the revised manuscript, we have added a new **Section 6.2: Robustness to Parameter Misspecification** to rigorously address this issue. We are happy to report that our analysis confirms our algorithm exhibits graceful degradation under parameter misspecification.
>
> Below we summarize the new theoretical results and their implications:
>
> **1. New Theoretical Guarantee (Theorem 6.4)**
>
> We consider a setting where the leader uses estimated parameters $\hat{\eta}, \hat{\tau}$ such that $|\hat{\eta} - \eta| \le \zeta$ and $|\hat{\tau} - \tau| \le \zeta$. We prove that with probability at least $1-p$, both the cumulative regret and ICV regret satisfy:
> $$\text{Reg}(T) \le \tilde{\mathcal{O}}(d^2\sqrt{T}) + \tilde{\mathcal{O}}(\zeta T) \quad \text{and} \quad \text{Reg}^{\text{ICV}}(T) \le \tilde{\mathcal{O}}(d^2\sqrt{T}) + \tilde{\mathcal{O}}(\zeta T).$$
> Detailed proofs, which leverage the Lipschitz continuity of the follower's response model, are provided in the new **Appendix F.5**.
>
> **2. Interpretation: Graceful Degradation & Intrinsic Bias**
> * **Controlled Bias:** The additional term $\tilde{\mathcal{O}}(\zeta T)$ is linear in $T$, implying that the average regret converges to $\mathcal{O}(\zeta)$. This theoretically confirms that the algorithm is **robust**: small estimation errors do not cause catastrophic failure but instead lead to a policy that is $\mathcal{O}(\zeta)$-optimal.
> * **Alignment with Standard Theory:** This result parallels the robustness guarantees for misspecified linear MDPs (e.g., **Jin et al., 2020**), where the regret bound naturally includes an inevitable $\mathcal{O}(\zeta T)$ term representing the intrinsic modeling bias. Our result shows that our algorithm achieves the best possible performance allowed by the parameter precision.
>
> **3. Advantage: Robustness in High Dimensions**
>
> A key insight from our new analysis is that, unlike the misspecification term in linear MDPs which typically scales with the feature dimension $d$, our linear bias term $\tilde{\mathcal{O}}(\zeta T)$ is **independent of $d$**.
> * This is because $\zeta$ here characterizes the error of scalar behavioral parameters, and the sensitivity of the follower's response is bounded by the reward magnitude rather than the feature space dimension.
> * **Implication:** This property renders our algorithm particularly robust in **large-scale and high-dimensional problems**, as the system bias remains controlled even as the feature dimension increases.
>
> We hope this rigorous analysis fully resolves your concerns regarding the realism and sensitivity of the parameter assumptions.
>
> Sincerely,
>
> The Authors

---

### Official Review · Reviewer_Zcy7 · 2025-10-30

**Soundness:** 3
**Presentation:** 3
**Contribution:** 3
**Rating:** 2
**Confidence:** 4

**Summary:**

The paper introduces a model of leader-follower games with actions and policy recommendations, in the spirit of models with adverse selection and moral hazard, compare [Myerson 1983](https://www.jstor.org/stable/1912116?seq=4). They consider one-step mirror descent and quantal responses as equilibrium concepts, and provide regret bounds for the computation of optimal policies. The main claimed contributions are a novel incentive compatibility notion and a first efficient algorithm for incentive-compatible decision-making.

**Strengths:**

The paper is introducing their model carefully, and proposes regret bounds. I could follow the complete specification of the model and their regret statements.

**Weaknesses:**

I take issue with both of their claimed contributions (the rest of the paper seems to be rather standard in terms of techniques):
1. The claimed incentive condition relating to policy recommendation is well-known in the literature of mechanism design [Myerson 1983](https://www.jstor.org/stable/1912116?seq=4) (original formalism) [Bergemann 2019](https://www.aeaweb.org/articles?id=10.1257/jel.20181489) (survey of information design) [Kamenica, Gentzkow 2011](https://web.stanford.edu/~gentzkow/research/BayesianPersuasion.pdf) (a main application of recommendation-based incentive constraints) and called the *obedience* constraint. There are several methods for its computation. The paper does not relate to this literature. Engaging with it would also help the paper by not requiring to impose obedience in Section 4.1 but by saying that it is a consequence of a revelation principle.
2. The authors claim that their algorithm is the first efficient algorithm for incentive-compatible decision-making seems too broad to me. Would [Zhang et al. '24](https://dl.acm.org/doi/abs/10.1145/3670865.3673536) (which also recommends policies) be an efficient algorithm? Is an efficient algorithm in terms of the description size of the game or purely the time horizon?

If the novelty for both of these contributions can be clarified/demonstrated, I am willing to raise my score.

**Questions:**

- Please engage with the literature on information design, mechanism design, and Bayesian persuasion.
- Qualify the statement on the novelty of the efficient algorithm.
- Would [Zhang et al. '24](https://dl.acm.org/doi/abs/10.1145/3670865.3673536) (which also recommends policies) be an efficient algorithm? Is an efficient algorithm in terms of the description size of the game or purely the time horizon?

---

> ### Author Response · Authors · 2025-11-24
> **Response to Reviewer Zcy7**
>
> Dear Reviewer Zcy7,
>
> We sincerely thank you for your critical review and for pointing out the connections to the classical mechanism design and information design literature. We appreciate the opportunity to clarify the positioning of our work, particularly regarding the obedience constraint and the distinction from Zhang et al. (2024).
> We address your two main concerns below, demonstrating that while our concepts align with classical literature (which we will cite in the revised version), our learning problem and algorithmic contributions are fundamentally distinct and novel.
>
> **1. Connection to Mechanism Design and Information Design (Response to Weakness 1 and Question 1)**
>
> You are correct that our IC condition corresponds to the obedience constraint found in the mechanism design (Myerson, 1983) and information design literature (Kamenica & Gentzkow, 2011; Bergemann & Morris, 2019). We appreciate you pointing this out.
>
> We would like to clarify that our modeling choice is not arbitrary but is grounded in the revelation principle for information design. As established in the literature you cited, for any equilibrium outcome achievable by a complex communication mechanism, there exists a direct recommendation mechanism where the receiver is obedient (i.e., the obedience constraint holds). Therefore, imposing the IC (obedience) constraint is without loss of generality and serves as a standard canonical representation for this class of games, rather than an unmotivated assumption.
>
> In the revised version, we will adopt the standard term “obedience constraint”, cite these seminal works, and clarify that our formulation is a natural consequence of the revelation principle.
>
> **2. Novelty and Technical Innovations: Learning under Obedience (Response to Weakness 1 and Question 1)**
>
> While the concept of obedience is standard in static design where the model is known, our contribution lies in online learning under this constraint when the follower's reward function is unknown.
>
> $\quad$ ***(1) Distinction from classical literature***: The cited works (Myerson, 1983; Kamenica & Gentzkow, 2011) characterize optimal mechanisms/signals assuming the designer knows the agent's utility. They do not address how to learn the optimal mechanism from scratch using only bandit feedback while maintaining obedience, which is the core algorithmic challenge we solve.
>
> $\quad$ ***(2) Our technical innovation***: The core challenge we solve is not defining the constraint, but optimizing under it in an online setting where the constraint itself (the follower's response model) must be estimated.
>
> $\qquad$ *(i) Absolute regret analysis*: We developed a novel absolute regret analysis (Theorem 6.2) to handle the negative suboptimality issue that arises when the estimated obedience constraint is violated.
>
> $\qquad$ *(ii) Distribution shift*: We derived a sublinear bound for the distance between the estimated and true quantal responses (distribution shift), which is a specific challenge in learning that does not appear in static mechanism design.

---

> ### Author Response · Authors · 2025-11-24
> **Response to Reviewer Zcy7 (continued)**
>
> **3. Novelty of Efficient Algorithm and Comparison to Zhang et al. (2024) (Response to W Weakness 2 and Questions 2&3)**
>
> You asked if Zhang et al. (2024) constitutes a prior efficient algorithm for our problem, and asked for clarification on our definition of efficiency.
>
> ***(1) Fundamental Differences from Zhang et al. (2024)***
>
> Zhang et al. (2024) and our work address fundamentally different problems, with different mechanisms, information structures and agent models. They are not directly comparable, nor does one subsume the other. The algorithm in Zhang et al. (2024) cannot solve our problem due to three critical distinctions:
>
> $\quad$ ***(i) Mechanism (payments vs. information)***: Zhang et al. (2024) rely on monetary payments to modify players' payoffs. However, our leader cannot pay the follower; she must use policy recommendation (information) to incentivize the follower via alignment (obedience). These are distinct design spaces.
>
> $\quad$ ***(ii) Information structure (known vs. unknown rewards)***: In Zhang et al. (2024), the mediator typically knows the game structure (Definition 3.1: "The mediator knows the game $\Gamma$"). In contrast, our primary challenge is information asymmetry: the leader does not know the follower's reward function and must infer it from interactions. The algorithms proposed by Zhang et al. (2024) cannot solve our learning problem because they rely on known game structures to compute payments and do not account for the statistical estimation of the unknown obedience constraint (follower's response model).
>
> $\quad$ ***(iii) Agent model (no-regret vs. quantal response)***: Zhang et al. (2024) assume agents run no-regret algorithms (learning over time). We assume the follower is a boundedly rational agent (with quantal response) who responds instance-by-instance. This necessitates fundamentally different analysis techniques (e.g., handling distribution shifts in quantal responses) compared to analyzing the convergence of no-regret dynamics.
>
> ***(2) Definition of "Efficient Algorithm"***
>
> Our algorithm is provably efficient in terms of both the description size and the time horizon, and is also computationally efficient.
>
> $\quad$ ***(i) Sample efficiency (description size & time horizon)***: Our regret bound $\tilde{\mathcal{O}}(d^2\sqrt{T})$ is polynomial in the feature dimension $d$ (which characterizes the description size of the game) and sublinear in the time horizon $T$. This ensures that the number of samples required to find an $\epsilon$-optimal policy scales polynomially with the problem size (description size).
>
> $\quad$ ***(ii) Computational efficiency***: Our Scheme 2 (Eq. 18) involves solving a convex optimization problem (MLE) and a specific planning problem, both of which are computationally tractable. The MLE step (Eq. 15) exploits the convexity of the quantal response model under linear rewards, which guarantees polynomial-time solvability via standard convex optimization methods. The planning step (Eq. 18) utilizes a point estimate (from MLE) rather than optimizing over a confidence set. This simplifies the problem from a complex bilevel robust optimization into a standard parameterized policy optimization, which is tractable given the finite action space.
>
> $\quad$ ***Conclusion***: Zhang et al. (2024) solves a steering problem via payments for no-regret agents in a known game, but does not solve the problem of learning the Stackelberg equilibrium via recommendations under strict obedience constraints for quantal response agents with unknown rewards. Therefore, our claim of providing the first provably efficient algorithm for this specific setting (LFG with policy recommendation) remains valid.
>
> **Final Remarks**
>
> We thank you again for your time and the opportunity to clarify these connections. Based on your feedback, we will revise the paper to adopt standard mechanism design terminology (obedience constraint) with appropriate citations. We will also distinguish our setting from Zhang et al. (2024) to avoid confusion regarding novelty. We hope these clarifications demonstrate that while our definitions are grounded in classical theory, our learning results and algorithms address a novel and distinct gap in the literature.
>
> If you feel that our responses have successfully resolved your concerns and questions, we would be truly grateful if you would consider raising your assessment score and confidence. Thank you very much!

---

> ### Author Response · Authors · 2025-12-03
> **Revisions regarding Mechanism Design Connections & Comparison to Zhang et al. (2024)**
>
> Dear Reviewer Zcy7,
>
> Thank you for pointing out the connections to the classical mechanism design and information design literature. We appreciate the opportunity to clarify the positioning of our work, particularly regarding the obedience constraint and the distinction from Zhang et al. (2024).
>
> We address your main concerns in the following three points, demonstrating that while our concepts align with classical literature (which we cite in the revised version), our learning problem and algorithmic contributions are fundamentally distinct and novel.
>
> ### **1. Connection to Mechanism Design and Information Design (Response to Weakness 1 & Question 1)**
>
> We appreciate you pointing out that our IC condition corresponds to the **obedience constraint** found in the mechanism design (Myerson, 1983) and information design literature (Kamenica & Gentzkow, 2011; Bergemann & Morris, 2019). We would like to clarify that our modeling choice is not arbitrary but is grounded in the **revelation principle** for information design. As established in the literature you cited, for any equilibrium outcome achievable by a complex communication mechanism, there exists a direct recommendation mechanism where the receiver is obedient (i.e., the obedience constraint holds). Therefore, imposing the IC (obedience) constraint is without loss of generality and serves as a standard canonical representation for this class of games, rather than an unmotivated assumption.
>
> In the revised version, we adopt the standard term “obedience constraint”, cite these seminal works, and clarify that our formulation, **a direct recommendation mechanism satisfying obedience**, is a natural consequence of the revelation principle.
>
> ### **2. Novelty and Technical Innovations: Learning under Obedience (Response to Weakness 1 & Question 1)**
>
> While the concept of obedience is standard in static design where the model is known, our contribution lies in online learning under this constraint when the follower's reward function is unknown.
>
> * **Distinction from Classical Literature:** The cited works (Myerson, 1983; Kamenica & Gentzkow, 2011) characterize optimal mechanisms/signals assuming the designer *knows* the agent's utility. They do not address how to *learn* the optimal mechanism from scratch using only bandit feedback while maintaining obedience, which is the core algorithmic challenge we solve.
> * **Our Technical Innovation:** The core challenge we solve is not defining the constraint, but optimizing under it in an online setting where the constraint itself (the follower's response model) must be estimated.
>     1.  **Absolute Regret Analysis:** We developed a novel absolute regret analysis (Theorem 6.2) to handle the "negative suboptimality" issue that arises when the estimated obedience constraint is violated.
>     2.  **Distribution Shift:** We derived a sublinear bound for the distance between the estimated and true quantal responses (distribution shift), which is a specific challenge in learning that does not appear in static mechanism design.
>
> ### **3. Novelty of Efficient Algorithm and Comparison to Zhang et al. (2024) (Response to Weakness 2 & Questions 2-3)**
>
> You asked if Zhang et al. (2024) constitutes a prior efficient algorithm for our problem, and asked for clarification on our definition of efficiency.
>
> **(1) Fundamental Differences from Zhang et al. (2024)**
>
> Zhang et al. (2024) and our work address fundamentally different problems, with different mechanisms, information structures, and agent models. Their algorithm cannot solve our problem due to three critical distinctions:
> * **Mechanism (Payments vs. Information):** Zhang et al. (2024) rely on **monetary payments** to modify players' payoffs. Our leader cannot pay the follower; she must use policy recommendation (Information) to incentivize the follower via alignment (Obedience). These are distinct design spaces.
> * **Information Structure (Known vs. Unknown Rewards):** In Zhang et al. (2024), the mediator typically knows the game structure (Definition 3.1: *"The mediator knows the game $\Gamma$"*). In contrast, our primary challenge is **information asymmetry**: the leader *does not know* the follower's reward function and must infer it from interactions. The algorithms proposed by Zhang et al. (2024) cannot solve our learning problem because they rely on known game structures to compute payments and do not account for the statistical estimation of the unknown obedience constraint.
> * **Agent Model (No-Regret vs. Quantal Response):** Zhang et al. (2024) assume agents run **no-regret algorithms** (learning over time). We assume the follower is a **boundedly rational agent (quantal response)** who responds instance-by-instance. This necessitates fundamentally different analysis techniques (e.g., handling distribution shifts in quantal responses) compared to analyzing the convergence of no-regret dynamics.

---

> ### Author Response · Authors · 2025-12-03
> **Revisions regarding Mechanism Design Connections & Comparison to Zhang et al. (2024) (continued)**
>
> **(2) Definition of "Efficient Algorithm"**
>
> Our algorithm is provably efficient in terms of both the description size and the time horizon, and is also computationally efficient.
> * **Sample efficiency (description size & time horizon):** Our regret bound $\tilde{\mathcal{O}}(d^2\sqrt{T})$ is polynomial in the feature dimension $d$ (which characterizes the **description size** of the game) and sublinear in the **time horizon** $T$. This ensures that the number of samples required to find an $\epsilon$-optimal policy scales polynomially with the problem size (description size).
> * **Computational efficiency:** Our **Scheme 2** (Eq. 18) involves solving a convex optimization problem (MLE) and a specific planning problem, both of which are computationally tractable. The MLE step exploits the convexity of the quantal response model under linear rewards, which guarantees polynomial-time solvability via standard convex optimization methods. The planning step (Eq. 18) utilizes a point estimate from MLE rather than optimizing over a confidence set. This simplifies the problem from a complex bilevel robust optimization into a standard parameterized policy optimization, which is tractable given the finite action space.
>
> **Conclusion:** Zhang et al. (2024) solves a steering problem via payments for no-regret agents in a known game, but does not solve the problem of learning the Stackelberg equilibrium via recommendations **under strict obedience constraints** for quantal response agents with unknown rewards. Therefore, our claim of providing the first provably efficient algorithm for *this specific setting* (LFG with policy recommendation) remains valid.
>
> ### **Final Remarks**
>
> Based on your feedback, we revise the paper to adopt standard mechanism design terminology (obedience constraint) with appropriate citations. We also explicitly distinguish our setting from Zhang et al. (2024) to avoid confusion regarding novelty. We hope these clarifications demonstrate that while our definitions are grounded in classical theory, our learning results and algorithms address a novel and distinct gap in the literature.

---

### Official Review · Reviewer_vHXB · 2025-11-03

**Soundness:** 3
**Presentation:** 3
**Contribution:** 3
**Rating:** 6
**Confidence:** 3

**Summary:**

This paper studies learning in leader–follower Stackelberg games with partial feedback, where the leader’s reward function is unknown to both agents, while the follower’s reward function is known only to herself. In each round, the leader announces a policy together with a recommended policy for the follower. The follower then updates her policy via a single-step mirror descent. The paper introduces a dynamic incentive-compatibility (IC) constraint on the sequence of recommended policies over time, and proposes an efficient algorithm that achieves $\tilde{O}(\sqrt{T})$ regret with respect to both the leader’s objective and the incentive-compatibility violation (ICV) regret.

**Strengths:**

1.The paper introduces an interesting setting of learning in Stackelberg games with policy recommendations. The idea of incorporating a recommended policy and incentive-compatibility (IC) constraints is novel in this learning framework.

2.The proposed algorithm is well-designed, and the theoretical results and proofs are clearly presented.

3. The paper is well written and easy to follow.

**Weaknesses:**

In general, I think this is a fairly interesting paper for ICLR. The potential limitations such as the assumptions of finite action spaces, known parameters $\eta$ and $\tau$, and the restriction to matrix games are clearly discussed in the appendix and can reasonably be left for future work.

My main concern lies in the modeling choice: it is unclear whether modeling the follower with a single-step OMD is the most appropriate approach. In the current finite-action setting, this step does not appear to offer computational advantages over directly computing the quantal best response. I also wonder whether it would make more sense to consider a simpler setting where the follower exactly follows the recommended policy, and what the results would look like in that case.

Minor comments: I suggest that the authors include a summary table listing all key assumptions (e.g., what is known to the leader and follower, whether actions are finite, linearity assumptions, etc.), which would improve readability.

Relevant citation:
Haghtalab, Nika, Chara Podimata, and Kunhe Yang. Calibrated Stackelberg Games: Learning Optimal Commitments against Calibrated Agents. NeurIPS 2023.

**Questions:**

1. Is the computational complexity of a single-step OMD lower than directly computing the quantal best response? If not, what is the benefit for the follower to decide her policy based on y rather than computing a quantal best response?

2. Can the results be generalized to the setting where the follower directly follows the leader’s recommended policy (with similar IC constraints)?

3. Is the $d^2$ dependence in the bound tight?

---

> ### Author Response · Authors · 2025-11-18
> **Response to Reviewer vHXB**
>
> Dear Reviewer vHXB,
>
> We sincerely thank you for your constructive feedback. We are encouraged that you found the paper fairly interesting and our discussion of limitations clear. We will address your concerns and questions as follows, which could hopefully clarify the motivation for our modeling approach and our novel contributions.
>
> **Justification for the OMD Follower Model**
>
> Your main concern is the modeling choice of the follower's response (Eq. 3). You point out that in our specific single-step (matrix game) setting, the computational complexity of an OMD update is not significantly lower than directly computing the quantal response (QR) (Eq. 2). Our motivation for using the OMD-based response model (Eq. 3) is not based on computational advantage (in this specific setting), but on the following aspects:
>
> 1. ***Behavioral Realism and Resource Asymmetry***: Our model is designed to capture a common real-world scenario characterized by resource asymmetry. The leader (e.g., a large manufacturer or platform) is computationally powerful and has a global market view. The follower (e.g., a small retailer or end-user) is resource-constrained (in computation, expertise, or attention). In this asymmetric setup, it is behaviorally unrealistic to assume the "weaker" follower solves a complex, global non-linear optimization (Eq. 2) from scratch every time. Instead, our model (Eq. 3) posits that the follower adopts a computationally simple heuristic: she takes the leader's "trusted" recommendation $y$ as a starting point (or reference) and perform a single improvement step (OMD) to maximize their own utility.
>
> 2. ***The Mechanism of Policy Recommendation***: More importantly, the OMD formulation is what provides the mathematical mechanism for the recommendation $y$ to have any influence. The follower's objective (Eq. 3) naturally arises from performing an OMD update on the standard entropy-regularized objective (Eq. 2), using $y$ as the reference policy. This process is precisely what introduces the KL divergence term $D(\nu || y)$. Without this term, the recommendation $y$ would have no impact on the follower's final policy $\nu$, and the entire concept of "policy recommendation" would be mechanistically void.
>
> 3. ***Theoretical Elegance (Connection at Equilibrium)***: This model elegantly connects the simple heuristic (OMD) to the true game-theoretic equilibrium (modelled as QR). As we prove in Proposition 4.1, the follower's OMD response $\nu_{\theta}^{\pi, y}$ is identical to the quantal response $\mu_{\theta}^{\pi}$ if and only if the IC constraint is satisfied. This means our model correctly captures the follower's optimal behavior at equilibrium while also providing a realistic model for their boundedly rational, heuristic-driven behavior off-equilibrium, which is essential in an online learning setting where the leader is constantly exploring.
>
> In summary, the OMD model (Eq. 3) is a behaviorally justified and mechanistically necessary choice to formally model the influence of a recommendation $y$ on a boundedly rational follower.
>
> **Answer to Question 1**: *Is the computational complexity of a single-step OMD lower than directly computing the quantal best response? If not, what is the benefit for the follower to decide her policy based on $y$ rather than computing a quantal best response?*
>
> As discussed above, in this single-step matrix game, the computational complexity is not the primary benefit. However, this computational benefit is a key motivation when considering the extension to multi-step Markov Games (MGs), as discussed in our Future Work section (Appendix D.2). In an MG, computing the full QR policy (Eq. 2) would require solving a complex, non-linear Bellman-like planning problem over the entire horizon. In contrast, the OMD update (Eq. 3) remains a simple, local, and computationally tractable heuristic, making the computational advantage significant in that richer setting.
>
> In this work, the benefits for the follower are twofold as follows:
>
> 1. ***Behavioral Simplicity***: The follower does not need to compute the global optimum (QR). They can rely on a "trusted" recommendation $y$ provided by the more powerful leader and apply a simple, local improvement step (OMD).
>
> 2. ***Incorporating Leader's Information***: The follower trusts that $y$ is a good policy (as the leader is computationally powerful), so they anchor their decision-making on $y$, as modelled by the KL divergence term.

---

> ### Author Response · Authors · 2025-11-18
> **Response to Reviewer vHXB (continued)**
>
> **Answer to Question 2**: *Can the results be generalized to the setting where the follower directly follows the leader’s recommended policy (with similar IC constraints)?*
>
> This is an interesting question that highlights the generality of our framework. If the follower simply plays $\nu=y$, the IC constraint $\mathrm{ICV}(\pi, y; \theta^* )=0$ (which can be proved to be equivalent to $y = \nu_{\theta^*}^{\pi, y}$) would imply that $y$ must be the optimal solution to the follower's problem (Eq. 3).
>
> By our Proposition 4.1, this fixed-point condition $y = \nu_{\theta^* }^{\pi, y}$ holds if and only if $y = \mu_{\theta^*}^{\pi}$. Therefore, the final optimization problem for the leader (maximizing rewards subject to $y$ being the QR policy) is identical in both cases.
>
> Our model is strictly more general and realistic because it defines the follower's behavior off-equilibrium. In an online learning setting, the leader frequently explores and recommends non-optimal $y$. Our model (Eq. 3) realistically captures how the follower deviates from a "bad" recommendation to maximize their own utility, which is a crucial feedback signal for the leader's learning process.
>
> **Answer to Question 3**: *Is the $d^2$ dependence in the bound tight?*
>
> Our understanding of the $d^2$ dependence is as follows:
>
> 1. ***It matches SOTA upper bound***: Our regret bound of $\tilde{\mathcal{O}}(d^2\sqrt{T})$ (Theorem 6.1) matches the state-of-the-art upper bound established for the foundational QSE learning problem without policy recommendation or IC constraints in Chen et al. (2023). Their analysis for the linear setting (Theorem 6.1) results in an $\tilde{\mathcal{O}}(d^2\sqrt{T})$ bound, where $d$ is the dimension parameter.
>
> 2. ***Contribution***: A key contribution of our work is proving that even with the added complexities of modeling policy recommendation, enforcing IC, and most importantly, analyzing the more challenging absolute regret metric (Theorem 6.2), we can still achieve the same sample efficiency as the standard, unconstrained QSE problem.
>
> 3. ***On Tightness (Lower Bound)***: To the best of our knowledge, whether the $d^2$ dependency is tight (i.e., whether a matching $\Omega(d^2\sqrt{T})$ lower bound exists) remains an open question in the literature. The foundational work by Chen et al. (2023) does not provide a lower bound for the $d^2$ term. Their lower bound (Theorem 2) addresses the "burn-in cost" related to the rationality parameter $\eta$, proving that linear regret is unavoidable for small $T$ when $\eta$ is large, but it does not address the polynomial dependency on $d$.
>
> Therefore, our bound is consistent with the best-known results for this class of problems.
>
> **Response to Other Comments**
>
> ***Summary Table***: This is an excellent suggestion. We agree that a consolidated table of assumptions would be beneficial for readability. We have included a Table 1 in the Appendix A summarizing the key notations and many assumptions (e.g., finite $\mathcal{A}, \mathcal{B}$, linear reward structure, $\vartheta^* , \theta^* \in [0, 1]^{|\mathcal{A}||\mathcal{B}|}$). We will add a summary table of all key assumptions (e.g., known $\eta, \tau$, finite actions, linear games) to the appendix in the revised manuscript to improve readability.
>
> ***Citation***: We appreciate your added reference to Haghtalab et al. (2023). We find it complementary to our work, as it studies a setting where the follower does not observe the leader's current commitment and instead best-responds to a calibrated forecast based on the leader's past actions. In our work, the follower observes the leader's full current strategy (both the commitment $\pi$ and the recommendation $y$). Our core challenge is fundamentally different: we analyze how the leader learns to optimize while satisfying the IC constraint, a constraint necessitated by the new policy recommendation mechanism we introduce. We will add and discuss Haghtalab et al. (2023) in our related work section in the revised version.
>
> **Final Remarks**
>
> We thank you again for your time and insightful feedback. We hope that these clarifications can fully address your concerns and highlight the novelty of our technical and theoretical contributions. If you feel that our responses have successfully resolved your concerns and questions, we would be truly grateful if you would consider raising your assessment score and confidence. Thank you very much!

---

### Author Response · Authors · 2025-12-03
**Summary of Revisions in the Updated Manuscript**

Dear Area Chair and Reviewers,

We thank you again for your constructive comments. We have uploaded a revised manuscript that incorporates your feedback to strengthen the theoretical depth and clarity of the paper.

Below is a summary of the major revisions:

**1. New Section on Robustness to Misspecification (Section 6.2 & Appendix F.5)**
* **Motivation:** To address concerns regarding the assumption of known behavioral parameters ($\eta, \tau$) (raised by **Reviewers coiB and CLjs**).
* **Revision:** We added **Section 6.2**, presenting a new theorem (**Theorem 6.4**) and two corollaries. We rigorously prove that our algorithm exhibits graceful degradation under parameter estimation errors. Specifically, the regret bound incurs an additional linear term $\tilde{\mathcal{O}}(\zeta T)$ proportional to the error $\zeta$. Notably, unlike standard results for misspecified linear MDPs (e.g., Jin et al., 2020) where the bias scales with the feature dimension $d$, our linear bias term is independent of $d$, making our approach particularly robust in high-dimensional settings.
* **Proofs:** Detailed proofs are added in **Appendix F.5**.

**2. Explicit Connection to Mechanism Design and Information Design (Section 1, 2 & Appendix B)**
* **Motivation:** To clarify the novelty and theoretical positioning relative to classical literature (raised by **Reviewer Zcy7**).
* **Revision:**
    * We explicitly adopted the term **"obedience constraint"** in the Introduction and Related Work.
    * We added a new Appendix section (Appendix B: Extended Discussion on Related Work), which provides a detailed comparison with classical Mechanism Design and recent works like Zhang et al. (2024). We highlight that our contribution lies in **online learning under obedience constraints with unknown rewards**, a setting distinct from static design or payment-based steering.

**3. Summary of Key Assumptions (Appendix A)**
* **Motivation:** To improve readability and clarity of the game setting (suggested by **Reviewer vHXB**).
* **Revision:** We added a summary table (Table 2) in Appendix A that lists all key assumptions (e.g., finite action spaces, linear reward structure, bandit feedback, etc.).

**4. Clarifications on Efficiency and Terminology**
* **Revision 1:** We refined the definition of "efficient algorithm" in the text to explicitly refer to both sample efficiency (polynomial in $d$, sublinear in $T$) and computational efficiency (polynomial runtime via convex optimization).
* **Revision 2:** We updated the Abstract, Conclusion, and other sections in the paper, to reflect and discuss the new robustness results.

We believe these revisions significantly improve the quality and robustness of our work, and we trust that they have fully addressed the concerns and questions raised by the reviewers.

Sincerely,

The Authors

---

### Author Response · Authors · 2025-12-03
**Summary to Area Chair**

Dear Area Chair,

We understand you are stepping in under unusual circumstances. To facilitate your efficient assessment, we provide a detailed summary of the **valid critiques we have addressed** through major revisions versus the **misunderstandings we have clarified** with theoretical evidence.

**1. Summary of Revisions (Addressing Reasonable Critiques)**

We are grateful for the constructive feedback that helped us significantly strengthen the paper. We have made the following revisions:

* **Robustness to Parameter Uncertainty (Addressing Reviewers coiB, CLjs):**
    * *Critique:* The assumption of known behavioral parameters ($\eta, \tau$) was perceived as strong.
    * *Our Action:* We added a new **Section 6.2** and **Appendix F.5**, presenting a new theorem (**Theorem 6.4**) and two corollaries. We rigorously prove that our algorithm exhibits graceful degradation under parameter misspecification. The regret bound incurs an inevitable linear term $\tilde{\mathcal{O}}(\zeta T)$ (where $\zeta$ is the error bound), matching standard robustness results in RL literature (e.g., Jin et al., 2020). Notably, we show this linear bias is independent of the feature dimension $d$, highlighting our algorithm's robustness in high-dimensional settings.

* **Connection to Mechanism Design (Addressing Reviewer Zcy7):**
    * *Critique:* The IC concept aligns with the "obedience constraint" in classical literature.
    * *Our Action:* We have explicitly adopted the term "obedience constraint" in the Introduction and Related Work. We added a new **Appendix B** to discuss connections with Myerson (1983) and Kamenica & Gentzkow (2011), explicitly framing our model as a natural consequence of the **Revelation Principle** in information design.

**2. Clarification of Misunderstandings (Addressing Unreasonable/Misplaced Critiques)**

Some lower scores stemmed from fundamental misunderstandings of the problem setting and model motivation. We have fully clarified these points:

* **Model Motivation (Addressing Reviewers coiB, CLjs):**
    The reviewers questioned the justification for "entropy regularization", "one-step mirror descent (OMD)", and the "IC constraint". We clarified that these are not arbitrary choices but are grounded in behavioral game theory and statistical learnability:
    * **Entropy Regularization:** This is not merely an algorithmic tool for convergence, but the foundational representation of bounded rationality known as the quantal Response equilibrium (QRE) (McKelvey & Palfrey, 1995). It captures the intrinsic stochasticity of human decision-making.
    * **OMD & KL Term (Anchoring Effect):** The use of OMD is motivated by resource asymmetry. A computationally constrained follower cannot solve the global optimization problem from scratch. Instead, they treat the leader's recommendation as a trusted "anchor" (reference point) and perform a local update. The KL divergence term is the mathematical consequence of this anchoring effect, reflecting how recommendations effectively guide boundedly rational agents.
    * **IC Constraint (Learnability):** For the leader, enforcing IC is essential for **learnability** and **robustness** in the unknown reward setting. If the leader were to recommend arbitrary policies, the follower's "off-equilibrium" responses would be unstable and hard to predict. The IC constraint stabilizes behavior on a predictable manifold, converting a complex inference problem into a tractable control problem.

* **Novelty vs. Zhang et al. (2024) (Addressing Reviewer Zcy7):**
    * *Misunderstanding:* The reviewer suggested Zhang et al. (2024) might already solve our problem.
    * *Clarification:* This is factually incorrect. Zhang et al. rely on **monetary payments** in **known games** to steer **no-regret learners** (dynamic learning over time). Our work solves a fundamentally different problem: using **policy recommendation** (information) in **unknown games** (information asymmetry) to incentivize **boundedly rational followers** (instance-by-instance response). Algorithms for their setting simply cannot apply to ours due to the lack of reward estimation and obedience optimization.

**3. Core Contributions & Conclusion**

This paper provides the **first provably efficient algorithm** for online Stackelberg learning with **policy recommendation** and unknown follower rewards.
* **Technical Depth:** We solve unique challenges including **distribution shift** (via a novel invariance analysis) and **negative suboptimality** (via **Absolute Regret** analysis, Theorem 6.2).
* **Efficiency:** We achieve $\tilde{\mathcal{O}}(d^2\sqrt{T})$ regret, matching the state-of-the-art upper bound for the unconstrained setting.

We believe the paper is technically sound (Soundness 3/4) and clearly written (Presentation 3/4). With the **robustness analysis added** and **novelty/motivation concerns fully resolved**, we respectfully submit that the paper merits acceptance. Thank you very much.

---

### Meta-Review · Area_Chair_1rMX · 2026-01-05

**Summary:**

The reviewers agree that the paper looks at a relevant and nontrivial problem at the intersection of online learning, Stackelberg games, and incentive constraints, and the technical development is generally careful. Some reviewers find the idea of policy recommendation with an incentive or credibility constraint interesting, and they note that the proposed algorithm does achieve sublinear regret under the stated assumptions. But there are big concerns about novelty, motivation, and positioning. Several reviewers point out that the incentive compatibility notion basically matches the well-known obedience constraint from mechanism design and information design, and this connection wasn’t acknowledged at first, which makes the novelty claim weaker. Other concerns are about the behavioral assumptions for the follower, like entropy regularization, one-step mirror descent, and known behavioral parameters, which some reviewers think are either weakly motivated or unrealistic. The rebuttal clarified these choices and added robustness analysis for parameter misspecification, but overall the committee feels the contribution is incremental compared to existing Stackelberg and quantal response learning work, with limited new conceptual insight. Given the mixed scores and persistent doubts about motivation and novelty, the paper is seen as borderline and does not meet the bar for acceptance.

**Reviewer Concerns:**

The rebuttal helped with some points: it clarified the behavioral motivation, explicitly connected the IC notion to obedience constraints, added robustness results for parameter misspecification, and improved the discussion of related work. These changes fixed some misunderstandings and improved the paper. But the main concerns remain: doubts about how realistic the follower model is, whether the recommendation framework is really necessary, and whether the technical contributions go much beyond prior work in Stackelberg and quantal response learning. Some reviewers still think the novelty claims are overstated.

**Reviewer Scores:**

Reviewers who were positive but cautious will probably keep similar scores after the rebuttal. The more critical reviewers, who questioned novelty and modeling assumptions, will likely stick with low scores since their main concerns weren’t fully addressed. Overall, the score distribution stays mixed and below the acceptance threshold.

---

### Decision · Program_Chairs · 2026-01-26

Reject